# On the Generalization Power of the Overfitted Three-Layer Neural Tangent Kernel Model

**Peizhong Ju**
Department of ECE
The Ohio State University
Columbus, OH 43210
ju.171@osu.edu

**Xiaojun Lin**
School of ECE
Purdue University
West Lafayette, IN 47906
linx@purdue.edu

**Ness B. Shroff**
Department of ECE
The Ohio State University
Columbus, OH 43210
shroff.11@osu.edu

## Abstract

In this paper, we study the generalization performance of overparameterized 3-layer NTK models. We show that, for a specific set of ground-truth functions (which we refer to as the "learnable set"), the test error of the overfitted 3-layer NTK is upper bounded by an expression that decreases with the number of neurons of the two hidden layers. Different from 2-layer NTK where there exists only one hidden-layer, the 3-layer NTK involves interactions between two hidden-layers. Our upper bound reveals that, between the two hidden-layers, the test error descends faster with respect to the number of neurons in the second hidden-layer (the one closer to the output) than with respect to that in the first hidden-layer (the one closer to the input). We also show that the learnable set of 3-layer NTK without bias is no smaller than that of 2-layer NTK models with various choices of bias in the neurons. However, in terms of the actual generalization performance, our results suggest that 3-layer NTK is much less sensitive to the choices of bias than 2-layer NTK, especially when the input dimension is large.

## 1 Introduction

Neural tangent kernel (NTK) models (Jacot et al., 2018) have been recently studied as an important intermediate step to understanding the exceptional generalization power of overparameterized deep neural networks (DNNs). Deep neural networks (DNNs) usually have so many parameters that they can perfectly fit all train data, yet they still have good generalization performance (Zhang et al., 2017; Advani et al., 2020). This seems contradicting to the classical wisdom of "bias-variance-tradeoff" in the statistical machine learning methods (Bishop, 2006; Hastie et al., 2009; Stein, 1956; James & Stein, 1992; LeCun et al., 1991; Tikhonov, 1943). To understand this distinct behavior of DNNs, a recent line of work studies the so-called "double-descent" phenomenon, beginning with overfitted linear models. These results on linear models suggest that the test error indeed decreases again in the overparameterized region, as the model complexity increases beyond the number of samples (Belkin et al., 2018, 2019; Bartlett et al., 2020; Hastie et al., 2019; Muthukumar et al., 2019; Ju et al., 2020; Mei & Montanari, 2019). However, these studies use linear models with simple features such as Gaussian or Fourier features, and hence they fail to capture the non-linearity in neural networks. In contrast, NTK models adopt features generated by non-linear activation functions (i.e., neurons of DNNs), and thus they can be viewed as an intermediate step between simple linear models and DNNs. Along this line, the work in Ju et al. (2021) studies 2-layer NTK models, and shows that the 2-layer NTK model indeed exhibits better and different descent behavior in the overparameterized region, which might be closer to that of an actual neural network.

Motivated by Ju et al. (2021), it is of great interest to understand whether similar insights extend to deeper NTK models. In particular, in this paper we study NTK models with 3 layers. Although

36th Conference on Neural Information Processing Systems (NeurIPS 2022).

both 2-layer and 3-layer NTK models share similar assumptions (e.g., trained weights do not change much from initialization, and features are linearized around the initial state), their difference in structure leads to completely different feature formation. Compared with 2-layer NTK models that only contain one hidden-layer of neurons, 3-layer NTK models have two hidden-layers, which interact in more complex ways not observed in 2-layer NTK models. Specifically, let $p_1$ and $p_2$ denote the number of neurons in the two hidden layers. Then, the ultimate features of the 3-layer NTK models depend on both $p_1$ and $p_2$. This dependency leads to the following questions. First, the width of which layer is more important in governing the descent behavior, $p_1$ or $p_2$? Further, to get better descent behaviors, should $p_1$ and $p_2$ grow at the same speed, or should one of them grow faster than the other? Second, do 3-layer NTK models have any performance advantage over 2-layer NTK models?

To answer these questions, in this paper we study the generalization performance of overfitted min-$\ell_2$-norm solutions for 3-layer NTK models where the middle layer is trained. For a set of learnable functions (which we refer to as the "learnable set"), we provide an upper bound on the test error for finite values of $p_1$ and $p_2$. To the best of our knowledge, this upper bound is the first result that can reveal the dependency of the descent behavior on $p_1$ and $p_2$ separately. We then compare 3-layer NTK with 2-layer NTK with respect to the corresponding learnable set and the actual generalization performance. Our comparison reveals several important differences between 3-layer NTK and 2-layer NTK, in terms of the descent behavior, the size of the learnable set, and the sensitivity of the generalization performance to the choice of bias of the neurons.

**Analyzing the Generalization Error:** First, we show that the generalization error (denoted by the absolute value of the difference between the model output and the ground-truth for a test input) is upper bounded by the sum of several terms on the order of $O(1/\sqrt{n})$ ($n$ denotes the number of training data), $O(1/p_2)$ ($p_2$ denotes the number of neurons in the second hidden-layer), $O(\sqrt[4]{\log p_1/p_1})$ ($p_1$ denotes the number of neurons in the first hidden-layer), plus another term related to the magnitude of noise. Similar to 2-layer NTK (Arora et al., 2019; Ju et al., 2021; Satpathi & Srikant, 2021), our upper bound suggests that when there are infinitely many neurons, the generalization error decreases with the number of samples $n$ at the speed of $\sqrt{n}$ and will approach zero when $n \to \infty$ in the noiseless situation. Further, the noise term will not explode when the number of neurons goes to infinity, which is also similar to that for 2-layer NTK. However, our upper bound also reveals new insights that are different from the results for 2-layer NTK. Specifically, our upper bound decreases slower with respect to the number of neurons in the first hidden-layer $p_1$ at the speed of $\sqrt[4]{(\log p_1)/p_1}$, and decreases faster with respect to the number of neurons in the second[1] hidden-layer $p_2$ at the speed of $1/\sqrt{p_2}$. Further, our upper bounds hold regardless of how fast $p_1$ and $p_2$ increase relative to each other (e.g., they could increase at the same speed, or one could increase faster than the other).

**Characterizing the Learnable Set:** We then show that, even if we only train the middle-layer weights, the learnable set (i.e., the set of ground-truth functions for which the above upper bound holds) of the 3-layer NTK without bias contains all finite degree polynomials, which is strictly larger than that of the 2-layer NTK without bias and is at least as large as the 2-layer NTK with bias. Recently, Geifman et al. (2020); Chen & Xu (2020) show that when all layers are trained, 3-layer NTK leads to exactly the same reproducing kernel Hilbert space (RKHS) as 2-layer NTK with biased ReLU (although they assumed an infinite number of neurons, and did not characterize the descent behavior of the generalization error). Combining with their results, we can draw the conclusion that training only the middle-layer weights is at least as effective as training all layers in 3-layer NTK, in terms of the size of the learnable set.

**Sensitivity to the Choices of Bias:** Even though a similar learnable set can be attained by 3-layer NTK (with or without bias) and 2-layer NTK (with bias), our results suggest that the actual generalization performance can still differ significantly in terms of the sensitivity to the choice of bias, especially when the input dimension $d$ is large. One type of bias setting commonly used in literature (Ghorbani et al., 2021a; Satpathi & Srikant, 2021) is that the bias has a similar magnitude as each element of the input vector, which we refer to as "normal bias". However, we show that such a normal bias setting has a negative impact on the generalization error for overfitted 2-layer NTK when $d$ is large. To avoid this negative impact, it is important to use another type of bias setting where the bias has a similar magnitude as the norm of the whole input vector, which we refer to as

---

[1] In this paper, the first hidden-layer denotes the one closer to the input layer, while the second hidden-layer denotes the one closer to the output layer.

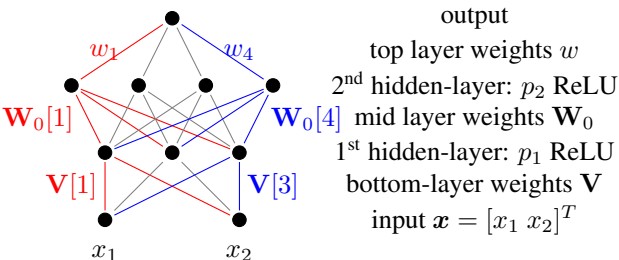

Figure 1: A fully-connected three-layer neural network where input dimension $d = 2$, the number of neurons of the first hidden-layer $p_1 = 3$, and the number of neurons of the second hidden-layer $p_2 = 4$.

"balanced bias". In contrast, for 3-layer NTK, different bias settings do not have an obvious effect on the generalization performance. In summary, compared with 2-layer NTK, the use of an extra non-linear layer in 3-layer NTK appears to significantly reduce the impact due to the choice of bias, and therefore makes the learning more robust.

Our work is related to the growing literature on the generalization performance of such shallow and fully-connected neural network. However, most of these studies focus on 2-layer neural networks. Among them, they differ in which layer to train. For example, Mei & Montanari (2019); d'Ascoli et al. (2020); Mei et al. (2022) consider the "random feature" (RF) model that only trains the top-layer weights and fixes the bottom-layer weights, while 2-layer NTK trains the bottom-layer weights. In contrast, our work on 3-layer NTK neither trains the bottom-layer or top-layer weights. Instead, we train the middle-layer weights, since the middle-layer of a 3-layer model involves the interaction between two hidden-layers, which does not exist in 2-layer models. The above studies of 2-layer network also differ in how the number of neurons/features $p$, the number of training samples $n$, and the input dimension $d$ grow. Mei & Montanari (2019); Mei et al. (2022) study the generalization performance of the RF model where the number of neurons $p$, the number of training data $n$, and the input dimension grow proportionally to infinity. While Ghorbani et al. (2021b) focuses on the approximation error (i.e., expressiveness) of both RF and NTK models, their analysis on generalization error is only on the limit $n$ or $p \to \infty$. All of these studies are quite different from ours with fixed $n$ and finite $p$. Other works such as Arora et al. (2019); Satpathi & Srikant (2021); Fiat et al. (2019) study the situation where the number of training samples $n$ is given and the number of neurons $p$ is larger than a threshold, which is closer to our setup. However, these studies usually do not quantify how the generalization performance depends on the number of neurons $p$. Specifically, they usually provide an upper bound on the generalization error when the number of neurons $p$ is greater than a threshold, while the upper bound itself does not depend on $p$. Thus, such an upper bound cannot explain the descent behavior of NTK models. The work in Ju et al. (2021) does study the descent behavior with respect to $p$, and is therefore the closest to our work. However, as we have explained earlier, there are crucial differences between 2 and 3 layers in both the descent behavior and the learnable set of ground-truth functions. In addition to the above references, our work is also related to Allen-Zhu et al. (2019) (which studies NTK without overfitting) and Ji & Telgarsky (2019) (which studies classification by NTK). Their settings are however different from ours in that we consider overfitted solutions for regression. In summary, our paper is the first to provide a high-probability upper bound on the generalization error of the overfitted 3-layer NTK (where only its middle layer weights are trained), and to characterize how the generalization error decreases with the number of neurons $p_1$ and $p_2$.

## 2    System Model

Let $f : \mathbb{R}^d \mapsto \mathbb{R}$ denote the ground-truth function. Let $(\mathbf{X}_i, f(\mathbf{X}_i) + \epsilon_i)$, $i = 1, 2, \cdots, n$ denote $n$ pieces of training data, where $\mathbf{X} \in \mathbb{R}^{n \times d}$ is the matrix, each column of which is the input of one training sample, $\epsilon \in \mathbb{R}^{n \times 1}$ denotes the noise in the output of training data. We define the training output vector generated by the ground-truth function as $\mathbf{F}(\mathbf{X}) := [f(\mathbf{X}_1) \ f(\mathbf{X}_2) \ \cdots \ f(\mathbf{X}_n)]^T \in \mathbb{R}^n$.

We use $\boldsymbol{a}[j]$ to denote the $j$-th part (sub-vector) of the vector $\boldsymbol{a}$. The part size depends on $\boldsymbol{a}$. Specifically, If $\boldsymbol{a}$ has $p_1$ elements, then each part has 1 elements. If $\boldsymbol{a}$ has $dp_1$ elements, then each part has $d$ elements. If $\boldsymbol{a}$ has $p_1 p_2$ elements, then each part has $p_1$ elements.

We consider a fully-connected 3-layer neural network as illustrated in Fig. 1, which consists of normalized $d$-dimensional input $\boldsymbol{x} \in \mathcal{S}^{d-1}$ (a unit hyper-sphere), $p_1$ ReLUs (rectifier linear units $\max(\cdot, 0)$) at the first hidden-layer, $p_2$ ReLUs at the second hidden-layer, bottom-layer weights (between input and 1st hidden-layer) $\mathbf{V} \in \mathbb{R}^{(p_1 d) \times 1}$, middle-layer weights (between 1st hidden-layer and 2nd hidden-layer) $\mathbf{W}_0 \in \mathbb{R}^{(p_1 p_2) \times 1}$, and top-layer weights $w \in \mathbb{R}^{p_2 \times 1}$ (between 2nd hidden-layer and output).

## 2.1 Overfitted NTK solution

In this subsection, we will derive the overfitted solution for this 3-layer neural network using the NTK approximation. Let $\boldsymbol{h}_{\mathbf{V},\boldsymbol{x}}^{\mathrm{RF}} \in \mathbb{R}^{p_1 \times 1}$ denote the output of the first hidden-layer. We then have

$$\boldsymbol{h}_{\mathbf{V},\boldsymbol{x}}^{\mathrm{RF}}[j] := (\boldsymbol{x}^T \mathbf{V}[j]) 1_{\{\boldsymbol{x}^T \mathbf{V}[j] > 0\}}, \; j = 1, 2, \cdots, p_1. \tag{1}$$

(We use the superscript "RF" because $\boldsymbol{h}_{\mathbf{V},\boldsymbol{x}}^{\mathrm{RF}}$ is indeed the feature vector of a random feature model (Mei & Montanari, 2019).) After training the middle-layer weights, $\mathbf{W}_0$ changes to $\mathbf{W}_1 := \mathbf{W}_0 + \overline{\Delta \mathbf{W}}$. Then, the change of the output is

$$\sum_{k=1}^{p_2} w_k 1_{\{\mathbf{W}_1[k]^T \boldsymbol{h}_{\mathbf{V},\boldsymbol{x}}^{\mathrm{RF}} > 0\}} \mathbf{W}_1[k]^T \boldsymbol{h}_{\mathbf{V},\boldsymbol{x}}^{\mathrm{RF}} - \sum_{k=1}^{p_2} w_k 1_{\{\mathbf{W}_0[k]^T \boldsymbol{h}_{\mathbf{V},\boldsymbol{x}}^{\mathrm{RF}} > 0\}} \mathbf{W}_0[k]^T \boldsymbol{h}_{\mathbf{V},\boldsymbol{x}}^{\mathrm{RF}}.$$

The NTK model (Jacot et al., 2018) assumes that $\overline{\Delta \mathbf{W}}$ is very small and thus the activation pattern does not change much. In other words, we can approximate $1_{\{\mathbf{W}_1[k]^T \boldsymbol{h}_{\mathbf{V},\boldsymbol{x}}^{\mathrm{RF}} > 0\}}$ by $1_{\{\mathbf{W}_0[k]^T \boldsymbol{h}_{\mathbf{V},\boldsymbol{x}}^{\mathrm{RF}} > 0\}}$. Define $\Delta \mathbf{W} \in \mathbb{R}^{(p_1 p_2) \times 1}$ as $\Delta \mathbf{W}[k] := w_k \cdot \overline{\Delta \mathbf{W}}[k]$, $k = 1, 2, \cdots, p_2$. Define $\boldsymbol{h}_{\mathbf{V},\mathbf{W}_0,\boldsymbol{x}}^{\mathrm{Three}} \in \mathbb{R}^{1 \times (p_1 p_2)}$ such that

$$\boldsymbol{h}_{\mathbf{V},\mathbf{W}_0,\boldsymbol{x}}^{\mathrm{Three}}[k] := (\boldsymbol{h}_{\mathbf{V},\boldsymbol{x}}^{\mathrm{RF}})^T \cdot 1_{\{(\boldsymbol{h}_{\mathbf{V},\boldsymbol{x}}^{\mathrm{RF}})^T \mathbf{W}_0[k] > 0\}}, \tag{2}$$

where $k = 1, 2, \cdots, p_2$. Therefore, the change of the output can be approximated by

$$\sum_{k=1}^{p_2} w_k 1_{\{\mathbf{W}_0[k]^T \boldsymbol{h}_{\mathbf{V},\boldsymbol{x}}^{\mathrm{RF}} > 0\}} \overline{\Delta \mathbf{W}}[k]^T \boldsymbol{h}_{\mathbf{V},\boldsymbol{x}}^{\mathrm{RF}} = \boldsymbol{h}_{\mathbf{V},\mathbf{W}_0,\boldsymbol{x}}^{\mathrm{Three}} \Delta \mathbf{W}.$$

We thus obtain a linear model in $\Delta \mathbf{W}$. We provide an illustration of the formation and structure of these vectors in Fig. 4, Appendix A.1 in Supplementary Material. Define the design matrix $\mathbf{H} \in \mathbb{R}^{n \times (p_1 p_2)}$ such that its $i$-th row is $\mathbf{H}_i = \boldsymbol{h}_{\mathbf{V},\mathbf{W}_0,\mathbf{X}_i}^{\mathrm{Three}}$. Notice that overfitted gradient descent on a linear model converges to the min $\ell_2$-norm solution[2], which is denoted by

$$\Delta \mathbf{W}^{\ell_2} := \underset{\boldsymbol{w} \in \mathbb{R}^{(p_1 p_2) \times 1}}{\arg \min} \|\boldsymbol{w}\|_2 \text{ subject to } \mathbf{H} \boldsymbol{w} = \mathbf{F}(\mathbf{X}) + \boldsymbol{\epsilon}.$$

When $\mathbf{H}$ is full row-rank (which holds with high probability under certain conditions), the trained model is then

$$\hat{f}^{\ell_2}(\boldsymbol{x}) = \boldsymbol{h}_{\mathbf{V},\mathbf{W}_0,\boldsymbol{x}}^{\mathrm{Three}} \Delta \mathbf{W}^{\ell_2} = \boldsymbol{h}_{\mathbf{V},\mathbf{W}_0,\boldsymbol{x}}^{\mathrm{Three}} \mathbf{H}^T (\mathbf{H} \mathbf{H}^T)^{-1} (\mathbf{F}(\mathbf{X}) + \boldsymbol{\epsilon}). \tag{3}$$

Notice that the trained model is determined by multiple random variables.

In order to analyze the generalization performance of the trained model, we have to make assumptions on the distribution of those random variables. Let $\mu(\cdot)$, $\lambda(\cdot)$, and $\gamma(\cdot)$ denote the probability density function of $\boldsymbol{x}$, $\mathbf{V}[j]$, and $\mathbf{W}_0[k]$, respectively. For simplicity, we make the following assumption that all random variables follow uniform distribution.

---

[2] As suggested by other prior works Satpathi & Srikant (2021); Hastie et al. (2019); Ju et al. (2021), if we perform gradient descent training on a linear model from zero initial point until the training error is zero (i.e., overfitting), then the solution will be exactly the min $\ell_2$-norm solution. Note that we do not need to be concerned about the training dynamics here, because the min $\ell_2$-norm overfitted solution can be written down exactly as in Eq. (3).

**Assumption 1.** *The input $\boldsymbol{x}$ and the bottom-layer initial weights $\mathbf{V}[j]$'s ($j = 1, 2, \cdots, p_1$) are i.i.d. and uniformly distributed in $\mathcal{S}^{d-1}$. In other words, $\mu(\cdot)$ and $\lambda(\cdot)$ are both $\mathsf{unif}(\mathcal{S}^{d-1})$. The middle-layer initial weights $\mathbf{W}_0[k]$'s ($k = 1, 2, \cdots, p_2$) are i.i.d. and uniformly distributed in $\mathcal{S}^{p_1-1}$. In other words, $\gamma(\cdot)$ is $\mathsf{unif}(\mathcal{S}^{p_1-1})$. The top-layer weights $w$ are all non-zero[3].*

*Remark* 1. Readers may be curious why we only train the middle-layer weights. Part of the reason is technicality: if the bottom layer is also trained, the aggregate output of the first hidden-layer may have changed so much that the second hidden-layer's inputs and ReLU activation patterns change significantly from initialization, which may violate the NTK assumption. The work in Geifman et al. (2020); Chen & Xu (2020) is not concerned about this difficulty, since they are mostly interested in the expressive power of the RKHS, assuming an infinite number of neurons. In contrast, we wish to capture the effect of finite width, and thus train only the middle layer to avoid this difficulty. More importantly, the middle-layer weights interact with both the first hidden layer and the second hidden-layer, and are the major structural distinction compared with 2-layer NTK. This setting thus helps us to answer the following interesting question: will training the middle layer alone already achieve the same (potential) benefit as training all layers (especially given that the latter encounters more technical difficulty)?

## 3 Generalization Performance

In this section, we will show our main results about the generalization performance of the aforementioned 3-layer NTK model for a specific set of functions. We first introduce a set of ground-truth functions that may be learnable and then provide a high-probability upper bound on the test error. We then discuss some useful implications of our upper bound.

### 3.1 A set of ground-truth functions that may be learnable

We define kernel functions $K^{\mathrm{RF}}$, $K^{\mathrm{Two}}$, and $K^{\mathrm{Three}} : [-1, 1] \mapsto \mathbb{R}$ as follows (whose meanings will be explained soon):

$$K^{\mathrm{RF}}(a) := \frac{\sqrt{1-a^2} + a \cdot (\pi - \arccos(a))}{2d\pi}, \tag{4}$$

$$K^{\mathrm{Two}}(a) := a \cdot \frac{\pi - \arccos(a)}{2\pi}, \tag{5}$$

$$K^{\mathrm{Three}}(a) := \frac{K^{\mathrm{Two}}\left(2d \cdot K^{\mathrm{RF}}(a)\right)}{2d}. \tag{6}$$

(Notice that $2d \cdot K^{\mathrm{RF}}(a) \in [0, 1]$ for all $a \in [-1, 1]$ by Lemma 43 in Supplementary Material, Appendix I, and hence $K^{\mathrm{Three}}(\cdot)$ is well defined.) We define a set $\mathcal{F}_{(3)}^{\ell_2}$ of ground-truth functions based on those kernels:

**Definition 1** (learnable set of 3-layer NTK).

$$\mathcal{F}_{(3)}^{\ell_2} := \left\{ f_g : \mathcal{S}^{d-1} \mapsto \mathbb{R} \;\middle|\; f_g(\boldsymbol{x}) = \int_{\mathcal{S}^{d-1}} K^{\mathit{Three}}(\boldsymbol{x}^T \boldsymbol{z}) g(\boldsymbol{z}) d\mu(\boldsymbol{z}), \; \|g\|_\infty < \infty \right\}, \tag{7}$$

*where* $\|g\|_\infty := \sup_{\boldsymbol{z} \in \mathcal{S}^{d-1}} |g(\boldsymbol{z})|$.

To see why functions in $\mathcal{F}_{(3)}^{\ell_2}$ may be learnable, we can check what the learned result $\hat{f}^{\ell_2}$ in Eq. (3) should look like. When there are infinite number of neurons and there is no noise (i.e., $\epsilon = \mathbf{0}$), what remains on the right-hand-side of Eq. (3) can be viewed as the product of two terms, $\boldsymbol{h}_{\mathbf{V},\mathbf{W}_0,\boldsymbol{x}}^{\mathrm{Three}} \mathbf{H}^T$ and $(\mathbf{H}\mathbf{H}^T)^{-1}\mathbf{F}(\mathbf{X})$. For the first term $\boldsymbol{h}_{\mathbf{V},\mathbf{W}_0,\boldsymbol{x}}^{\mathrm{Three}} \mathbf{H}^T$, note that each row of $\mathbf{H}$ is given by $\boldsymbol{h}_{\mathbf{V},\mathbf{W}_0,\mathbf{X}_i}^{\mathrm{Three}}$ for $i = 1, 2, \cdots, n$. Thus, when $p_1, p_2 \to \infty$, the $i$-th element of $\boldsymbol{h}_{\mathbf{V},\mathbf{W}_0,\boldsymbol{x}}^{\mathrm{Three}} \mathbf{H}^T$, which is the inner product between $\boldsymbol{h}_{\mathbf{V},\mathbf{W}_0,\boldsymbol{x}}^{\mathrm{Three}}$ and $\boldsymbol{h}_{\mathbf{V},\mathbf{W}_0,\mathbf{X}_i}^{\mathrm{Three}}$, converges in probability to $K^{\mathrm{Three}}(\boldsymbol{h}_{\mathbf{V},\mathbf{W}_0,\boldsymbol{x}}^{\mathrm{Three}}(\boldsymbol{h}_{\mathbf{V},\mathbf{W}_0,\mathbf{X}_i}^{\mathrm{Three}})^T)$, which is exactly the kernel function of 3-layer NTK. By representing the second term $(\mathbf{H}\mathbf{H}^T)^{-1}\mathbf{F}(\mathbf{X})$ with a certain $g(\cdot)$, $\hat{f}^{\ell_2}$ must then approach the form

---

[3]We do not need to specify the distribution of $w$, since $w$ is absorbed into the regressor $\Delta\mathbf{W}$ by definition $\Delta\mathbf{W}[k] := w_k \cdot \overline{\Delta\mathbf{W}}[k]$, $k = 1, 2, \cdots, p_2$.

in Eq. (7). (See Supplementary Material, Appendix B for details.) Intuitively, $K^{\text{Three}}$ can be thought of as the composition of the kernels of each of the two layers, which are $K^{\text{Two}}$ and $K^{\text{RF}}$ given in Eq. (5) and Eq. (4). Specifically, suppose that we fix the output of the first hidden layer (i.e., $h_{\mathbf{V},\boldsymbol{x}}^{\text{RF}}$) and regard it as the input of a 2-layer NTK formed by the top two layers of the 3-layer neural network. By letting $p_2 \to \infty$, we can show that the inner product between $h_{\mathbf{V},\mathbf{W}_0,\boldsymbol{x}}^{\text{Three}}$ and $h_{\mathbf{V},\mathbf{W}_0,\mathbf{X}_i}^{\text{Three}}$ approaches $K^{\text{Two}}((h_{\mathbf{V},\boldsymbol{x}}^{\text{RF}})^T h_{\mathbf{V},\mathbf{X}_i}^{\text{RF}})$ (with necessary normalization of $h_{\mathbf{V},\boldsymbol{x}}^{\text{RF}}$ and $h_{\mathbf{V},\mathbf{X}_i}^{\text{RF}}$), where $K^{\text{Two}}$ is exactly the kernel of 2-layer NTK in Ju et al. (2021). Second, when $p_1 \to \infty$, we can show that $(h_{\mathbf{V},\boldsymbol{x}}^{\text{RF}})^T h_{\mathbf{V},\mathbf{X}_i}^{\text{RF}}$ approaches $K^{\text{RF}}(\boldsymbol{x}^T \mathbf{X}_i)$, where $K^{\text{RF}}$ is exactly the kernel of the random-feature model (Mei & Montanari, 2019). In summary, we expect that functions in $\mathcal{F}_{(3)}^{\ell_2}$ can be approximated by $\hat{f}^{\ell_2}(\cdot)$. However, we note that the above deviation is only about the expressiveness of 3-layer NTK and it does not precisely reveal its generalization performance.

## 3.2 An upper bound on the generalization error

We now present the first main result of this paper, which is an upper bound that quantifies the relationship between the generalization performance and system parameters.

**Theorem 1.** *For any ground-truth function* $f(x) = f_g(x) \in \mathcal{F}_{(3)}^{\ell_2}$, *when* $d$ *is fixed and* $p_1, p_2$ *are much larger than* $n$, *(with high probability) we have*

$$|\hat{f}^{\ell_2}(\boldsymbol{x}) - f(\boldsymbol{x})| = \underbrace{O\left(\frac{\|g\|_\infty}{\sqrt{n}}\right)}_{\textit{Term A}} + \Bigg(\underbrace{O\left(\frac{\|g\|_1}{\sqrt{p_2}}\right)}_{\textit{Term B}} +$$

$$\underbrace{O\left(\|g\|_1 \sqrt[4]{\frac{\log p_1}{p_1}}\right)}_{\textit{Term C}} + \underbrace{\frac{\|\boldsymbol{\epsilon}\|_2}{\sqrt{n}}}_{\textit{Term D}}\Bigg) \cdot \underbrace{O\left(n^{\frac{2}{d-1}+\frac{1}{2}} \cdot \sqrt{\log n}\right)}_{\textit{Term E}}. \tag{8}$$

A more precise version of the upper bound and the condition of Theorem 1 as well as its derivation can be found in Supplementary Material, Appendix C.

As we can see, Eq. (8) captures how the test error depends on finite values of parameters $n$, $p_1$, $p_2$, $\|\boldsymbol{\epsilon}\|_2$, and $g$. Later in this section we will examine more closely how $n$, $p_1$, and $p_2$ affect the value of the upper bound. Regarding the dependency on $g$, Eq. (8) works as long as $\|g\|_1$ and $\|g\|_\infty$ are finite[4]. Intuitively, the norm of $g$ represents the complexity of the ground-truth function in $\mathcal{F}_{(3)}^{\ell_2}$. When the norm of $g$ is larger, then the right-hand side of Eq. (8) becomes larger, which indicates that such ground-truth function is harder to learn. A simple example is that if we enlarge a ground truth function $f_g \in \mathcal{F}_{(3)}^{\ell_2}$ by 2 times (which means $g$ is 2 times larger), then since the model is linear, the test error $|\hat{f}^{\ell_2}(\boldsymbol{x}) - f(\boldsymbol{x})|$ will become 2 times larger. We will discuss more about which types of functions satisfy the condition of finite norm of $g$ in Section 4.

Next, we will discuss some implications of this upper bound of 3-layer NTK. While some of them are similar to 2-layer NTK, others are significantly different, revealing the complexity due to having more layers.

## 3.3 Interpretations similar to 2-layer NTK

Based on the upper bound in Theorem 3, we have the following insights for 3-layer NTK, which are similar to those for 2-layer NTK shown in Ju et al. (2021). These similarities may reveal some intrinsic properties of the NTK models regardless of the number of layers.

**Zero test error with** $n \to \infty$ **in the ideal situation:** In the ideal situation where there are infinitely many neurons and no noise, the only remaining term in Eq. (8) is Term A. Notice that Term A decreases to zero as $n \to \infty$, which indicates that the generalization error decreases to zero when

---

[4]Indeed, as long as $\|g\|_\infty < \infty$, then $\|g\|_1 < \infty$. That is why we only include the condition $\|g\|_\infty < \infty$ in Eq. (7). Notice that the assumption $\|g\|_\infty < \infty$ can be relaxed to $\|g\|_1 < \infty$ by similar methods showing in Ju et al. (2021). However, as shown in Ju et al. (2021), such relaxation leads to a different upper bound with slower descent speed with respect to $n$.

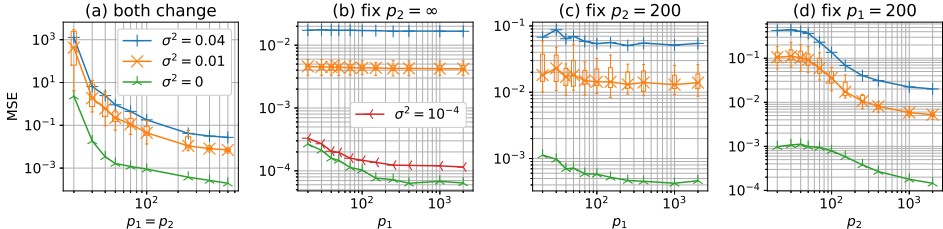

Figure 2: Curves of MSE for 3-layer NTK (no-bias) with respect to $p_1$ or $p_2$ when there exists Gaussian noise whose mean is zero and the variance is $\sigma^2$. The ground-truth function is $f(\boldsymbol{x}) = \left(\boldsymbol{x}^T \boldsymbol{e}_1\right)^2 + \left(\boldsymbol{x}^T \boldsymbol{e}_1\right)^3$ where $d = 3$. Sample size is $n = 200$. Every curve is the median of 20 random simulations.

more training data are provided in the ideal situation. Term A suggests that such decreasing speed is at least $1/\sqrt{n}$. Such result is consistent with that of 2-layer NTK, e.g., in Arora et al. (2019).

### 3.4 Insights that are distinct compared with 2-layer NTK

Compared with 2-layer NTK, an important difference for 3-layer NTK is that there are more than one hidden-layers. Therefore, the speed of the descent of 3-layer NTK involves the interaction between two hidden-layers.

**Descent with respect to the number of neurons:** In Eq. (8), Term B and Term C contain $p_1$ and $p_2$, respectively. For any given $n$ and noise level $\|\boldsymbol{\epsilon}\|_2$, Terms A and D do not change, and Term E decreases with $p_1$ and $p_2$. (More discussion about Term E can be found in Supplementary Material, Appendix D, where we discuss the noise effect.) Therefore, by increasing $p_1$ and $p_2$, Term B and Term C keep decreasing. In summary, right-hand-side of Eq. (8) decreases as the number of neurons $p_1$ and $p_2$ increases, which validates the descent in the overparameterized region of 3-layer NTK.

**Different descent speed:** As shown in Eq. (8), $p_1$ and $p_2$ play different roles in the descent of the generalization error. Comparing Term B and Term C of Eq. (8), we can see that the upper bound of the test error $|\hat{f}^{\ell_2}(\boldsymbol{x}) - f(\boldsymbol{x})|$ decreases faster with respect to $p_2$ (at the speed of $\sqrt{p_2}$) and slower with respect to $p_1$ (at the speed of $\sqrt[4]{p_1/\log p_1}$). We emphasize that this difference is not due to the number of weights/parameters contributed by the number of neurons in each hidden-layer of $p_1$ and $p_2$ [5]. Instead, we conjecture that such difference in the speed of descent may be due to the different positions in this 3-layer neural network structure, where the second hidden-layer takes the trained middle-layer weights as its input (and thus utilizes the trained weights better than the first hidden-layer).

We use numerical results to illustrate the different roles of $p_1$ and $p_2$ in reducing the generalization error. We fix $p_2 = 200$ and plot the MSE with respect to $p_1$ in Fig. 2(b). Although the test error decreases when $p_1$ increases, the decreasing speed is slow, especially for the noisy situation. Such a slow decreasing speed with $p_1$ remains even when $p_2$ is fixed to a much higher value. For example, in Fig. 2(b), we fix $p_2 = \infty$, we still observe the similarly slow decreasing speed with $p_1$ as shown by Fig. 2(c). In contrast, the descent with respect to $p_2$ should be easier to observe and can reach a lower test MSE. In Fig. 2(d), we fix $p_1 = 200$ and increase $p_2$ (i.e., we exchange the values of $p_1$ and $p_2$ in Fig. 2(c)(d)). As we can see, all three curves in Fig. 2(d) have a more obvious descent and decrease to lower MSE compared with those in Fig. 2(c), which validates our conjecture that the descent speed with respect to the number of neurons of the second hidden-layer is faster.

Notice that our upper bound Eq. (8) also suggests a descent when both $p_1$ and $p_2$ increase simultaneously. We use simulation result by Fig. 2(a) to support this point. We fixed $n = 200$ and let $p_1 = p_2$ increase simultaneously. The ground-truth model in this figure is $f(\boldsymbol{x}) = (\boldsymbol{x}^T \boldsymbol{e})^2 + (\boldsymbol{x}^T \boldsymbol{e})^3$ where $d = 3$. The green, orange, and blue curves denote the situations of $\sigma^2 = 0$ (no noise), $\sigma^2 = 0.01$, and $\sigma^2 = 0.04$, respectively. Every point in this figure is the median of 20 simulation runs. We also

---

[5]Specifically, the number of weights that get trained equals to $p_1 p_2$ and the total number of weights for bottom, middle, and top layers equals to $d p_1 + p_1 p_2 + p_2$. In other words, the number of weights (either for the middle layer or for all layers) does not increase faster by increasing $p_2$ instead of $p_1$.

provide the box plot[6] of the situation of $\sigma^2 = 0.01$ (correspond to the orange curve). It is obvious that all three curves descend, which verifies that the generalization error of the overfitted 3-layer NTK model decreases when $p_1$ and $p_2$ increase simultaneously at the same speed. By observing the box plot for the situation $\sigma^2 = 0.01$ (the orange curve), we also notice that when $p_1 = p_2$ becomes large, the variance becomes small. This is because all initial weights are *i.i.d.* random and a large number of weights may reduce the variance of the model due to the law of large numbers. Our upper bound in Theorem 3 also suggests such reduced variance as the probability in Theorem 3 increases as $p_1$ increases.

## 4 Types of Ground-Truth Functions

Are 3-layer (i.e., deeper) networks better than 2-layer networks in any way for generalization performance? In the last section, we have seen that both 3-layer NTK and 2-layer NTK can achieve zero test error when $n \to \infty$ in the ideal noiseless situation, when the ground-truth functions are in their respective learnable set[7]. A natural question is then to compare the learnable sets between these two models, and to compare the generalization performance when the ground-truth function belongs to both learnable sets. In this section, we provide some answers by studying various types of ground-truth functions and their effects on the generalization performance.

### 4.1 Size of the learnable set

For a 2-layer NTK, as shown in Ju et al. (2021), when no bias is used in ReLU, the corresponding learnable set $\mathcal{F}^{\ell_2}_{(2)}$ contains all even polynomials and linear functions, but does not contain other odd polynomials. In order to learn both even and odd polynomials, it is critical that bias is added to ReLU (Satpathi & Srikant, 2021; Ju et al., 2021). In contrast, we prove the following result:

**Proposition 2.** $\mathcal{F}^{\ell_2}_{(3)}$ *(with unbiased ReLU, middle layer being trained) already contains all polynomials with finite degree (i.e., including both even and odd polynomials). Further, the learnable set $\mathcal{F}^{\ell_2}_{(3)}$ of 3-layer NTK is strictly larger than that of the 2-layer NTK with unbiased ReLU, and is at least as large as that of the 2-layer NTK with biased ReLU.*

This independence to bias shown by Proposition 2 can be seen as one performance advantage of 3-layer NTK compared to 2-layer NTK. Details (including more precise statement) about this result is in Supplementary Material, Appendix J. Notice that Geifman et al. (2020); Chen & Xu (2020) show that when training all layers, 3-layer NTK leads to the same RKHS as 2-layer NTK with biased ReLU. However, it is unclear whether training one layer is already sufficient for achieving the same RKHS as training all layers. Our result in Proposition 2 answers this question positively, i.e., only training the middle layer has already achieved all benefits of training all layers in terms of the size of the learnable set. (In other words, training all three layers will not expand the learnable set over training only the middle layer.)

### 4.2 Different bias settings with high input dimension

Even when a ground-truth function belongs to both $\mathcal{F}^{\ell_2}_{(2),b}$ and $\mathcal{F}^{\ell_2}_{(3)}$, their generalization performance may still exhibit some differences. In this subsection, we will show that when the input dimension $d$ is high, some specific choice of bias of the 2-layer NTK has better generalization performance than others. In contrast, the 3-layer NTK is less sensitive to different bias settings.

Notice that adding bias to each ReLU in 2-layer NTK is equivalent to appending a constant to $\boldsymbol{x}$ while still using ReLU without bias. Specifically, the input vector for biased 2-layer NTK is

$$\boldsymbol{x}_b := \begin{bmatrix} \sqrt{1-b^2} \cdot \boldsymbol{x} \\ b \end{bmatrix} \in \mathbb{R}^{d+1}, \tag{9}$$

---

[6]From bottom to top, the five horizontal lines of each marker of a box plot represent the minimum (excluding outliers), first quartile (25%), median (50%), third quartile (75%), and maximum (excluding outliers), respectively. See (McGill et al., 1978) for more details.

[7]We illustrate the generalization performance of ground-truth functions outside the learnable set in Supplementary Material, Appendix J.3.

| Model | Learnable functions set | Category |
|---|---|---|
| 3-layer NTK, no-bias | $\mathcal{F}_{(3)}^{\ell_2} = \left\{\int_{\mathcal{S}^{d-1}} K^{\text{Three}}(\boldsymbol{x}^T\boldsymbol{z})g(\boldsymbol{z})d\mu(\boldsymbol{z})\right\}$ | (i) |
| 2-layer NTK, no-bias | $\mathcal{F}_{(2)}^{\ell_2} = \left\{\int_{\mathcal{S}^{d-1}} K^{\text{Two}}(\boldsymbol{x}^T\boldsymbol{z})g(\boldsymbol{z})d\mu(\boldsymbol{z})\right\}$ | (ii) |
| 2-layer NTK, normal-bias | $\mathcal{F}_{(2),\text{NLB}}^{\ell_2} = \left\{\int_{\mathcal{S}^{d-1}} K^{\text{Two}}(\frac{d}{d+1}\boldsymbol{x}^T\boldsymbol{z} + \frac{1}{d+1})g(\boldsymbol{z})d\mu(\boldsymbol{z})\right\}$ | (i) |
| 2-layer NTK, balanced-bias | $\mathcal{F}_{(2),\text{BB}}^{\ell_2} = \left\{\int_{\mathcal{S}^{d-1}} K^{\text{Two}}(\frac{1}{2}\boldsymbol{x}^T\boldsymbol{z} + \frac{1}{2})g(\boldsymbol{z})d\mu(\boldsymbol{z})\right\}$ | (i) |

Table 1: Learnable functions for different NTK models. Category: (i) can learn both even- and odd-power polynomials; (ii) cannot learn other odd-power polynomials except linear functions. (We omit the condition $\|g\|_\infty < \infty$ in the expression of learnable sets to save space.)

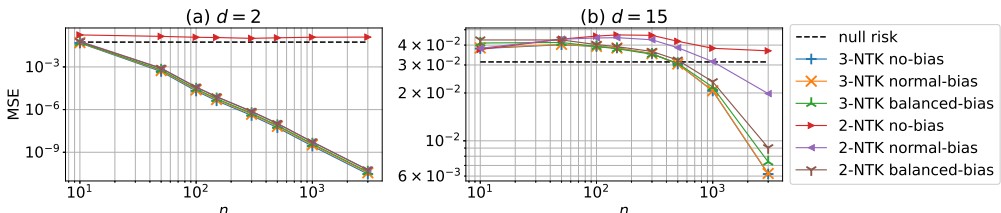

Figure 3: Comparison of test MSE with respect to $n$ between different NTK models when the number of neurons is infinite and without noise. The ground-truth function is $f(\boldsymbol{x}) = \frac{d+2}{3}\left(\boldsymbol{x}^T\boldsymbol{e}_1\right)^3 - \boldsymbol{x}^T\boldsymbol{e}_1$. Every curve is the average of 10 random simulations.

where $b \in (0,1)$ denotes the initial bias. We also normalize the first $d$ elements of $\boldsymbol{x}_b$ by $\sqrt{1-b^2}$ in Eq. (9) to make sure that $\|\boldsymbol{x}_b\|_2 = 1$. Under this biased setting, the 2-layer NTK model has the learnable set $\mathcal{F}_{(2),b}^{\ell_2} := \{\int_{\mathcal{S}^{d-1}} K^{\text{Two}}\left((1-b^2)\boldsymbol{x}^T\boldsymbol{z} + b^2\right)g(\boldsymbol{z})d\mu(\boldsymbol{z}), \|g\|_\infty < \infty\}$.

A common setup for the initial magnitude of the bias of each ReLU is to use a value that is close or equal to the average magnitude of each element of input $\boldsymbol{x}$, e.g., Satpathi & Srikant (2021); Ghorbani et al. (2021a). Specifically, we let $b = \frac{1}{\sqrt{d+1}}$ in Eq. (9), and denote the corresponding learnable set by $\mathcal{F}_{(2),\text{NLB}}^{\ell_2}$. We refer to this setting as the "normal-bias" setting. Alternatively, the initial magnitude of the bias can be chosen to be close or equal to $\|\boldsymbol{x}\|_2$. Specifically, we let $b = \frac{1}{\sqrt{2}}$ in Eq. (9) and denote the corresponding learnable set by $\mathcal{F}_{(2),\text{BB}}^{\ell_2}$. We refer to this second setting as "balanced-bias". The specific expression of $\mathcal{F}_{(2)}^{\ell_2}$, $\mathcal{F}_{(2),\text{NLB}}^{\ell_2}$, and $\mathcal{F}_{(2),\text{BB}}^{\ell_2}$ can be derived by using similar methods shown in Section 3.1 (results are listed in Table 1).

We now discuss how the two different bias settings could affect the generalization performance when $d$ is large. For 2-layer NTK under the normal-bias setting, the kernel is $K^{\text{Two}}(\frac{d}{d+1}\boldsymbol{x}^T\boldsymbol{z} + \frac{1}{d+1})$. Although it contains both even and odd power polynomials, we notice that when $d$ increases, $K^{\text{Two}}$ approaches its no-bias counterpart $K^{\text{Two}}(\boldsymbol{x}^T\boldsymbol{z})$, which only contains even power polynomials and linear term. Thus, we conjecture that, by increasing $d$, the generalization performance of 2-layer NTK with normal-bias will deteriorate for those ground-truth functions inside $\mathcal{F}_{(2),\text{NLB}}^{\ell_2}$ but far away from $\mathcal{F}_{(2)}^{\ell_2}$ (e.g., odd-degree non-linear polynomials). In contrast, for 2-layer NTK under the balanced-bias setting, the kernel is $K^{\text{Two}}(\frac{1}{2}\boldsymbol{x}^T\boldsymbol{z} + \frac{1}{2})$, which does not change with $d$. Therefore, we expect that such deterioration should not happen. Note that in 3-layer NTK, although normal-bias setting still approaches no-bias setting when $d$ increases, there does not exist such performance deterioration, because $\mathcal{F}_{(3)}^{\ell_2}$ (the learnable set of 3-layer NTK without bias) already contains both even and odd power polynomials. These insights will be verified by the numerical results below.

We now use simulation results in Fig. 3 to validate the conjecture that 3-layer NTK models are less sensitive to different bias settings than 2-layer NTK models. We let the ground-truth function be $f(\boldsymbol{x}) = \frac{d+2}{3}\left(\boldsymbol{x}^T\boldsymbol{e}_1\right)^3 - \boldsymbol{x}^T\boldsymbol{e}_1$, which is orthogonal to $\mathcal{F}_{(2)}^{\ell_2}$. In Fig. 3(a) when $d = 2$, all settings have similar performance except 2-layer NTK without bias, whose test error is always above the null risk. In Fig. 3(b) when $d = 15$, the purple curve of 2-layer NTK with normal bias gets closer to the

red curve of 2-layer NTK without bias (and thus the generalization performance becomes worse), while other curves are still close to each other. This validates our conjecture that 3-layer NTK models are less sensitive to different bias settings than 2-layer NTK models. Further simulations can be found in Appendix A.2.

## 5 Conclusion

In this paper, we studied the generalization performance of overfitted 3-layer NTK models. Compared with 2-layer NTK models, 3-layer NTK is less sensitive to different bias settings. Further, training only the middle layer can get most of the performance advantage of 3-layer NTK, in terms of the learnable set. Possible future directions include: (i) studying whether training other layers will get the same benefit as training the middle layer; (ii) approximating the actual neural network where the learned result is far away from the initial state; (iii) investigating deeper network as well as other structures such as convolutional neural network (CNN) and recursive neural network (RNN).

## Acknowledgments and Disclosure of Funding

This work has been supported in part by NSF grants: 2112471 (also partly funded by DHS), CNS-2106933, CNS-1901057, CNS-2113893, and the Bilsland Dissertation Fellowship at Purdue University, and a grant from the Army Research Office: W911NF-21-1-0244.

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
