

through 1st hidden-layer: $\boldsymbol{x}^T\mathbf{V}_0[j]$ → ReLU: $\max(\cdot, 0)$ → $\boldsymbol{h}^{\text{RF}}_{\mathbf{V}_0,\boldsymbol{x}}[j]$

through 2nd hidden-layer: $\boldsymbol{h}^{\text{RF}}_{\mathbf{V}_0,\boldsymbol{x}}$ → $\mathbf{1}_{\{(\boldsymbol{h}^{\text{RF}}_{\mathbf{V}_0,\boldsymbol{x}})^T\mathbf{W}_0[k]>0\}}$ → $\boldsymbol{h}^{\text{Three}}_{\mathbf{V}_0,\mathbf{W}_0,\boldsymbol{x}}[k]$

Figure 4: Formation and structure of $\boldsymbol{h}^{\text{Three}}_{\mathbf{V},\mathbf{W}_0,\boldsymbol{x}}$ and $\boldsymbol{h}^{\text{RF}}_{\mathbf{V},\boldsymbol{x}}$.

# A   Additional Figures

## A.1   Formation of features

In Fig. 4, we illustrate the formation and the structure of the features shown in Section 2.

## A.2   About the conjecture in Section 4.2

We provide some additional simulation results (in addition to Fig. 3) to validate our conjecture in Section 4.2 that 2-layer NTK is more sensitive to different bias settings, especially when $d$ is large. Note that in Fig. 3, we only consider one type of ground-truth functions that contains only odd-power polynomials. Here, we also examine other types of ground-truth functions.

Similar to Fig. 3, in Fig. 5, we consider the ideal case where there are infinite number of neurons. We plot curves of MSE with respect to $n$ when $p, p_1, p_2 \to \infty$. The simulation setup is similar to Fig. 3, but here we consider more types of ground-truth functions (whose exact forms are given in the caption of Fig. 5). In sub-figures (a)(b), Type A function corresponds to even-power polynomials. We can see that all curves are close to each other in both low-dimensional case ($d = 2$) and high-dimensional case ($d = 15$). This is because the 2-layer NTK without bias can learn even-power polynomials. In other words, in high-dimensional cases, although the performance of the normal-bias setting approaches that of the no-bias setting, it does not hurt the generalization performance because the no-bias setting can already learn the Type A function. Sub-figures (c)(d) are exactly the same as Fig. 3, which uses the Type B ground-truth function corresponds to odd-power polynomials. Sub-figures (e)(f) adopt the Type C ground-truth function that contains both odd-power and even-power polynomials. The generalization performance shown by sub-figures (e)(f) is between that in sub-figures (a)(b) and that in sub-figures (c)(d). This is expected because Type C functions can be viewed as a mix of Type A and Type B functions.

We also consider the situation of finite number of neurons. In Fig. 6, we fix the number of training data and let the x-axis be $p$ (for 2-layer NTK) or $p_1$ (for 3-layer NTK with fixed $p_2 = 100$). The setup of Fig. 7 is similar to the setup of Fig. 6 except that for 3-layer NTK we fix $p_1 = 100$ and change $p_2$. Both in Fig. 6 and Fig. 7, when $d$ is large and the ground-truth function is Type B (i.e., sub-figure (d)), we can see that the curve of 2-layer NTK with normal-bias (the purple curve marked by ◀) is closer to the curve of 2-layer NTK without bias (the red curve marked by ▶). This validates our conjecture in Section 4.2 that 2-layer NTK is more sensitive to different bias settings, especially when $d$ is large.

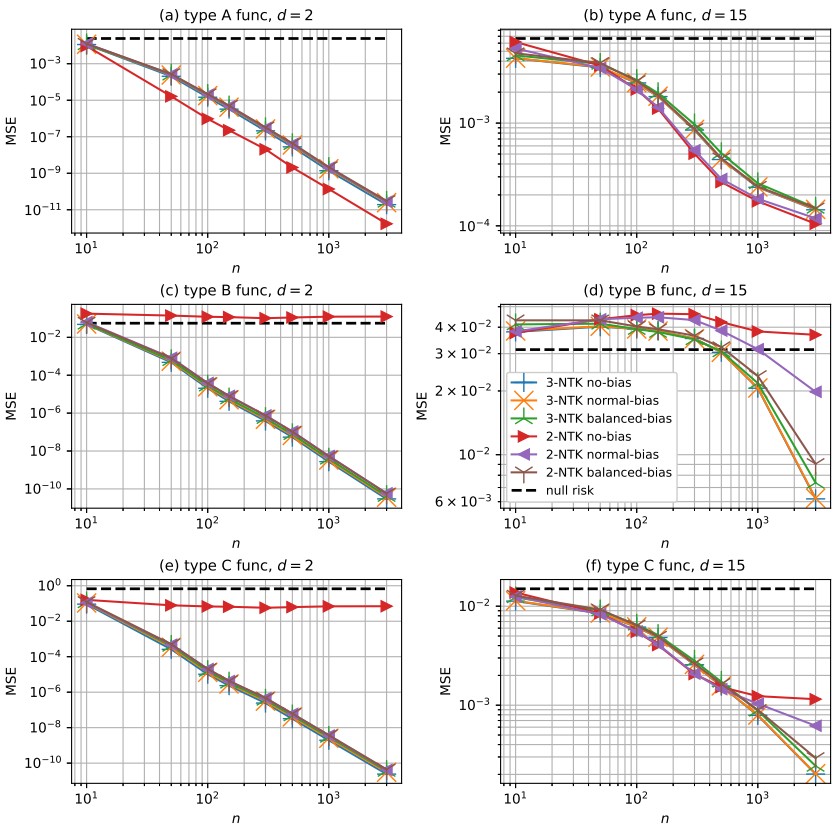

Figure 5: Curves of MSE with respect to $n$ for 2-layer and 3-layer NTK models when $p, p_1, p_2 \to \infty$ and $\boldsymbol{\epsilon} = \boldsymbol{0}$. Let $\boldsymbol{e}_1 = [1\,0\,0\,\cdots\,0]^T \in \mathbb{R}^d$. Type A function is $f(\boldsymbol{x}) = \left(\boldsymbol{x}^T\boldsymbol{e}_1\right)^4 - \left(\boldsymbol{x}^T\boldsymbol{e}_1\right)^2$. Type B function is $f(\boldsymbol{x}) = \frac{d+2}{3}\left(\boldsymbol{x}^T\boldsymbol{e}_1\right)^3 - \boldsymbol{x}^T\boldsymbol{e}_1$. Type C function is $f(\boldsymbol{x}) = \left(\boldsymbol{x}^T\boldsymbol{e}_1\right)^2 + \left(\boldsymbol{x}^T\boldsymbol{e}_1\right)^3$. Every curve is the average of 10 random simulations.

# B  Derivation of the learnable set $\mathcal{F}_{(3)}^{\ell_2}$

For the derivation of the learnable set, we assume that the noise $\boldsymbol{\epsilon}$ is zero in Eq. (3). We first rewrite Eq. (3) as the sum of terms contributed by each sample. Recall that $\mathbf{H}^T = [\mathbf{H}_1^T \;\cdots\; \mathbf{H}_n^T] \in \mathbb{R}^{(p_1 p_2)\times n}$ where $\mathbf{H}_i \in \mathbb{R}^{1\times(p_1 p_2)}$, $i = 1, 2, \cdots, n$. Thus, we have $\boldsymbol{h}_{\mathbf{V},\mathbf{W}_0,\boldsymbol{x}}^{\text{Three}}\mathbf{H}^T = \sum_{i=1}^{n}\left(\boldsymbol{h}_{\mathbf{V},\mathbf{W}_0,\boldsymbol{x}}^{\text{Three}}\mathbf{H}_i^T\right)\boldsymbol{e}_i^T$ where $\boldsymbol{e}_i \in \mathbb{R}^n$ denotes the $i$-th standard basis (i.e., the $i$-th element is 1 while all other elements are 0). Thus, we have

$$\hat{f}^{\ell_2}(\boldsymbol{x}) = \boldsymbol{h}_{\mathbf{V},\mathbf{W}_0,\boldsymbol{x}}^{\text{Three}}\mathbf{H}^T(\mathbf{H}\mathbf{H}^T)^{-1}\mathbf{F}(\mathbf{X}) = \sum_{i=1}^{n}\left(\frac{1}{p_2}\boldsymbol{h}_{\mathbf{V},\mathbf{W}_0,\boldsymbol{x}}^{\text{Three}}\mathbf{H}_i^T\right)p_2\boldsymbol{e}_i^T(\mathbf{H}\mathbf{H}^T)^{-1}\mathbf{F}(\mathbf{X}). \quad (10)$$

For any $\boldsymbol{a}, \boldsymbol{b} \in \mathbb{R}^{p_1}$, we define a set

$$\mathcal{C}_{\boldsymbol{a},\boldsymbol{b}}^{\mathbf{W}_0} := \left\{k \in \{1, 2, \cdots, p_2\} \;\middle|\; \boldsymbol{a}^T\mathbf{W}_0[k] > 0,\; \boldsymbol{b}^T\mathbf{W}_0[k] > 0\right\}, \quad (11)$$

whose cardinality is given by

$$\left|\mathcal{C}_{\boldsymbol{a},\boldsymbol{b}}^{\mathbf{W}_0}\right| = \sum_{k=1}^{p_2}\mathbf{1}_{\{\boldsymbol{a}^T\mathbf{W}_0[k]>0,\;\boldsymbol{b}^T\mathbf{W}_0[k]>0\}}.$$

Intuitively, $\mathcal{C}_{\boldsymbol{a},\boldsymbol{b}}^{\mathbf{W}_0}$ denotes the indices of the ReLU in the second hidden-layer that are activated both when the output of the first layer is $\boldsymbol{a}$ and when the output of the first layer is $\boldsymbol{b}$. Then, by Eq. (2),

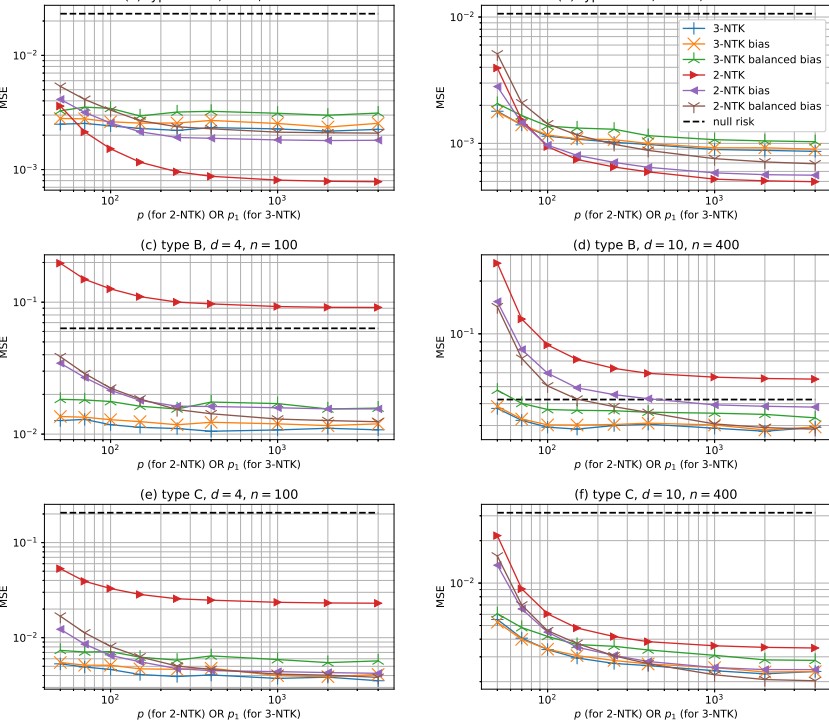

Figure 6: Curves of MSE with respect to $p$ (for 2-layer NTK) or $p_1$ (for 3-layer NTK where $p_2 = 100$). Other settings such as types of ground-truth functions and $\epsilon = 0$ are the same as those in Fig. 5. Every curve is the average of 10 random simulations.

we have

$$\frac{1}{p_2} \boldsymbol{h}_{\mathbf{V},\mathbf{W}_0,\boldsymbol{x}}^{\text{Three}} \mathbf{H}_i^T = (\boldsymbol{h}_{\mathbf{V},\boldsymbol{x}}^{\text{RF}})^T \boldsymbol{h}_{\mathbf{V},\mathbf{X}_i}^{\text{RF}} \frac{\left| \mathcal{C}_{\boldsymbol{h}_{\mathbf{V},\boldsymbol{x}}^{\text{RF}}, \boldsymbol{h}_{\mathbf{V},\mathbf{X}_i}^{\text{RF}}}^{\mathbf{W}_0} \right|}{p_2}. \tag{12}$$

By Assumption 1, which gives the distribution of $\mathbf{W}_0$, we can calculate the limiting value of Eq. (12) when there are an infinite number of neurons in the second hidden-layer. Specifically, since

$$\frac{\left| \mathcal{C}_{\boldsymbol{h}_{\mathbf{V},\boldsymbol{x}}^{\text{RF}}, \boldsymbol{h}_{\mathbf{V},\mathbf{X}_i}^{\text{RF}}}^{\mathbf{W}_0} \right|}{p_2} \xrightarrow{\text{P}} \frac{\pi - \arccos\left( \frac{(\boldsymbol{h}_{\mathbf{V},\boldsymbol{x}}^{\text{RF}})^T \boldsymbol{h}_{\mathbf{V},\mathbf{X}_i}^{\text{RF}}}{\left\| \boldsymbol{h}_{\mathbf{V},\boldsymbol{x}}^{\text{RF}} \right\|_2 \cdot \left\| \boldsymbol{h}_{\mathbf{V},\mathbf{X}_i}^{\text{RF}} \right\|_2} \right)}{2\pi}, \text{ as } p_2 \to \infty, \tag{13}$$

where $\xrightarrow{\text{P}}$ denotes convergence in probability, we have

$$\text{Eq. (12)} \xrightarrow{\text{P}} \left\| \boldsymbol{h}_{\mathbf{V},\boldsymbol{x}}^{\text{RF}} \right\|_2 \cdot \left\| \boldsymbol{h}_{\mathbf{V},\mathbf{X}_i}^{\text{RF}} \right\|_2 \cdot K^{\text{Two}}\left( \frac{(\boldsymbol{h}_{\mathbf{V},\boldsymbol{x}}^{\text{RF}})^T \boldsymbol{h}_{\mathbf{V},\mathbf{X}_i}^{\text{RF}}}{\left\| \boldsymbol{h}_{\mathbf{V},\boldsymbol{x}}^{\text{RF}} \right\|_2 \cdot \left\| \boldsymbol{h}_{\mathbf{V},\mathbf{X}_i}^{\text{RF}} \right\|_2} \right), \text{ as } p_2 \to \infty.$$

Note that $K^{\text{Two}}$ is known to be the kernel of 2-layer NTK (Ju et al., 2021). It is natural that $K^{\text{Two}}$ appears here, since we can regard the output of the first hidden-layer as the input of a 2-layer network consisting of the top- and middle-layer of the 3-layer network.

To further simplify the above expression, it remains to calculate $(\boldsymbol{h}_{\mathbf{V},\boldsymbol{x}}^{\text{RF}})^T \boldsymbol{h}_{\mathbf{V},\mathbf{X}_i}^{\text{RF}}$. Similar to the derivation above, when the first hidden-layer has an infinite number of neurons, we have

$$(\boldsymbol{h}_{\mathbf{V},\boldsymbol{x}}^{\text{RF}})^T \boldsymbol{h}_{\mathbf{V},\mathbf{X}_i}^{\text{RF}} \xrightarrow{\text{P}} K^{\text{RF}}(\boldsymbol{x}^T \mathbf{X}_i), \text{ as } p_1 \to \infty. \tag{14}$$

(Eq. (13) and Eq. (14) can be derived from integration over a hyper-sphere, which is shown in Lemma 20 and Lemma 21 in Appendix F.6, respectively.) Note that $K^{\text{RF}}$ is also the kernel of the

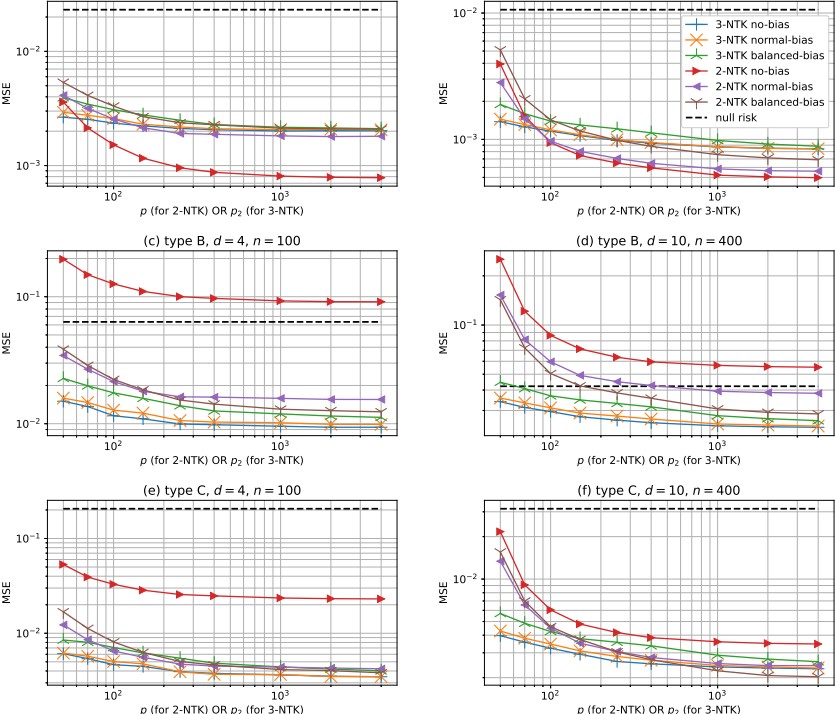

Figure 7: Curves of MSE with respect to $p$ (for 2-layer NTK) or $p_2$ (for 3-layer NTK where $p_1 = 100$). Other settings such as types of ground-truth functions and $\epsilon = 0$ are the same as those in Fig. 5. Every curve is the average of 10 random simulations.

random-feature model (Mei & Montanari, 2019). It is natural that $K^{\mathrm{RF}}$ appears here since Eq. (14) represents the situation that the bottom-layer has infinite width, which also appears in a random feature model. Notice that $K^{\mathrm{RF}}(\mathbf{X}_i^T \mathbf{X}_i) = K^{\mathrm{RF}}(\boldsymbol{x}^T \boldsymbol{x}) = K^{\mathrm{RF}}(1) = \frac{1}{2d}$. Thus, we have

$$\left\| \boldsymbol{h}_{\mathbf{V},\boldsymbol{x}}^{\mathrm{RF}} \right\|_2 \cdot \left\| \boldsymbol{h}_{\mathbf{V},\mathbf{X}_i}^{\mathrm{RF}} \right\|_2 \xrightarrow{\mathrm{P}} \frac{1}{2d}, \text{ as } p_1 \to \infty. \tag{15}$$

Plugging Eq. (13)(14)(15) into Eq. (12) and recalling Eq. (6), we thus have

$$\frac{1}{p_2} \boldsymbol{h}_{\mathbf{V},\mathbf{W}_0,\boldsymbol{x}}^{\mathrm{Three}} \mathbf{H}_i^T \xrightarrow{\mathrm{P}} K^{\mathrm{Three}}(\boldsymbol{x}^T \mathbf{X}_i), \text{ as } p_1, p_2 \to \infty.$$

If we let

$$g(\boldsymbol{z}) = \sum_{i=1}^n p_2 \boldsymbol{e}_i^T (\mathbf{H}\mathbf{H}^T)^{-1} \mathbf{F}(\mathbf{X}) \delta_{\mathbf{X}_i}(\boldsymbol{z}),$$

(where $\delta_{\boldsymbol{z}_0}(\boldsymbol{z})$ denotes a $\delta$-function, i.e., it has zero value for all $\boldsymbol{z} \in \mathcal{S}^{d-1} \setminus \{\boldsymbol{z}_0\}$, but its $L_1$-norm is $\|\delta_{\boldsymbol{z}_0}\|_1 := \int_{\mathcal{S}^{d-1}} \delta_{\boldsymbol{z}_0}(\boldsymbol{z}) d\mu(\boldsymbol{z}) = 1$), then as $p_1$ and $p_2 \to \infty$, Eq. (10) approaches $\int_{\mathcal{S}^{d-1}} K^{\mathrm{Three}}(\boldsymbol{x}^T \boldsymbol{z}) g(\boldsymbol{z}) d\mu(\boldsymbol{z})$, which is in the same form[8] as functions in $\mathcal{F}_{(3)}^{\ell_2}$.

---

[8] We acknowledge that the form here is still not exactly the same as $\mathcal{F}_{(3)}^{\ell_2}$ because the $\delta$-function does not satisfy the constrain of finite $\|g\|_\infty$. Nonetheless, $\mathcal{F}_{(3)}^{\ell_2}$ can be relaxed to allow finite $\|g\|_1$, which then includes the $\delta$-function. See footnote 4 on Page 6.

## C  A Precise Form of the Upper Bound in Theorem 1

We first introduce some extra notations and a condition about large $p_1$ that will be used later in our upper bound of the generalization error. Define

$$C(n, d, q) := \frac{\pi - 1}{4\pi} \min\left\{\frac{1}{2}, \left(\frac{(d-1)^2}{8d}\right)^{\frac{1}{d-1}} (qn)^{-\frac{4}{d-1}}\right\},\tag{16}$$

$$J(n, p_1, p_2, d, q) := \frac{1}{16\pi d}\sqrt{\frac{C(n, d, q)}{\log(4n)}} - \left(qn^2\sqrt{\frac{2d}{p_1}} + \frac{q^2 n^3 d}{p_1} + \frac{qn^2}{\sqrt{p_2}}\right),\tag{17}$$

$$Q(p_1, d) := 8d\sqrt{\frac{2(d+1)\log(p_1 + 1)}{p_1}}.\tag{18}$$

*Condition* 1. (Given $n$, $d$, and $q > 0$) $p_1$ and $p_2$ are sufficiently large such that $9d \cdot Q(p_1, d) \le 1$, $p_1 \ge \left(\frac{10dnq\sqrt{2d}}{C(n,d,q)}\right)^2$, and $J(n, p_1, p_2, d, q) > 0$.

**Theorem 3.** *Given a ground-truth function $f(x) = f_g(x) \in \mathcal{F}^{\ell_2}_{(3)}$, for any $q > 0$, under Condition 1, we must have*

$$\Pr_{\mathbf{V}, \mathbf{W}_0, \mathbf{X}}\left\{|\hat{f}^{\ell_2}(\boldsymbol{x}) - f(\boldsymbol{x})| \le \frac{q\|g\|_\infty}{\sqrt{n}} + \frac{q\|g\|_1}{\sqrt{p_2}} + \sqrt{\frac{Q(p_1, d)}{d}}\|g\|_1\right.$$

$$\left. + \frac{\sqrt{n}\|g\|_1\left(\frac{q}{\sqrt{p_2}} + \sqrt{\frac{Q(p_1,d)}{d}}\right) + \|\boldsymbol{\epsilon}\|_2}{\sqrt{J(n, p_1, p_2, d, q)}}\right\} \ge 1 - \frac{10}{q^2} - \frac{2d^2}{(p_1 + 1)e^{d+1}}.$$

A proof sketch can be found in Appendix E. To better illustrate the meaning of this upper bound, we provide a simplification in Theorem 1 when $p_1$ and $p_2$ are much larger than $n$. If we view $d$ as a constant, we have $C(n, d, q) = O(n^{-\frac{4}{d-1}})$. When $p_1$ and $p_2$ are much larger than $n$, we have $\frac{\sqrt{n}}{\sqrt{J(n,p_1,p_2,d,q)}} = O\left(n^{\frac{2}{d-1}+\frac{1}{2}} \cdot \sqrt{\log(n)}\right)$ and $\sqrt{\frac{Q(p_1,d)}{d}} = O\left(\sqrt[4]{\frac{\log p_1}{p_1}}\right)$. Therefore, when $d$ is fixed and when both $p_1$ and $p_2$ are much larger than $n$, Theorem 3 can be simplified to Eq. (8) (with high probability).

(In the above reduction to Eq. (8), we ignore the stand-alone term $\frac{q\|g\|_1}{\sqrt{p_2}}$ appeared in Theorem 3, since it is much smaller than the product of Term B and Term E. Similarly, we ignore the stand-alone term $\sqrt{\frac{Q(p_1,d)}{d}}\|g\|_1$, since it is much smaller than the product of Term C and Term E.)

## D  Noise Effect

Before we present the proof of Theorem 3 in Appendix E, we elaborate on how Theorem 3 reveals the impact of noise on the generalization error. Note that in Eq. (8), Term D denotes the average noise power in each training sample, and Term E denotes the extra multiplication factor with which the noise impacts the generalization error. As we see in Theorem 3 in Appendix C, the precise form of Term E is $\frac{\sqrt{n}}{\sqrt{J(n,p_1,p_2,d,q)}}$. Therefore, we will refer to the multiplication of $\|\boldsymbol{\epsilon}\|_2/\sqrt{n}$ with this factor as the "noise effect". Note that although the precise form $\frac{\sqrt{n}}{\sqrt{J(n,p_1,p_2,d,q)}}$ of this factor in Theorem 3 decreases with respect to both $p_1$ and $p_2$ by Eq. (17), when $p_1$ and $p_2$ are much larger than $n$, it can be simplified to Term E, which does not depend on $p_1$ and $p_2$.

In the following, we will analyze the relationship between the noise effect and various system parameters. First, we are interested in know how the numbers of neurons in two hidden-layers $p_1$ and $p_2$ impact the noise effect. Since Term E is an approximation when $p_1$ and $p_2$ are large and it does not contain $p_1$ or $p_2$, we conjecture that even when $p_1$ and $p_2$ are extremely large (e.g., $p_1, p_2 \to \infty$), the noise effect will neither grow dramatically nor go to zero. Further, when $p_1$ and $p_2$ are not so large, by Eq. (17), we know that the precise form of Term E in Theorem 3 decreases when $p_1$ and

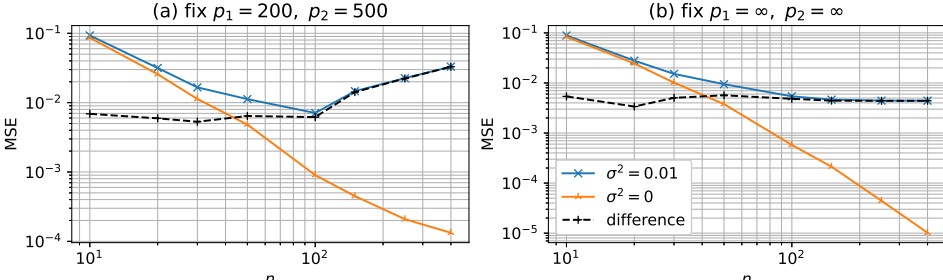

Figure 8: Noise effect on the test MSE of 3-layer NTK (no-bias) with respect to $n$. The noise follows *i.i.d.* Gaussian distribution with zero mean and variance $\sigma^2$. The ground-truth function is $f(\boldsymbol{x}) = \left(\boldsymbol{x}^T \boldsymbol{e}_1\right)^2 + \left(\boldsymbol{x}^T \boldsymbol{e}_1\right)^3$ where $d = 3$. Every curve is the average of 20 random simulations.

$p_2$ increase, which suggests that the noise will likely contribute more to the test error when the number of neurons is small. An intuitive explanation of such effect is that when $p_1$ and $p_2$ are small, the randomness of the initial weights brings some extra "pseudo-noise" to the model, and thus the generalization performance deteriorates.

Second, we are interested in how the noise effect changes with the number of training data $n$. We notice that Term E increases with $n$ at a speed faster than $\sqrt{n}$. However, since it is only an upper bound, the actual noise effect may grow much slower than $\sqrt{n}$. Therefore, precisely estimating the relationship between $n$ and the noise effect of NTK model would be an interesting future research direction.

We then use simulation to study the noise effect and compare them with the implications derived from our upper bound. In Fig. 8, we plot the curves of the test MSE with respect to $n$. The noise follows *i.i.d.* Gaussian $\mathcal{N}(0, \sigma^2)$. The blue curve denotes the situation where the noise level is $\sigma^2 = 0.01$. The orange curve denotes the noiseless situation. The noise effect (the value of the gap between the blue and the orange curves) is denoted by the dashed black curve. As we can see, when $n$ is large, the value of the black curve in Fig. 8(a) (fix $p_1 = 200, p_2 = 500$) is higher than that in Fig. 8(b) ($p_1, p_2 \to \infty$), which validates our conjecture that the noise contributes more to the test error when the number of neurons is small. Further, Fig. 8(b) shows that an infinite number of neurons does not make the noise effect diminish or explode for every $n$, which also confirms our previous analysis on the relationship between the number of neurons and the noise effect. We also notice that the black curve in Fig. 8(b) (where $p_1, p_2 \to \infty$) does not increase significantly with $n$, which suggests that our estimate on how fast Term E increases with $n$ could be further improved.

## E    Proof of Theorem 3

Recall that Theorem 3 is the precise form of Theorem 1, and is stated in Appendix C. To prove Theorem 3, we follow the line of analysis in Ju et al. (2021). We first study the class of the ground-truth functions that can be learned when weights $\mathbf{V}$ and $\mathbf{W}_0$ are fixed and there is no noise. We refer to them as *pseudo ground-truth* in the following definition, to differentiate them with the set $\mathcal{F}_{(3)}^{\ell_2}$ of learnable functions for random $\mathbf{V}$ and $\mathbf{W}_0$.

**Definition 2.** *Given* $\mathbf{V}$ *and* $\mathbf{W}_0$*, for any learnable ground-truth function* $f_g \in \mathcal{F}_{(3)}^{\ell_2}$ *with the corresponding function* $g(\cdot)$*, define the corresponding **pseudo ground-truth** as*

$$f_{\mathbf{V}, \mathbf{W}_0}^g(\boldsymbol{x}) := \int_{\mathcal{S}^{d-1}} \frac{(\boldsymbol{h}_{\mathbf{V}, \mathbf{W}_0, \boldsymbol{z}}^{Three})^T \boldsymbol{h}_{\mathbf{V}, \mathbf{W}_0, \boldsymbol{x}}^{Three}}{p_1 p_2} g(\boldsymbol{z}) d\mu(\boldsymbol{z})$$

$$= \int_{\mathcal{S}^{d-1}} (\boldsymbol{h}_{\mathbf{V}, \boldsymbol{z}}^{RF})^T \boldsymbol{h}_{\mathbf{V}, \boldsymbol{x}}^{RF} \frac{\left|\mathcal{C}_{\boldsymbol{h}_{\mathbf{V}, \boldsymbol{x}}^{RF}, \boldsymbol{h}_{\mathbf{V}, \boldsymbol{z}}^{RF}}^{\mathbf{W}_0}\right|}{p_1 p_2} g(\boldsymbol{z}) d\mu(\boldsymbol{z}). \tag{19}$$

*The last equality of Eq. (19) follows from Eq. (2) and Eq. (11). (The form of Eq. (19) can be derived using the similar process shown in Appendix B.)*

We prove Theorem 3 in several steps as follows.

**Step 1: use pseudo ground-truth as a "intermediary" .**

Recall the definition of pseudo ground-truth $f^g_{\mathbf{V},\mathbf{W}_0}(\cdot)$ in Eq. (19). We define

$$\mathbf{F}^g_{\mathbf{V},\mathbf{W}_0}(\mathbf{X}) := [f^g_{\mathbf{V},\mathbf{W}_0}(\mathbf{X}_1)\ f^g_{\mathbf{V},\mathbf{W}_0}(\mathbf{X}_2)\ \cdots\ f^g_{\mathbf{V},\mathbf{W}_0}(\mathbf{X}_n)]^T \in \mathbb{R}^n. \tag{20}$$

We then have

$$\begin{aligned}
\hat{f}^{\ell_2}(\boldsymbol{x}) =& \boldsymbol{h}^{\text{Three}}_{\mathbf{V},\mathbf{W}_0,\boldsymbol{x}} \mathbf{H}^T (\mathbf{H}\mathbf{H}^T)^{-1} \left( \mathbf{F}(\mathbf{X}) + \boldsymbol{\epsilon} \right) \text{ (by Eq. (3))} \\
=& \boldsymbol{h}^{\text{Three}}_{\mathbf{V},\mathbf{W}_0,\boldsymbol{x}} \mathbf{H}^T (\mathbf{H}\mathbf{H}^T)^{-1} \mathbf{F}^g_{\mathbf{V},\mathbf{W}_0}(\mathbf{X}) + \boldsymbol{h}^{\text{Three}}_{\mathbf{V},\mathbf{W}_0,\boldsymbol{x}} \mathbf{H}^T (\mathbf{H}\mathbf{H}^T)^{-1} \left( \mathbf{F}(\mathbf{X}) - \mathbf{F}^g_{\mathbf{V},\mathbf{W}_0}(\mathbf{X}) \right) \\
& + \boldsymbol{h}^{\text{Three}}_{\mathbf{V},\mathbf{W}_0,\boldsymbol{x}} \mathbf{H}^T (\mathbf{H}\mathbf{H}^T)^{-1} \boldsymbol{\epsilon}.
\end{aligned} \tag{21}$$

Thus, we have

$$\begin{aligned}
& |\hat{f}^{\ell_2}(\boldsymbol{x}) - f(\boldsymbol{x})| \\
=& |\hat{f}^{\ell_2}(\boldsymbol{x}) - f^g_{\mathbf{V},\mathbf{W}_0}(\boldsymbol{x}) + f^g_{\mathbf{V},\mathbf{W}_0}(\boldsymbol{x}) - f(\boldsymbol{x})| \\
=& |\boldsymbol{h}^{\text{Three}}_{\mathbf{V},\mathbf{W}_0,\boldsymbol{x}} \mathbf{H}^T (\mathbf{H}\mathbf{H}^T)^{-1} \mathbf{F}^g_{\mathbf{V},\mathbf{W}_0}(\mathbf{X}) - f^g_{\mathbf{V},\mathbf{W}_0}(\boldsymbol{x}) \\
& + \boldsymbol{h}^{\text{Three}}_{\mathbf{V},\mathbf{W}_0,\boldsymbol{x}} \mathbf{H}^T (\mathbf{H}\mathbf{H}^T)^{-1} \left( \mathbf{F}(\mathbf{X}) - \mathbf{F}^g_{\mathbf{V},\mathbf{W}_0}(\mathbf{X}) \right) \\
& + f^g_{\mathbf{V},\mathbf{W}_0}(\boldsymbol{x}) - f(\boldsymbol{x}) + \boldsymbol{h}^{\text{Three}}_{\mathbf{V},\mathbf{W}_0,\boldsymbol{x}} \mathbf{H}^T (\mathbf{H}\mathbf{H}^T)^{-1} \boldsymbol{\epsilon}| \text{ (by Eq. (21))} \\
\leq& \underbrace{|\boldsymbol{h}^{\text{Three}}_{\mathbf{V},\mathbf{W}_0,\boldsymbol{x}} \mathbf{H}^T (\mathbf{H}\mathbf{H}^T)^{-1} \mathbf{F}^g_{\mathbf{V},\mathbf{W}_0}(\mathbf{X}) - f^g_{\mathbf{V},\mathbf{W}_0}(\boldsymbol{x})|}_{\text{term A}} \\
& + \underbrace{|\boldsymbol{h}^{\text{Three}}_{\mathbf{V},\mathbf{W}_0,\boldsymbol{x}} \mathbf{H}^T (\mathbf{H}\mathbf{H}^T)^{-1} \left( \mathbf{F}(\mathbf{X}) - \mathbf{F}^g_{\mathbf{V},\mathbf{W}_0}(\mathbf{X}) \right)|}_{\text{term B}} \\
& + \underbrace{|f^g_{\mathbf{V},\mathbf{W}_0}(\boldsymbol{x}) - f(\boldsymbol{x})|}_{\text{term C}} + \underbrace{|\boldsymbol{h}^{\text{Three}}_{\mathbf{V},\mathbf{W}_0,\boldsymbol{x}} \mathbf{H}^T (\mathbf{H}\mathbf{H}^T)^{-1} \boldsymbol{\epsilon}|}_{\text{term D}}.
\end{aligned} \tag{22}$$

In Eq. (22), term A denotes the test error when using the pseudo ground-truth function, term B denotes the effect of replacing the original ground-truth function by the pseudo ground-truth function in the training samples, term C denotes the difference between the original ground-truth function and the pseudo ground-truth function on the test input, term D denotes the noise effect. Next, we bound these terms one by one.

**Step 2: estimate term A.**

The following proposition gives an upper bound of the test error when the data model is based on the pseudo ground-truth and the NTK model uses exactly the same $\mathbf{V}$ and $\mathbf{W}_0$.

**Proposition 4.** *Assume fixed $\mathbf{V}$ and $\mathbf{W}_0$, (thus $p_1$, $p_2$ and $d$ are also fixed), and there is no noise. If the ground-truth function is $f = f^g_{\mathbf{V},\mathbf{W}_0}$ in Definition 2 and $\|g\|_\infty < \infty$, then for any $\boldsymbol{x} \in \mathcal{S}^{d-1}$ and $q > 0$, we must have*

$$\Pr_{\mathbf{X}} \left\{ |f^g_{\mathbf{V},\mathbf{W}_0}(\boldsymbol{x}) - \hat{f}^{\ell_2}(\boldsymbol{x})| \geq \frac{q\|g\|_\infty}{\sqrt{n}} \right\} \leq \frac{1}{q^2}.$$

The proof of Proposition 4 is in Appendix G. Proposition 4 captures how the test error decreases with the number of training samples $n$, if the data model is based on a pseudo ground-truth function with the same $\mathbf{V}$ and $\mathbf{W}_0$ as the NTK. The result shown in Proposition 4 contributes to Term A in Eq. (8). Here we sketch the proof of Proposition 4. By Eq. (19), we can find a vector $\Delta\mathbf{W}^* \in \mathbb{R}^{(p_1 p_2) \times 1}$ and rewrite $f^g_{\mathbf{V},\mathbf{W}_0}$ as $f^g_{\mathbf{V},\mathbf{W}_0} = \boldsymbol{h}^{\text{Three}}_{\mathbf{V},\mathbf{W}_0,\boldsymbol{x}} \Delta\mathbf{W}^*$. The specific form of $\Delta\mathbf{W}^*$ can be found in Eq. (34) in Appendix G. Then, by Eq. (3), we can see that the learned model is $\hat{f}^{\ell_2}(\boldsymbol{x}) = \boldsymbol{h}^{\text{Three}}_{\mathbf{V},\mathbf{W}_0,\boldsymbol{x}} \mathbf{P} \Delta\mathbf{W}^*$ where $\mathbf{P} := \mathbf{H}^T (\mathbf{H}\mathbf{H}^T)^{-1}\mathbf{H}$ (an orthogonal projection to the row-space of $\mathbf{H}$). Thus, we have $|f^g_{\mathbf{V},\mathbf{W}_0}(\boldsymbol{x}) - \hat{f}^{\ell_2}(\boldsymbol{x})| = |\boldsymbol{h}^{\text{Three}}_{\mathbf{V},\mathbf{W}_0,\boldsymbol{x}} (\mathbf{P} - \mathbf{I}) \Delta\mathbf{W}^*| \leq \|\boldsymbol{h}^{\text{Three}}_{\mathbf{V},\mathbf{W}_0,\boldsymbol{x}}\|_2 \cdot \|(\mathbf{P} - \mathbf{I})\Delta\mathbf{W}^*\|_2$. Further, it is easy to show that $\|\boldsymbol{h}^{\text{Three}}_{\mathbf{V},\mathbf{W}_0,\boldsymbol{x}}\|_2 \leq \sqrt{p_1 p_2}$. It then remains to estimate $\|(\mathbf{P} - \mathbf{I})\Delta\mathbf{W}^*\|_2$, which is

upper bounded by $\min_{\boldsymbol{a} \in \mathbb{R}^n} \left\| \Delta \mathbf{W}^* - \mathbf{H}^T \boldsymbol{a} \right\|_2$ (because $\mathbf{P}$ is an orthogonal projection). The rest of proof focuses on how to choose a vector $\boldsymbol{a}$ to make $\left\| \Delta \mathbf{W}^* - \mathbf{H}^T \boldsymbol{a} \right\|_2$ as small as possible. Notice that although the similar method of choosing a suitable $\boldsymbol{a}$ is also used for 2-layer NTK (Ju et al., 2021), the process of estimating $\left\| \Delta \mathbf{W}^* - \mathbf{H}^T \boldsymbol{a} \right\|_2$ is much more complicated than that in Ju et al. (2021), since the feature vector $\boldsymbol{h}^{\text{Three}}_{\mathbf{V}, \mathbf{W}_0, \boldsymbol{x}}$ of 3-layer NTK involves non-linear activation for two hidden-layers (instead of one in 2-layer NTK).

With Proposition 4, now we are ready to estimate term A of Eq. (22). We have

$$
\Pr_{\mathbf{X}, \mathbf{V}, \mathbf{W}_0} \left\{ \text{term A} \geq \frac{q \|g\|_\infty}{\sqrt{n}} \right\}
$$
$$
= \int_{\mathbb{R}^{dp_1}} \int_{\mathbb{R}^{p_1 p_2}} \Pr_{\mathbf{X}} \left\{ \text{term A} \geq \frac{q \|g\|_\infty}{\sqrt{n}} \right\} d\Lambda_w(\mathbf{W}_0) d\Lambda_v(\mathbf{V})
$$
(where $\Lambda_w(\cdot)$ and $\Lambda_v(\cdot)$ are probability distribution of $\mathbf{W}_0$ and $\mathbf{V}$, respectively)
$$
\leq \frac{1}{q^2} \text{ (by Proposition 4)}.
$$

**Step 3: estimate term C.**

Intuitively, when $p_1$ and $p_2$ become larger, the randomness brought by $\mathbf{V}$ and $\mathbf{W}_0$ in the pseudo ground-truth $f^g_{\mathbf{V}, \mathbf{W}_0}$ will be "averaged out", and thus $f^g_{\mathbf{V}, \mathbf{W}_0}(\boldsymbol{x})$ will approach $f(\boldsymbol{x})$ (i.e., term C will approaches zero). The following proposition makes this statement rigorous.

**Proposition 5.** *For any $\boldsymbol{x} \in \mathcal{S}^{d-1}$ and $q > 0$, we must have*

$$
\Pr_{\mathbf{V}, \mathbf{W}_0} \left\{ \left| f^g_{\mathbf{V}, \mathbf{W}_0}(\boldsymbol{x}) - f(\boldsymbol{x}) \right| \geq \frac{q \|g\|_1}{\sqrt{p_2}} + \sqrt{\frac{Q(p_1, d)}{d}} \|g\|_1 \right\} \leq \frac{d^2}{(p_1 + 1)e^{d+1}} + \frac{1}{q^2}.
$$

The proof of Proposition 5 is in Appendix I.1. Note that as $p_1$ and $p_2$ increase, both $\frac{q \|g\|_1}{\sqrt{p_2}}$ and $\sqrt{\frac{Q(p_1, d)}{d}} \|g\|_1$ decrease, which implies that the pseudo ground-truth $f^g_{\mathbf{V}, \mathbf{W}_0}(\boldsymbol{x})$ approaches $f(\boldsymbol{x})$ with high probability. The above result thus directly bounds term C.

**Step 4: estimate terms B and D.**

We note that both terms B and D are of a similar form. Specifically, we can view the difference between $\mathbf{F}(\mathbf{X})$ and $\mathbf{F}^g_{\mathbf{V}, \mathbf{W}_0}(\mathbf{X})$ as a special type of "noise" due to random $\mathbf{V}$ and $\mathbf{W}_0$ (which will approaches zero when $p_1, p_2 \to \infty$). Then, both terms B and D are the multiplication of $\boldsymbol{h}^{\text{Three}}_{\mathbf{V}, \mathbf{W}_0, \boldsymbol{x}} \mathbf{H}^T (\mathbf{H}\mathbf{H}^T)^{-1}$ with the noise (either real noise or the special "noise" above). Further, we can show that the magnitude of $\boldsymbol{h}^{\text{Three}}_{\mathbf{V}, \mathbf{W}_0, \boldsymbol{x}} \mathbf{H}^T (\mathbf{H}\mathbf{H}^T)^{-1}$ can be upper bounded by a quantity inversely proportional to the minimum eigenvalue of $\mathbf{H}\mathbf{H}^T$. Thus, a key step of the proof is to estimate the minimum eigenvalue of $\mathbf{H}\mathbf{H}^T$. We prove the following proposition about $\min \text{eig}(\mathbf{H}\mathbf{H}^T)$ in Appendix H.

**Proposition 6.** *Recall the definition of $J(\cdot)$ in Eq. (17). For any $q > 0$, when Condition 1 is satisfied, we must have*

$$
\Pr_{\mathbf{X}, \mathbf{V}, \mathbf{W}_0} \left\{ \frac{1}{p_1 p_2} \min \text{eig}(\mathbf{H}\mathbf{H}^T) \leq J(n, p_1, p_2, d, q) \right\} \leq \frac{7}{q^2}.
$$

Using Proposition 6, we can then bound terms B and D by the following Proposition 7.

**Proposition 7.** *For any $q > 0$, when Condition 1 is satisfied, we must have*

$$
\Pr_{\mathbf{X}, \mathbf{V}, \mathbf{W}_0} \left\{ \text{term D} + \text{term B of Eq. (22)} \geq \frac{\sqrt{n} \|g\|_1 \left( \frac{q}{\sqrt{p_2}} + \sqrt{\frac{Q(p_1, d)}{d}} \right) + \|\boldsymbol{\epsilon}\|_2}{\sqrt{J(n, p_1, p_2, d, q)}} \right\}
$$
$$
\leq \frac{d^2}{(p_1 + 1)e^{d+1}} + \frac{8}{q^2}.
$$

Note that $\sqrt{n}\|g\|_1 \left( \frac{q}{\sqrt{p_2}} + \sqrt{\frac{Q(p_1,d)}{d}} \right)$ and $\|\epsilon\|_2$ correspond to the magnitude of the special "noise" $(\mathbf{F}(\mathbf{X}) - \mathbf{F}^g_{\mathbf{V},\mathbf{W}_0}(\mathbf{X}))$ (which can be bounded just like Proposition 5) and the real noise $\epsilon$, respectively. The proof of Proposition 7 is in Appendix I.2.

Plugging the results in Steps 2, 3, and 4 into Eq. (22), the result of Theorem 3 thus follows. Appendices G to I will prove the above propositions, after we present some supporting lemmas in Appendix F.

## F Useful Notations and Lemmas

We first collect some useful notations and lemmas, which will be used in the proofs of propositions appeared in Appendix E, as well as the analysis of learnable functions. Let $I.(\cdot, \cdot)$ denote the regularized incomplete beta function Dutka (1981). Let $B(\cdot, \cdot)$ denote the beta function Chaudhry et al. (1997). Specifically,

$$B(x, y) := \int_0^1 t^{x-1}(1-t)^{y-1}dt, \tag{23}$$

$$I_x(a, b) := \frac{\int_0^x t^{a-1}(1-t)^{b-1}dt}{B(a, b)}. \tag{24}$$

Define a cap on a unit hyper-sphere $\mathcal{S}^{d-1}$ as the intersection of $\mathcal{S}^{d-1}$ with an open ball in $\mathbb{R}^d$ centered at $\boldsymbol{v}_*$ with radius $r$, i.e.,

$$\mathcal{B}^r_{\boldsymbol{v}_*} := \left\{ \boldsymbol{v} \in \mathcal{S}^{d-1} \mid \|\boldsymbol{v} - \boldsymbol{v}_*\|_2 < r \right\}. \tag{25}$$

*Remark* 2. For ease of exposition, we will sometimes neglect the subscript $\boldsymbol{v}_*$ of $\mathcal{B}^r_{\boldsymbol{v}_*}$ and use $\mathcal{B}^r$ instead, when the quantity that we are estimating only depends on $r$ but not $\boldsymbol{v}_*$. For example, where we are interested in the area of $\mathcal{B}^r_{\boldsymbol{v}_*}$, it only depends on $r$ but not $\boldsymbol{v}_*$. Thus, we write $\lambda_{d-1}(\mathcal{B}^r)$ instead.

### F.1 Quantities related to the area of a cap on a hyper-sphere

The lemmas of this subsection support for the proof of Proposition 6. The following lemma is introduced by Li (2011), which gives the area of a cap on a hyper-sphere with respect to the colatitude angle.

**Lemma 8.** *Let $\phi \in [0, \frac{\pi}{2}]$ denote the colatitude angle of the smaller cap on the unit hyper-sphere $\mathcal{S}^{a-1}$, then the area (in the measure of $\lambda_{a-1}$) of this hyper-spherical cap is*

$$\frac{1}{2}\lambda_{a-1}(\mathcal{S}^{a-1})I_{\sin^2 \phi}\left( \frac{a-1}{2}, \frac{1}{2} \right),$$

*or equivalently[9],*

$$\lambda_{a-1}(\mathcal{B}^r) = \frac{1}{2}\lambda_{a-1}(\mathcal{S}^{a-1})I_{r^2(1-\frac{r^2}{4})}\left( \frac{a-1}{2}, \frac{1}{2} \right).$$

*where $r \le \sqrt{2}$.*

The following lemma is shown by Lemma 35 of Ju et al. (2021).

**Lemma 9.** *For any $x \in [0, 1]$, we must have*

$$I_x\left( \frac{a-1}{2}, \frac{1}{2} \right) \in \left[ \frac{2x^{\frac{a-1}{2}}}{B(\frac{a-1}{2}, \frac{1}{2}) \cdot (a-1)}, \frac{2x^{\frac{a-1}{2}}}{B(\frac{a-1}{2}, \frac{1}{2}) \cdot (a-1)\sqrt{1-x}} \right].$$

The following lemma is shown by Lemma 32 of Ju et al. (2021).

---

[9]Proof of this equivalence can be found in Lemma 9 of Ju et al. (2021).

**Lemma 10.** *For any integer $a \geq 2$,*

$$B\left(\frac{a-1}{2}, \frac{1}{2}\right) \in \left[\frac{1}{\sqrt{a}}, \pi\right].$$

*Further, if $a \geq 5$, we have*

$$B\left(\frac{a-1}{2}, \frac{1}{2}\right) \in \left[\frac{1}{\sqrt{a}}, \frac{4}{\sqrt{a-3}}\right].$$

### F.2 Estimation of certain norms

In our proofs, we will often need to estimate the norms of the NTK feature vectors. We list some useful lemmas below.

**Lemma 11.** *For any $\boldsymbol{x} \in \mathcal{S}^{d-1}$, we have*

$$\|\boldsymbol{h}_{\mathbf{V},\boldsymbol{x}}^{RF}\|_2 \leq \sqrt{p_1}, \quad \|\boldsymbol{h}_{\mathbf{V},\mathbf{W}_0,\boldsymbol{x}}^{Three}\|_2 \leq \sqrt{p_1 p_2}.$$

*Proof.* Notice that $\|\boldsymbol{x}\|_2 = 1$ and $\|\mathbf{V}[j]\|_2 = 1$ for all $j \in \{1, 2 \cdots, p_1\}$. By Eq. (1), we have

$$\|\boldsymbol{h}_{\mathbf{V},\boldsymbol{x}}^{\mathrm{RF}}\|_2 = \sqrt{\sum_{j=1}^{p_1}\left((\boldsymbol{x}^T\mathbf{V}[j])1_{\{\boldsymbol{x}^T\mathbf{V}[j]>0\}}\right)^2} \leq \sqrt{p_1}.$$

Thus, by Eq. (2), we have

$$\|\boldsymbol{h}_{\mathbf{V},\mathbf{W}_0,\boldsymbol{x}}^{\mathrm{Three}}\|_2 = \sqrt{\sum_{k=1}^{p_2}\|\boldsymbol{h}_{\mathbf{V},\boldsymbol{x}}^{\mathrm{RF}}1_{\{(\boldsymbol{h}_{\mathbf{V},\boldsymbol{x}}^{\mathrm{RF}})^T\mathbf{W}_0[k]>0\}}\|_2^2} \leq \sqrt{\sum_{k=1}^{p_2}\|\boldsymbol{h}_{\mathbf{V},\boldsymbol{x}}^{\mathrm{RF}}\|_2^2} \leq \sqrt{p_1 p_2}.$$

$\square$

The following lemma is from Lemma 12 of Ju et al. (2021), but we repeat here for the convenience of the readers.

**Lemma 12.** *If $\mathbf{C} = \mathbf{AB}$, then $\|\mathbf{C}\|_2 \leq \|\mathbf{A}\|_2 \cdot \|\mathbf{B}\|_2$. Here $\mathbf{A}$, $\mathbf{B}$, and $\mathbf{C}$ could be scalars, vectors, or matrices.*

*Proof.* This lemma directly follows the definition of matrix norm. $\square$

*Remark* 3. Note that the ($\ell_2$) matrix-norm (i.e., spectral norm) of a vector is exactly its $\ell_2$ vector-norm (i.e., Euclidean norm)[10]. Therefore, when applying Lemma 12, we do not need to worry about whether $\mathbf{A}$, $\mathbf{B}$, and $\mathbf{C}$ are matrices or vectors.

**Lemma 13.** *For any $\mathbf{A}, \mathbf{B} \in \mathbb{R}^{k \times k}$, we must have*

$$\|\mathbf{A} - \mathbf{B}\|_2 \leq k \cdot \max_{i,j}|\mathbf{A}_{i,j} - \mathbf{B}_{i,j}|.$$

*Consequently, if both $\mathbf{A}$ and $\mathbf{B}$ are positive semi-definite, then*

$$|\min\mathrm{eig}(\mathbf{A}) - \min\mathrm{eig}(\mathbf{B})| \leq k \cdot \max_{i,j}|\mathbf{A}_{i,j} - \mathbf{B}_{i,j}|.$$

---

[10]To see this, consider a (row or column) vector $\boldsymbol{a}$. The matrix norm of $\boldsymbol{a}$ is

$$\max_{|x|=1}\|\boldsymbol{a}x\|_2 \text{ (when } \boldsymbol{a} \text{ is a column vector)},$$

$$\text{or } \max_{\|\boldsymbol{x}\|_2=1}\|\boldsymbol{a}\boldsymbol{x}\|_2 \text{ (when } \boldsymbol{a} \text{ is a row vector)}.$$

In both cases, the value of the matrix-norm equals to $\sqrt{\sum a_i^2}$, which is exactly the $\ell_2$-norm (Euclidean norm) of $\boldsymbol{a}$.

*Proof.* Let $\mathbf{C} := \mathbf{A} - \mathbf{B}$. For any $\boldsymbol{a} \in \mathcal{S}^{k-1}$, we have

$$\|\mathbf{C}\boldsymbol{a}\|_2^2 = \sum_{i=1}^k \left( \sum_{j=1}^k \mathbf{C}_{i,j} a_j \right)^2$$

$$\leq k (\max_{i,j} \mathbf{C}_{i,j})^2 \left( \sum_{j=1}^k a_j \right)^2$$

$$\leq k^2 (\max_{i,j} \mathbf{C}_{i,j})^2 \sum_{j=1}^k a_j^2 \text{ (by Cauchy–Schwarz inequality)}$$

$$= k^2 (\max_{i,j} \mathbf{C}_{i,j})^2 \text{ (because } \|\boldsymbol{a}\|_2 = 1\text{).}$$

Because $\|\mathbf{C}\|_2 = \max_{\boldsymbol{a} \in \mathcal{S}^{k-1}} \|\mathbf{C}\boldsymbol{a}\|_2$, we have $\|\mathbf{A} - \mathbf{B}\|_2 \leq k \cdot \max_{i,j} |\mathbf{A}_{i,j} - \mathbf{B}_{i,j}|$.

Let $\boldsymbol{a}^* \in \arg\min_{\boldsymbol{a} \in \mathcal{S}^{k-1}} \|\mathbf{B}\boldsymbol{a}\|_2$. We have

$$\min\text{eig}(\mathbf{A}) = \min_{\boldsymbol{a} \in \mathcal{S}^{k-1}} \|\mathbf{A}\boldsymbol{a}\|_2$$

$$\leq \|\mathbf{A}\boldsymbol{a}^*\|_2$$

$$= \|(\mathbf{A} - \mathbf{B})\boldsymbol{a}^* + \mathbf{B}\boldsymbol{a}^*\|_2$$

$$\leq \|(\mathbf{A} - \mathbf{B})\boldsymbol{a}^*\|_2 + \|\mathbf{B}\boldsymbol{a}^*\|_2$$

$$\leq \|\mathbf{A} - \mathbf{B}\|_2 + \min\text{eig}(\mathbf{B}) \text{ (by the definition of } \boldsymbol{a}^*\text{).}$$

Thus, we have $\min\text{eig}(\mathbf{A}) - \min\text{eig}(\mathbf{B}) \leq \|\mathbf{A} - \mathbf{B}\|_2 \leq k \cdot \max_{i,j} |\mathbf{C}_{i,j}|$. Similarly, we have $\min\text{eig}(\mathbf{B}) - \min\text{eig}(\mathbf{A}) \leq k \cdot \max_{i,j} |\mathbf{C}_{i,j}|$. The result of this lemma thus follows. $\square$

### F.3 Estimates of certain tail probabilities

**Lemma 14** (Chebyshev's inequality on the sum of *i.i.d.* random variables/vectors). *Let $X_1, X_2, \cdots, X_k$ be* i.i.d. *random variables and $|X_i| \leq U$ for all $i = 1, 2, \cdots, k$. Then, for any $m > 0$,*

$$\Pr\left\{ \left| \left( \frac{1}{k} \sum_{i=1}^k X_i \right) - \mathsf{E} X_1 \right| \geq \frac{mU}{\sqrt{k}} \right\} \leq \frac{1}{m^2}.$$

*This inequality also holds when $X_1, X_2, \cdots, X_k$ are* i.i.d. *random vectors and $\|X_i\|_2 \leq U$ for all $i = 1, 2, \cdots, k$.*

*Proof.* Because $|X_1| \leq U$, we have

$$\mathsf{Var}[X_1] = \mathsf{E}[(X_1 - \mathsf{E}[X_1])^2] = \mathsf{E}[X_1^2] - (\mathsf{E}[X_1])^2 \leq \mathsf{E}[X_1^2] \leq U^2.$$

Because all $X_i$'s are *i.i.d.*, we have

$$\mathsf{Var}\left[ \frac{1}{k} \sum_{i=1}^k X_i \right] \leq \frac{U^2}{k}, \quad \mathsf{E}\left[ \frac{1}{k} \sum_{i=1}^k X_i \right] = \mathsf{E}[X_1].$$

The result of this lemma thus follows by applying Chebyshev's inequality on $\frac{1}{k} \sum_{i=1}^k X_i$. For the situation that $X_1, X_2, \cdots, X_k$ are vectors, the proof is the same by using the generalized Chebyshev's inequality for random vectors which we state in Lemma 15 as follows. $\square$

The following is the Chebyshev's inequality for random vectors that can be found in many textbooks of probability theory (see, e.g., pp. 446-451 of Laha & Rohatgi (1979)).

**Lemma 15** (Chebyshev's inequality for random vectors). *For a random vector $\boldsymbol{w} \in \mathbb{R}^a$ with probability distribution $\Lambda(\cdot)$, for any $\delta > 0$, we must have*

$$\Pr\{\|\boldsymbol{w} - \mathsf{E}(\boldsymbol{w})\|_2 \geq \delta\} \leq \frac{\mathsf{Var}(\boldsymbol{w})}{\delta^2},$$

*where*

$$\mathsf{Var}(\boldsymbol{w}) := \int_{\boldsymbol{v} \in \mathbb{R}^a} \|\boldsymbol{v} - \mathsf{E}(\boldsymbol{w})\|_2^2 \, d\Lambda(\boldsymbol{v}).$$

### F.4 Estimation about double factorial

Let $m$ be a positive integer. A double factorial can be defined by

$$(2m)!! := \prod_{i=1}^{m}(2i), \quad (2m-1)!! := \prod_{i=1}^{m}(2i-1). \tag{26}$$

They are useful in our study of learnable functions. The following lemma is proven by Chen & Qi (2005).

**Lemma 16** (Improved Wallis' Inequality). *For all natural numbers $k$, let $k!!$ denote a double factorial. Then*

$$\frac{1}{\sqrt{\pi \left(k + \frac{4}{\pi} - 1\right)}} \leq \frac{(2k-1)!!}{(2k)!!} < \frac{1}{\sqrt{\pi \left(k + \frac{1}{4}\right)}}.$$

*Further, the constants $\frac{4}{\pi} - 1$ and $\frac{1}{4}$ are the best possible.*

### F.5 Taylor expansion of kernels

The following Taylor expansions are related to the NTK kernel functions, which will also be used in our characterization of the learnable functions.

**Lemma 17.** *For any $\theta \in [0, \pi]$,*

$$\cos\theta \frac{(\pi - \theta)}{2\pi} = \frac{\cos\theta}{4} + \frac{1}{2\pi} \sum_{k=0}^{\infty} \frac{(2k)!}{(k!)^2} \frac{4}{2k+1} \left(\frac{\cos\theta}{2}\right)^{2k+2},$$

$$\frac{\sin\theta + (\pi - \theta)\cos\theta}{\pi} = \frac{1}{\pi}\left(1 + \frac{\pi}{2}\cos\theta + \sum_{k=0}^{\infty} \frac{2(2k)!}{(k+1)(2k+1)(k!)^2}\left(\frac{\cos\theta}{2}\right)^{2k+2}\right).$$

*Consequently, recalling Eq. (5) and Eq. (4), by letting $a = \cos\theta$, we have*

$$K^{Two}(a) = a\frac{\pi - \arccos a}{2\pi} = \frac{a}{4} + \frac{1}{2\pi}\sum_{k=0}^{\infty}\frac{(2k)!}{(k!)^2}\frac{4}{2k+1}\left(\frac{a}{2}\right)^{2k+2},$$

$$2d \cdot K^{RF}(a) = \frac{\sqrt{1-a^2} + a(\pi - \arccos a)}{\pi}$$

$$= \frac{1}{\pi}\left(1 + \frac{\pi}{2}a + \sum_{k=0}^{\infty}\frac{2(2k)!}{(k+1)(2k+1)(k!)^2}\left(\frac{a}{2}\right)^{2k+2}\right).$$

*Proof.* Using Taylor expansion on $\arccos x$, we have

$$\arccos(x) = \frac{\pi}{2} - \sum_{k=0}^{\infty}\frac{(2k)!}{2^{2k}(k!)^2}\frac{x^{2k+1}}{2k+1}.$$

We then have

$$\theta = \arccos(\cos\theta) = \frac{\pi}{2} - \sum_{k=0}^{\infty}\frac{(2k)!}{(k!)^2}\frac{2}{2k+1}\left(\frac{\cos\theta}{2}\right)^{2k+1}.$$

Thus, we have

$$\cos\theta\frac{(\pi - \theta)}{2\pi} = \cos\theta \cdot \left(\frac{1}{2} - \frac{1}{2\pi}\left(\frac{\pi}{2} - \sum_{k=0}^{\infty}\frac{(2k)!}{(k!)^2}\frac{2}{2k+1}\left(\frac{\cos\theta}{2}\right)^{2k+1}\right)\right)$$

$$= \frac{\cos\theta}{4} + \frac{1}{2\pi}\sum_{k=0}^{\infty}\frac{(2k)!}{(k!)^2}\frac{4}{2k+1}\left(\frac{\cos\theta}{2}\right)^{2k+2}. \tag{27}$$

Using Taylor expansion on $\sqrt{1+x}$, we have

$$\sqrt{1+x} = 1 - \sum_{k=0}^{\infty} \frac{2}{k+1}\binom{2k}{k}\left(-\frac{x}{4}\right)^{k+1},$$

Replacing $x$ by $-\cos^2\theta$, we thus have

$$\sin\theta = \sqrt{1-\cos^2\theta} = 1 - \sum_{k=0}^{\infty} \frac{2}{k+1}\binom{2k}{k}\left(\frac{\cos\theta}{2}\right)^{2k+2}.$$

Therefore, using Eq. (27) again, we have

$$\frac{\sin\theta + (\pi-\theta)\cos\theta}{\pi} = \frac{1}{\pi}\left(1 + \frac{\pi}{2}\cos\theta + \sum_{k=0}^{\infty}\left(\frac{2}{2k+1} - \frac{1}{k+1}\right)\frac{2(2k)!}{(k!)^2}\left(\frac{\cos\theta}{2}\right)^{2k+2}\right)$$

$$= \frac{1}{\pi}\left(1 + \frac{\pi}{2}\cos\theta + \sum_{k=0}^{\infty}\frac{2(2k)!}{(k+1)(2k+1)(k!)^2}\left(\frac{\cos\theta}{2}\right)^{2k+2}\right).$$

The result of this lemma thus follows. $\qquad\square$

### F.6 Calculation of certain integrals

**Lemma 18.** *For any integer $k \geq 2$, we have*

$$\int_0^\pi \sin^k\varphi\, d\varphi = \frac{k-1}{k}\int_0^\pi \sin^{k-2}\varphi\, d\varphi.$$

*Proof.* We have

$$\int_0^\pi \sin^k\varphi\, d\varphi = \int_0^\pi \sin\varphi \cdot \sin^{k-1}\varphi\, d\varphi$$

$$= -\cos\varphi \cdot \sin^{k-1}\varphi\Big|_0^\pi + (k-1)\int_0^\pi \cos^2\varphi \cdot \sin^{k-2}\varphi\, d\varphi$$

$$\text{(integration by parts)}$$

$$= (k-1)\int_0^\pi (1-\sin^2\varphi)\sin^{k-2}\varphi\, d\varphi$$

$$= (k-1)\int_0^\pi \sin^{k-2}\varphi\, d\varphi - (k-1)\int_0^\pi \sin^k\varphi\, d\varphi.$$

Moving the second term of the right hand side to the left hand side, we have

$$k\int_0^\pi \sin^k\varphi\, d\varphi = (k-1)\int_0^\pi \sin^{k-2}\varphi\, d\varphi.$$

The result of this lemma thus follows. $\qquad\square$

**Lemma 19.** *For any $\theta \in [0, \pi]$,*

$$\int_{-\frac{\pi}{2}+\theta}^{\frac{\pi}{2}} \cos(\alpha)\cos(\alpha-\theta)\, d\alpha = \frac{\sin\theta}{2} + \frac{(\pi-\theta)\cos\theta}{2}.$$

*Proof.* Notice that

$$\frac{\partial(\sin(2\alpha-\theta)+2\alpha\cos\theta)}{\partial\alpha} = 2\cos(2\alpha-\theta) + 2\cos\theta$$

$$= 2\cos(\alpha+(\alpha-\theta)) + 2\cos(\alpha-(\alpha-\theta))$$

$$= 4\cos(\alpha)\cos(\alpha-\theta).$$

Thus, we have

$$\int \cos(\alpha)\cos(\alpha - \theta)d\alpha = \frac{\sin(2\alpha - \theta) + 2\alpha\cos(\theta)}{4} + \text{constant}.$$

Notice that

$$\sin(2\alpha - \theta)\Big|_{\alpha=-\frac{\pi}{2}+\theta}^{\frac{\pi}{2}} = \sin(\pi - \theta) - \sin(\theta - \pi) = 2\sin\theta,$$

$$2\alpha\cos(\theta)\Big|_{\alpha=-\frac{\pi}{2}+\theta}^{\frac{\pi}{2}} = 2(\pi - \theta)\cos\theta.$$

The result of this lemma thus follows. $\qquad\square$

**Lemma 20.** *Recall that $\gamma(\cdot)$ denotes the probability density function of $\mathbf{W}_0[k]$ and is $\mathsf{unif}(\mathcal{S}^{p_1-1})$ by Assumption 1. For any $\boldsymbol{a}, \boldsymbol{b} \in \mathbb{R}^{p_1}$, we have*

$$\int_{\mathcal{S}^{p_1-1}} \boldsymbol{a}^T\boldsymbol{b} \cdot 1_{\{\boldsymbol{a}^T\boldsymbol{w}>0,\, \boldsymbol{b}^T\boldsymbol{w}>0\}} d\gamma(\boldsymbol{w}) = \boldsymbol{a}^T\boldsymbol{b}\frac{\pi - \arccos\left(\frac{\boldsymbol{a}^T\boldsymbol{b}}{\|\boldsymbol{a}\|_2\|\boldsymbol{b}\|_2}\right)}{2\pi}.$$

*(Although the right hand side is not defined when $\boldsymbol{a} = \boldsymbol{0}$ or $\boldsymbol{b} = \boldsymbol{0}$, we can artificially re-define the value of the right hand side as $0$ when $\boldsymbol{a} = \boldsymbol{0}$ or $\boldsymbol{b} = \boldsymbol{0}$, so the equation still holds.)*

*Proof.* The result holds trivially when $\boldsymbol{a} = 0$ or $\boldsymbol{b} = 0$. When $\boldsymbol{a}$ and $\boldsymbol{b}$ are both non-zero, it suffices to prove that

$$\int_{\mathcal{S}^{p_1-1}} 1_{\{\boldsymbol{a}^T\boldsymbol{w}>0,\, \boldsymbol{b}^T\boldsymbol{w}>0\}} d\gamma(\boldsymbol{w}) = \frac{\pi - \arccos\left(\frac{\boldsymbol{a}^T\boldsymbol{b}}{\|\boldsymbol{a}\|_2\|\boldsymbol{b}\|_2}\right)}{2\pi},$$

which has been proven by Lemma 17 of Ju et al. (2021) (where its geometric explanation is given as well). $\qquad\square$

**Lemma 21.** *For any $\boldsymbol{x}, \boldsymbol{z} \in \mathcal{S}^{d-1}$, we have*

$$\int_{\mathcal{S}^{d-1}} (\boldsymbol{x}^T\boldsymbol{v})(\boldsymbol{z}^T\boldsymbol{v})1_{\{\boldsymbol{z}^T\boldsymbol{v}>0,\, \boldsymbol{x}^T\boldsymbol{v}>0\}} d\lambda(\boldsymbol{v}) = \frac{\sin\theta + (\pi - \theta)\cos\theta}{2d\pi}, \qquad (28)$$

*where $\theta$ denotes the angle between $\boldsymbol{x}$ and $\boldsymbol{z}$, i.e.,*

$$\theta = \arccos(\boldsymbol{x}^T\boldsymbol{z}) \in [0,\ \pi]. \qquad (29)$$

To help readers understand the correctness of Lemma 21, we first give a simple proof for the special case that $d = 2$, i.e., when vectors $\boldsymbol{x}$, $\boldsymbol{z}$, and $\boldsymbol{v}$ are all in the 2-D plane. Then we prove Lemma 21 for the general cases that $d = 2, 3, 4, \cdots$.

*Proof (of the case when $d = 2$):* Without loss of generality, we let

$$\boldsymbol{v} = \begin{bmatrix} \cos\alpha \\ \sin\alpha \end{bmatrix}, \quad \boldsymbol{z} = \begin{bmatrix} 1 \\ 0 \end{bmatrix}, \quad \text{and } \boldsymbol{x} = \begin{bmatrix} \cos\theta \\ \sin\theta \end{bmatrix}.$$

Thus, we have

$$\text{The left-hand-side of Eq. (28)} = \frac{1}{2\pi} \int_{\left(\theta-\frac{\pi}{2},\, \theta+\frac{\pi}{2}\right)\cap\left(-\frac{\pi}{2},\, \frac{\pi}{2}\right)} (\cos\alpha\cos\theta + \sin\alpha\sin\theta)\cos\alpha \, d\alpha$$

$$= \frac{1}{2\pi} \int_{\theta-\frac{\pi}{2}}^{\frac{\pi}{2}} \cos(\alpha - \theta)\cos\alpha \, d\alpha \quad (\text{since } \theta \in [0, \pi])$$

$$= \frac{\sin\theta + (\pi - \theta)\cos\theta}{4\pi} \quad (\text{by Lemma 19}).$$

*Proof (of the general case).* Due to symmetry, we know that the integral in the left-hand-side of Eq. (28) only depends on the angle between $\boldsymbol{x}$ and $\boldsymbol{z}$. Thus, without loss of generality, we let

$$\boldsymbol{x} = [\boldsymbol{x}_1\, \boldsymbol{x}_2\, \cdots\, \boldsymbol{x}_d] = [0\,0\,\cdots\,0\,1\,0]^T, \quad \boldsymbol{z} = [0\,0\,\cdots\,0\,\cos\theta\,\sin\theta]^T.$$

Thus, for any $\boldsymbol{v} = [\boldsymbol{v}_1\,\boldsymbol{v}_2\,\cdots\,\boldsymbol{v}_d]^T$, in order for $\boldsymbol{z}^T\boldsymbol{v} > 0$ and $\boldsymbol{x}^T\boldsymbol{v} > 0$ to hold, it only needs to satisfy

$$[\cos\theta\,\sin\theta]\begin{bmatrix}\boldsymbol{v}_{d-1}\\ \boldsymbol{v}_d\end{bmatrix} > 0, \quad [1\,0]\begin{bmatrix}\boldsymbol{v}_{d-1}\\ \boldsymbol{v}_d\end{bmatrix} > 0. \tag{30}$$

We use the spherical coordinate $\boldsymbol{\varphi_x} = [\varphi_1^{\boldsymbol{x}}\,\varphi_2^{\boldsymbol{x}}\,\cdots\,\varphi_{d-1}^{\boldsymbol{x}}]^T$ where $\varphi_1^{\boldsymbol{x}}, \cdots, \varphi_{d-2}^{\boldsymbol{x}} \in [0,\pi]$ and $\varphi_{d-1}^{\boldsymbol{x}} \in [0, 2\pi)$ with the convention that

$$\begin{aligned}
\boldsymbol{x}_1 &= \cos(\varphi_1^{\boldsymbol{x}}),\\
\boldsymbol{x}_2 &= \sin(\varphi_1^{\boldsymbol{x}})\cos(\varphi_2^{\boldsymbol{x}}),\\
\boldsymbol{x}_3 &= \sin(\varphi_1^{\boldsymbol{x}})\sin(\varphi_2^{\boldsymbol{x}})\cos(\varphi_3^{\boldsymbol{x}}),\\
&\vdots\\
\boldsymbol{x}_{d-1} &= \sin(\varphi_1^{\boldsymbol{x}})\sin(\varphi_2^{\boldsymbol{x}})\cdots\sin(\varphi_{d-2}^{\boldsymbol{x}})\cos(\varphi_{d-1}^{\boldsymbol{x}}),\\
\boldsymbol{x}_d &= \sin(\varphi_1^{\boldsymbol{x}})\sin(\varphi_2^{\boldsymbol{x}})\cdots\sin(\varphi_{d-2}^{\boldsymbol{x}})\sin(\varphi_{d-1}^{\boldsymbol{x}}).
\end{aligned}$$

Thus, we have $\boldsymbol{\varphi_x} = [\pi/2\,\pi/2\,\cdots\,\pi/2\,0]^T$. Similarly, the spherical coordinate for $\boldsymbol{z}$ is $\boldsymbol{\varphi_z} = [\pi/2\,\pi/2\,\cdots\pi/2\,\theta]^T$. Let the spherical coordinates for $\boldsymbol{v}$ be $\boldsymbol{\varphi_v} = [\varphi_1^{\boldsymbol{v}}\,\varphi_2^{\boldsymbol{v}}\,\cdots\,\varphi_{d-1}^{\boldsymbol{v}}]^T$. Thus, Eq. (30) is equivalent to

$$\boldsymbol{z}^T\boldsymbol{v} = \sin(\varphi_1^{\boldsymbol{v}})\sin(\varphi_2^{\boldsymbol{v}})\cdots\sin(\varphi_{d-2}^{\boldsymbol{v}})\left(\cos\theta\cos(\varphi_{d-1}^{\boldsymbol{v}}) + \sin\theta\sin(\varphi_{d-1}^{\boldsymbol{v}})\right) > 0, \tag{31}$$

$$\boldsymbol{x}^T\boldsymbol{v} = \sin(\varphi_1^{\boldsymbol{v}})\sin(\varphi_2^{\boldsymbol{v}})\cdots\sin(\varphi_{d-2}^{\boldsymbol{v}})\cos(\varphi_{d-1}^{\boldsymbol{v}}) > 0. \tag{32}$$

Because $\varphi_1^{\boldsymbol{v}}, \cdots, \varphi_{d-2}^{\boldsymbol{v}} \in [0,\pi]$ (by the convention of spherical coordinates), we have

$$\sin(\varphi_1^{\boldsymbol{v}})\sin(\varphi_2^{\boldsymbol{v}})\cdots\sin(\varphi_{d-2}^{\boldsymbol{v}}) \geq 0.$$

Thus, for Eq. (31) and Eq. (32) to hold, we must have

$$\cos(\theta - \varphi_{d-1}^{\boldsymbol{v}}) > 0, \quad \cos(\varphi_{d-1}^{\boldsymbol{v}}) > 0,$$

i.e., $\varphi_{d-1}^{\boldsymbol{v}} \in (-\pi/2,\,\pi/2) \cap (\theta - \pi/2,\,\theta + \pi/2) \pmod{2\pi}$. By Eq. (29), we thus have

$$\varphi_{d-1} \in \left(-\frac{\pi}{2} + \theta,\,\frac{\pi}{2}\right) \pmod{2\pi}.$$

Let

$$A(\theta, \varphi_{d-1}^{\boldsymbol{v}}) := \left(\cos\theta\cos(\varphi_{d-1}^{\boldsymbol{v}}) + \sin\theta\sin(\varphi_{d-1}^{\boldsymbol{v}})\right)\cos(\varphi_{d-1}^{\boldsymbol{v}}) = \cos(\varphi_{d-1}^{\boldsymbol{v}} - \theta)\cos\varphi_{d-1}^{\boldsymbol{v}}.$$

By Eq. (31) and Eq. (32), we have

$$(\boldsymbol{x}^T\boldsymbol{v})(\boldsymbol{z}^T\boldsymbol{v})\mathbf{1}_{\{\boldsymbol{z}^T\boldsymbol{v}>0,\,\boldsymbol{x}^T\boldsymbol{v}>0\}} = \sin^2(\varphi_1^{\boldsymbol{v}})\sin^2(\varphi_2^{\boldsymbol{v}})\cdots\sin^2(\varphi_{d-2}^{\boldsymbol{v}})A(\theta, \varphi_{d-1}^{\boldsymbol{v}}).$$

Integrating using such spherical coordinates, we have

$$\begin{aligned}
&\int_{\mathcal{S}^{d-1}} (\boldsymbol{x}^T\boldsymbol{v})(\boldsymbol{z}^T\boldsymbol{v})\mathbf{1}_{\{\boldsymbol{z}^T\boldsymbol{v}>0,\,\boldsymbol{x}^T\boldsymbol{v}>0\}}d\lambda(\boldsymbol{v})\\
&= \frac{\int_{-\frac{\pi}{2}+\theta}^{\frac{\pi}{2}} A(\theta, \varphi_{d-1}^{\boldsymbol{v}}) \int_0^\pi \cdots \int_0^\pi \sin^d(\varphi_1)\sin^{d-1}(\varphi_2)\cdots\sin^3(\varphi_{d-2})\,d\varphi_1\,d\varphi_2\cdots d\varphi_{d-1}}{\int_0^{2\pi}\int_0^\pi\cdots\int_0^\pi \sin^{d-2}(\varphi_1)\sin^{d-3}(\varphi_2)\cdots\sin(\varphi_{d-2})\,d\varphi_1\,d\varphi_2\cdots d\varphi_{d-1}}\\
&= \frac{\int_{-\frac{\pi}{2}+\theta}^{\frac{\pi}{2}} A(\theta, \varphi_{d-1}^{\boldsymbol{v}}) \cdot d\varphi_{d-1}}{\int_0^{2\pi} d\varphi_{d-1}} \cdot \frac{d-1}{d}\frac{d-2}{d-1}\cdots\frac{2}{3} \text{ (by Lemma 18)}\\
&= \frac{\sin\theta + (\pi - \theta)\cos\theta}{2d \cdot \pi} \text{ (by Lemma 19)}.
\end{aligned}$$

The result of this lemma thus follows. $\qquad\square$

## F.7 Convergence of $\frac{1}{p_1}(h_{\mathbf{V},\boldsymbol{x}}^{\mathbf{RF}})^T h_{\mathbf{V},\boldsymbol{z}}^{\mathbf{RF}}$ with respect to $p_1$

**Lemma 22** (Theorem 4.2 of Wainwright (2015)). *Let $\mathscr{F}$ be a class of real-valued functions $f$ such that $\|f\|_\infty \leq b$ for all $f \in \mathscr{F}$. Then for all $k \geq 1$ and $\delta \geq 0$, we have*

$$\Pr\left\{\sup_{f \in \mathscr{F}} \left|\frac{1}{k}\sum_{i=1}^k f(X_i) - \mathop{\mathsf{E}}_{x \sim \mathcal{X}(\cdot)} f(x)\right| \leq 2\mathcal{R}_k(\mathscr{F}) + \delta\right\} \geq 1 - \exp\left(-\frac{k\delta^2}{8b^2}\right),$$

*where $\mathcal{R}_k(\mathscr{F})$ denotes the Rademacher complexity, $X_1, X_2, \cdots, X_k$ are* i.i.d. *random variables/vectors that follow the distribution $\mathcal{X}(\cdot)$.*

**Polynomial discrimination**. A class $\mathscr{F}$ of functions with domain $\mathcal{X}$ has polynomial discrimination of order $\nu \geq 1$ if for each positive integer $k$ and collection $X_1^k = \{X_1, \cdots, X_k\}$ of $k$ points in $\mathcal{X}$, the set $\mathscr{F}(X_1^k)$ has cardinality upper bounded by

$$\mathrm{card}(\mathscr{F}(X_1^k)) \leq (k+1)^\nu.$$

**Lemma 23** (Lemma 4.1 and Eq. (4.23) of Wainwright (2015)). *Suppose that $\mathscr{F}$ has polynomial discrimination of order $\nu$ and $\|f\|_\infty \leq b$ for all $f \in \mathscr{F}$. Then*

$$\mathcal{R}_k(\mathscr{F}) \leq 3\sqrt{\frac{b^2\nu\log(k+1)}{k}} \quad \text{for all } k \geq 10.$$

Given a function $h : \mathcal{S}^{d-1} \mapsto \mathbb{R}$ such that $\|h\|_\infty < \infty$ and given any $\delta > 0$, consider the function class $\mathscr{F}_*$ that consists of functions $h(\boldsymbol{v})1_{\{\boldsymbol{x}^T\boldsymbol{v}>0,\boldsymbol{z}^T\boldsymbol{v}>0\}}$, which maps $\boldsymbol{v} \in \mathcal{S}^{d-1}$ to either 0 or $h(\boldsymbol{v})$. By Lemma 20 of Ju et al. (2021), we have

$$\mathrm{card}(\mathscr{F}_*(X_1^k)) \leq (k+1)^{2(d+1)}.$$

(Here $X_1^k$ corresponds to $\{\mathbf{V}[1], \cdots, \mathbf{V}[k]\}$.) Thus, combined with Lemma 22 and Lemma 23, we have

$$\Pr_{\mathbf{V}}\left\{\max_{\boldsymbol{x},\boldsymbol{z}}\left|\frac{1}{p_1}\sum_{j=1}^{p_1} h(\mathbf{V}[j])1_{\{\boldsymbol{x}^T\mathbf{V}[j]>0,\boldsymbol{z}^T\mathbf{V}[j]>0\}} - \mathop{\mathsf{E}}_{\boldsymbol{v}}[h(\boldsymbol{v})1_{\{\boldsymbol{x}^T\boldsymbol{v}>0,\boldsymbol{z}^T\boldsymbol{v}>0\}}]\right|\right.$$

$$\left.\leq 6\sqrt{\frac{\|h\|_\infty^2 2(d+1)\log(p_1+1)}{p_1}} + \delta\right\} \geq 1 - \exp\left(-\frac{p_1\delta^2}{8\|h\|_\infty^2}\right).$$

Further, if we let $\delta = 2\sqrt{\frac{\|h\|_\infty^2 2(d+1)\log(p_1+1)}{p_1}}$, we have proven the following lemma.

**Lemma 24.** *For any given function $h : \mathcal{S}^{d-1} \mapsto \mathbb{R}$ that $\|h\|_\infty < \infty$, when $p_1 \geq 10$, we have*

$$\Pr_{\mathbf{V}}\left\{\max_{\boldsymbol{x},\boldsymbol{z}}\left|\frac{1}{p_1}\sum_{j=1}^{p_1} h(\mathbf{V}[j])1_{\{\boldsymbol{x}^T\mathbf{V}[j]>0,\boldsymbol{z}^T\mathbf{V}[j]>0\}} - \mathop{\mathsf{E}}_{\boldsymbol{v}}[h(\boldsymbol{v})1_{\{\boldsymbol{x}^T\boldsymbol{v}>0,\boldsymbol{z}^T\boldsymbol{v}>0\}}]\right|\right.$$

$$\left.\leq 8\sqrt{\frac{\|h\|_\infty^2 2(d+1)\log(p_1+1)}{p_1}}\right\} \geq 1 - \frac{1}{(p_1+1)e^{d+1}}.$$

By Eq. (1), we have

$$\frac{1}{p_1}(h_{\mathbf{V},\boldsymbol{x}}^{\mathbf{RF}})^T h_{\mathbf{V},\boldsymbol{z}}^{\mathbf{RF}} = \frac{1}{p_1}\sum_{j=1}^{p_1}(\boldsymbol{x}^T\mathbf{V}[j])(\mathbf{V}[j]^T\boldsymbol{z})1_{\{\boldsymbol{x}^T\mathbf{V}[j]>0,\boldsymbol{z}^T\mathbf{V}[j]>0\}}$$

$$= \boldsymbol{x}^T\left(\frac{1}{p_1}\sum_{j=1}^{p_1}(\mathbf{V}[j]\mathbf{V}[j]^T)1_{\{\boldsymbol{x}^T\mathbf{V}[j]>0,\boldsymbol{z}^T\mathbf{V}[j]>0\}}\right)\boldsymbol{z}.$$

Notice that $\mathbf{V}[j]\mathbf{V}[j]^T$ is a $d \times d$ matrix. Define

$$\mathbf{K}_j := (\mathbf{V}[j]\mathbf{V}[j]^T)1_{\{\boldsymbol{x}^T\mathbf{V}[j]>0,\boldsymbol{z}^T\mathbf{V}[j]>0\}} \in \mathbb{R}^{d \times d}.$$

Thus, we have

$$\max_{\boldsymbol{x},\boldsymbol{z}} \left| \frac{1}{p_1}(\boldsymbol{h}^{\mathrm{RF}}_{\mathbf{V},\boldsymbol{x}})^T \boldsymbol{h}^{\mathrm{RF}}_{\mathbf{V},\boldsymbol{z}} - K^{\mathrm{RF}}(\boldsymbol{x},\boldsymbol{z}) \right|$$

$$= \max_{\boldsymbol{x},\boldsymbol{z}} \left| \boldsymbol{x}^T \left( \frac{1}{p_1}\sum_{j=1}^{p_1} \mathbf{K}_j - \mathop{\mathsf{E}}_{\boldsymbol{v}\sim\lambda(\cdot)}(\boldsymbol{v}\boldsymbol{v}^T)1_{\{\boldsymbol{x}^T\boldsymbol{v}>0,\boldsymbol{z}^T\boldsymbol{v}>0\}} \right) \boldsymbol{z} \right|$$

$$\leq \max_{\boldsymbol{x},\boldsymbol{z}} \|\boldsymbol{x}^T\|_2 \cdot \left\| \left( \frac{1}{p_1}\sum_{j=1}^{p_1} \mathbf{K}_j - \mathop{\mathsf{E}}_{\boldsymbol{v}\sim\lambda(\cdot)}(\boldsymbol{v}\boldsymbol{v}^T)1_{\{\boldsymbol{x}^T\boldsymbol{v}>0,\boldsymbol{z}^T\boldsymbol{v}>0\}} \right) \right\|_2 \cdot \|\boldsymbol{z}\|_2 \text{ (by Lemma 12)}$$

$$= \max_{\boldsymbol{x},\boldsymbol{z}} \left\| \frac{1}{p_1}\sum_{j=1}^{p_1} \mathbf{K}_j - \mathop{\mathsf{E}}_{\boldsymbol{v}\sim\lambda(\cdot)}(\boldsymbol{v}\boldsymbol{v}^T)1_{\{\boldsymbol{x}^T\boldsymbol{v}>0,\boldsymbol{z}^T\boldsymbol{v}>0\}} \right\|_2 \text{ (because } \|\boldsymbol{x}\|_2 = \|\boldsymbol{z}\|_2 = 1). \qquad (33)$$

For any $k,l \in \{1,2,\cdots,d\}$, define the $(k,l)$-th element of $\mathbf{K}_j$ as $K_{j,k,l}$. Thus, by Lemma 24 (notice that $|K_{j,k,l}| \leq 1$), we have

$$\Pr_{\mathbf{V}} \left\{ \max_{\boldsymbol{x},\boldsymbol{z}} \left| \frac{1}{p_1}\sum_{j=1}^{p_1} K_{j,k.l} - \left( \mathop{\mathsf{E}}_{\boldsymbol{v}\sim\lambda(\cdot)}(\boldsymbol{v}\boldsymbol{v}^T)1_{\{\boldsymbol{x}^T\boldsymbol{v}>0,\boldsymbol{z}^T\boldsymbol{v}>0\}} \right)_{k,l} \right| \leq 8\sqrt{\frac{2(d+1)\log(p_1+1)}{p_1}} \right\}$$

$$\geq 1 - \frac{1}{(p_1+1)e^{d+1}}.$$

Applying the union bound on all $d \times d$ elements of $\mathbf{K}_j$ and by Lemma 13, we have

$$\Pr_{\mathbf{V}} \left\{ \max_{\boldsymbol{x},\boldsymbol{z}} \left\| \frac{1}{p_1}\sum_{j=1}^{p_1} \mathbf{K}_j - \mathop{\mathsf{E}}_{\boldsymbol{v}\sim\lambda(\cdot)}(\boldsymbol{v}\boldsymbol{v}^T)1_{\{\boldsymbol{x}^T\boldsymbol{v}>0,\boldsymbol{z}^T\boldsymbol{v}>0\}} \right\|_2 \leq 8d\sqrt{\frac{2(d+1)\log(p_1+1)}{p_1}} \right\}$$

$$\geq 1 - \frac{d^2}{(p_1+1)e^{d+1}}.$$

Plugging it into Eq. (33), we thus have proven the following lemma.

**Lemma 25.** *Recall the definition of $Q(\cdot,\cdot)$ in Eq. (18). When $p_1 \geq 10$, we have*

$$\Pr_{\mathbf{V}} \left\{ \max_{\boldsymbol{x},\boldsymbol{z}} \left| \frac{1}{p_1}(\boldsymbol{h}^{RF}_{\mathbf{V},\boldsymbol{x}})^T \boldsymbol{h}^{RF}_{\mathbf{V},\boldsymbol{z}} - K^{RF}(\boldsymbol{x},\boldsymbol{z}) \right| \leq Q(p_1,d) \right\} \geq 1 - \frac{d^2}{(p_1+1)e^{d+1}}.$$

### F.8 Some useful lemmas about multinomial expansion

**Lemma 26** (Multinomial theorem (multinomial expansion)). *For any positive integer $i$ and non-negative integer $j$,*

$$(x_1 + x_2 + \cdots + x_i)^j = \sum_{k_1+k_2+\cdots k_i=j} (k_1,k_2,\cdots,k_i)! \cdot x_1^{k_1} x_2^{k_2} \cdots x_i^{k_i},$$

*where*

$$(k_1,k_2,\cdots,k_i)! = \frac{(k_1+k_2+\cdots+k_i)!}{k_1!k_2!\cdots k_i!}$$

*denotes the multinomial coefficient.*

**Lemma 27.** *We have*

$$\left( \sum_{i=0}^{\infty} a_i x^i \right)^j = \sum_{s=0}^{\infty} \left( \sum_{\substack{k_0+k_1+\cdots+k_s=j \\ k_1+2k_2+\cdots+sk_s=s \\ k_0,k_1,\cdots,k_s\in\mathbb{Z}_{\geq 0}}} (k_1,k_2,\cdots,k_s)! \cdot a_0^{k_0} a_1^{b_1} \cdots a_s^{k_s} \right) x^s.$$

*Proof.* The result directly follows from Lemma 26. Notice that $a_i x^i$ will not contribute to $x^s$ when $i > s$. $\qquad \square$

# G Proof of Proposition 4

Define

$$\Delta\mathbf{W}^*[k] := \int_{\mathcal{S}^{d-1}} 1_{\{(\boldsymbol{h}_{\mathbf{V},\boldsymbol{z}}^{\mathrm{RF}})^T \mathbf{W}_0[k]>0\}} \boldsymbol{h}_{\mathbf{V},\boldsymbol{z}}^{\mathrm{RF}} \frac{g(\boldsymbol{z})}{p_1 p_2} d\mu(\boldsymbol{z}), \quad k = 1, 2, \cdots, p_2. \tag{34}$$

Notice that $\Delta\mathbf{W}^*[k]$ is a vector of size $p_1 \times 1$ (same as the size of $\boldsymbol{h}_{\mathbf{V},\boldsymbol{z}}^{\mathrm{RF}}$ and $\boldsymbol{h}_{\mathbf{V},\boldsymbol{x}}^{\mathrm{RF}}$). The connection between $\Delta\mathbf{W}^*$ and the pseudo ground-truth $f_{\mathbf{V},\mathbf{W}_0}^g$ is shown by the following lemma.

**Lemma 28.** *For all $\boldsymbol{x} \in \mathcal{S}^{d-1}$, we have*

$$\boldsymbol{h}_{\mathbf{V},\mathbf{W}_0,\boldsymbol{x}}^{Three} \cdot \Delta\mathbf{W}^* = f_{\mathbf{V},\mathbf{W}_0}^g(\boldsymbol{x}).$$

*Proof.* We have

$$\begin{aligned}
&\boldsymbol{h}_{\mathbf{V},\mathbf{W}_0,\boldsymbol{x}}^{\mathrm{Three}} \cdot \Delta\mathbf{W}^* \\
&= \sum_{k=1}^{p_2} (\boldsymbol{h}_{\mathbf{V},\mathbf{W}_0,\boldsymbol{x}}^{\mathrm{Three}}[k])^T \Delta\mathbf{W}^*[k] \\
&= \sum_{k=1}^{p_2} \int_{\mathcal{S}^{d-1}} 1_{\{(\boldsymbol{h}_{\mathbf{V},\boldsymbol{x}}^{\mathrm{RF}})^T \mathbf{W}_0[k]>0, (\boldsymbol{h}_{\mathbf{V},\boldsymbol{z}}^{\mathrm{RF}})^T \mathbf{W}_0[k]>0\}} (\boldsymbol{h}_{\mathbf{V},\boldsymbol{x}}^{\mathrm{RF}})^T \boldsymbol{h}_{\mathbf{V},\boldsymbol{z}}^{\mathrm{RF}} \frac{g(\boldsymbol{z})}{p_1 p_2} d\mu(\boldsymbol{z}) \\
&\quad \text{(by Eq. (2) and Eq. (34))} \\
&= \int_{\mathcal{S}^{d-1}} \sum_{k=1}^{p_2} \frac{1_{\{(\boldsymbol{h}_{\mathbf{V},\boldsymbol{x}}^{\mathrm{RF}})^T \mathbf{W}_0[k]>0, (\boldsymbol{h}_{\mathbf{V},\boldsymbol{z}}^{\mathrm{RF}})^T \mathbf{W}_0[k]>0\}}}{p_1 p_2} (\boldsymbol{h}_{\mathbf{V},\boldsymbol{x}}^{\mathrm{RF}})^T \boldsymbol{h}_{\mathbf{V},\boldsymbol{z}}^{\mathrm{RF}} g(\boldsymbol{z}) d\mu(\boldsymbol{z}) \\
&= \int_{\mathcal{S}^{d-1}} (\boldsymbol{h}_{\mathbf{V},\boldsymbol{z}}^{\mathrm{RF}})^T \boldsymbol{h}_{\mathbf{V},\boldsymbol{x}}^{\mathrm{RF}} \frac{\left| \mathcal{C}_{\boldsymbol{h}_{\mathbf{V},\boldsymbol{x}}^{\mathrm{RF}}, \boldsymbol{h}_{\mathbf{V},\boldsymbol{z}}^{\mathrm{RF}}}^{\mathbf{W}_0} \right|}{p_1 p_2} g(\boldsymbol{z}) d\mu(\boldsymbol{z}) \text{ (by Eq. (11))} \\
&= f_{\mathbf{V},\mathbf{W}_0}^g(\boldsymbol{x}) \text{ (by Eq. (19))}.
\end{aligned}$$

$\square$

The following lemma bounds the test error for the pseudo ground-truth function with respect to the distance between $\Delta\mathbf{W}^*$ and the row-space of $\mathbf{H}$.

**Lemma 29.** *For all $\boldsymbol{a} \in \mathbb{R}^n$, we have*

$$|f_{\mathbf{V},\mathbf{W}_0}^g(\boldsymbol{x}) - \hat{f}^{\ell_2}(\boldsymbol{x})| \leq \sqrt{p_1 p_2} \|\Delta\mathbf{W}^* - \mathbf{H}^T \boldsymbol{a}\|_2.$$

*Proof.* Define $\mathbf{P} := \mathbf{H}^T (\mathbf{H}\mathbf{H}^T)^{-1} \mathbf{H}$. It is easy to verify that $\mathbf{P}^2 = \mathbf{P} = \mathbf{P}^T$, so $\mathbf{P}$ is an orthogonal projection onto the space spanned by the rows of $\mathbf{H}$. By Lemma 28 and Eq. (3), when $\boldsymbol{\epsilon} = \mathbf{0}$ and the ground-truth function is $f_{\mathbf{V},\mathbf{W}_0}^g$, we have $\mathbf{F}(\mathbf{X}) = \mathbf{H}\Delta\mathbf{W}^*$ and

$$\hat{f}^{\ell_2}(\boldsymbol{x}) = \boldsymbol{h}_{\mathbf{V},\mathbf{W}_0,\boldsymbol{x}}^{\mathrm{Three}} \mathbf{H}^T (\mathbf{H}\mathbf{H}^T)^{-1} \mathbf{H}\Delta\mathbf{W}^* = \boldsymbol{h}_{\mathbf{V},\mathbf{W}_0,\boldsymbol{x}}^{\mathrm{Three}} \mathbf{P}\Delta\mathbf{W}^*.$$

Thus, by Lemma 28, we have

$$|f_{\mathbf{V},\mathbf{W}_0}^g(\boldsymbol{x}) - \hat{f}^{\ell_2}(\boldsymbol{x})| = |\boldsymbol{h}_{\mathbf{V},\mathbf{W}_0,\boldsymbol{x}}^{\mathrm{Three}}(\mathbf{P} - \mathbf{I})\Delta\mathbf{W}^*|. \tag{35}$$

Because $\mathbf{P} = \mathbf{H}^T (\mathbf{H}\mathbf{H}^T)^{-1} \mathbf{H}$, we have

$$\mathbf{P}\mathbf{H}^T = \mathbf{H}^T (\mathbf{H}\mathbf{H}^T)^{-1} \mathbf{H}\mathbf{H}^T = \mathbf{H}^T. \tag{36}$$

We then have

$$\begin{aligned}
\|(\mathbf{P} - \mathbf{I})\Delta\mathbf{W}^*\|_2 &= \|\mathbf{P}\Delta\mathbf{W}^* - \Delta\mathbf{W}^*\|_2 \\
&= \|\mathbf{P}(\mathbf{H}^T \boldsymbol{a} + \Delta\mathbf{W}^* - \mathbf{H}^T \boldsymbol{a}) - \Delta\mathbf{W}^*\|_2 \\
&= \|\mathbf{P}\mathbf{H}^T \boldsymbol{a} + \mathbf{P}(\Delta\mathbf{W}^* - \mathbf{H}^T \boldsymbol{a}) - \Delta\mathbf{W}^*\|_2 \\
&= \|\mathbf{H}^T \boldsymbol{a} + \mathbf{P}(\Delta\mathbf{W}^* - \mathbf{H}^T \boldsymbol{a}) - \Delta\mathbf{W}^*\|_2 \text{ (by Eq. (36))} \\
&= \|(\mathbf{P} - \mathbf{I})(\Delta\mathbf{W}^* - \mathbf{H}^T \boldsymbol{a})\|_2 \\
&\leq \|\Delta\mathbf{W}^* - \mathbf{H}^T \boldsymbol{a}\|_2 \text{ (because } \mathbf{P} \text{ is an orthogonal projection).} \tag{37}
\end{aligned}$$

Therefore, we have
$$\left| h_{\mathbf{V},\mathbf{W}_0,\boldsymbol{x}}^{\text{Three}}(\mathbf{P} - \mathbf{I})\Delta\mathbf{W}^* \right| = \left\| h_{\mathbf{V},\mathbf{W}_0,\boldsymbol{x}}^{\text{Three}}(\mathbf{P} - \mathbf{I})\Delta\mathbf{W}^* \right\|_2$$
$$\leq \| h_{\mathbf{V},\mathbf{W}_0,\boldsymbol{x}}^{\text{Three}} \|_2 \cdot \| (\mathbf{P} - \mathbf{I})\Delta\mathbf{W}^* \|_2 \text{ (by Lemma 12)}$$
$$\leq \sqrt{p_1 p_2} \| \Delta\mathbf{W}^* - \mathbf{H}^T \boldsymbol{a} \|_2 \text{ (by Lemma 11 and Eq. (37))}.$$
By Eq. (35), the result of this lemma thus follows. $\qquad\square$

Now we are ready to prove Proposition 4.

Define $\mathbf{K}_i \in \mathbb{R}^{(p_1 p_2) \times 1}$ (the same shape as $\mathbf{W}_0$) as
$$\mathbf{K}_i[k] := h_{\mathbf{V},\mathbf{X}_i}^{\text{RF}} 1_{\{(h_{\mathbf{V},\mathbf{X}_i}^{\text{RF}})^T \mathbf{W}_0[k]>0\}} \frac{g(\mathbf{X}_i)}{p_1 p_2}, \ i \in \{1, 2, \cdots, n\}, \ k \in \{1, 2, \cdots, p_2\}. \tag{38}$$
It is obvious that $\mathbf{K}_1, \mathbf{K}_2, \cdots, \mathbf{K}_n$ are *i.i.d.* with respect to the randomness of $\mathbf{X}$. By Eq. (34), for all $k = 1, 2, \cdots, p_2$, we have
$$\mathop{\mathbb{E}}_{\mathbf{X}_i} [\mathbf{K}_i[k]] = \Delta\mathbf{W}^*[k]. \tag{39}$$

Further, note that
$$\|\mathbf{K}_i[k]\|_2 \leq \frac{\|g\|_\infty}{p_1 p_2} \|h_{\mathbf{V},\mathbf{X}_i}^{\text{RF}}\|_2 \text{ (by Lemma 12 and Eq. (38))}$$
$$\leq \frac{\|g\|_\infty}{\sqrt{p_1 p_2}} \text{ (by Lemma 11)}.$$

Thus, we have
$$\|\mathbf{K}_i\|_2 = \sqrt{\sum_{k=1}^{p_2} \|\mathbf{K}_i[k]\|_2^2} \leq \frac{\|g\|_\infty}{\sqrt{p_1 p_2}},$$
i.e.,
$$\sqrt{p_1 p_2}\|\mathbf{K}_i\|_2 \leq \|g\|_\infty. \tag{40}$$

We now construct the vector $\boldsymbol{a} \in \mathbb{R}^n$ that we will use in Lemma 29. Its $i$-th element is $\boldsymbol{a}_i = \frac{g(\mathbf{X}_i)}{n p_1 p_2}$, $i = 1, 2, \cdots, n$. Then, for all $k \in \{1, 2, \cdots, p_2\}$, we have
$$(\mathbf{H}^T \boldsymbol{a})[k] = \sum_{i=1}^n \mathbf{H}_i^T[k]\boldsymbol{a}_i$$
$$= \sum_{i=1}^n h_{\mathbf{V},\mathbf{X}_i}^{\text{RF}} 1_{\{(h_{\mathbf{V},\mathbf{X}_i}^{\text{RF}})^T \mathbf{W}_0[k]>0\}} \frac{g(\mathbf{X}_i)}{n p_1 p_2} \text{ (by Eq. (2))}$$
$$= \frac{1}{n} \sum_{i=1}^n \mathbf{K}_i[k] \text{ (by Eq. (38))},$$
i.e.,
$$\mathbf{H}^T \boldsymbol{a} = \frac{1}{n} \sum_{i=1}^n \mathbf{K}_i. \tag{41}$$

Thus, by Lemma 14 (with $X_i = \sqrt{p_1 p_2}\mathbf{K}_i$, $U = \|g\|_\infty$, $m = q$), we have
$$\mathop{\Pr}_{\mathbf{X}} \left\{ \sqrt{p_1 p_2} \left\| \left(\frac{1}{n}\sum_{i=1}^n \mathbf{K}_i\right) - \mathop{\mathbb{E}}_{\mathbf{X}} \mathbf{K}_1 \right\|_2 \geq \frac{q\|g\|_\infty}{\sqrt{n}} \right\} \leq \frac{1}{q^2}.$$
Further, by Eq. (41) and Eq. (39), we have
$$\mathop{\Pr}_{\mathbf{X}} \left\{ \sqrt{p_1 p_2} \left\| \mathbf{H}^T \boldsymbol{a} - \Delta\mathbf{W}^* \right\|_2 \geq \frac{q\|g\|_\infty}{\sqrt{n}} \right\} \leq \frac{1}{q^2}.$$
By Lemma 29, we thus have
$$\mathop{\Pr}_{\mathbf{X}} \left\{ |f_{\mathbf{V},\mathbf{W}_0}^g(\boldsymbol{x}) - \hat{f}^{\ell_2}(\boldsymbol{x})| \geq \frac{q\|g\|_\infty}{\sqrt{n}} \right\} \leq \frac{1}{q^2}.$$
The result of Proposition 4 thus follows.

# H   Proof of Proposition 6 (Minimum Eigenvalue of $\mathrm{HH}^T$)

Define

$$\beta_{i,j} := \left\| \boldsymbol{h}^{\mathrm{RF}}_{\mathbf{V},\mathbf{X}_i} \right\|_2 \cdot \left\| \boldsymbol{h}^{\mathrm{RF}}_{\mathbf{V},\mathbf{X}_j} \right\|_2,$$

$$\theta^{\mathrm{RF}}_{i,j} := \arccos \left( \frac{(\boldsymbol{h}^{\mathrm{RF}}_{\mathbf{V},\mathbf{X}_i})^T \boldsymbol{h}^{\mathrm{RF}}_{\mathbf{V},\mathbf{X}_j}}{\beta_{i,j}} \right) \in \left[ 0, \frac{\pi}{2} \right],$$

$$\theta^{\mathrm{RF}}_{\min} := \min_{i \neq j} \theta^{\mathrm{RF}}_{i,j}.$$

(By Eq. (1), we know that every element of $\boldsymbol{h}^{\mathrm{RF}}_{\mathbf{V},\mathbf{X}_i}$ and $\boldsymbol{h}^{\mathrm{RF}}_{\mathbf{V},\mathbf{X}_j}$ are non-negative, and hence $\theta^{\mathrm{RF}}_{i,j} \in [0, \frac{\pi}{2}]$.)

Define $\tilde{\mathbf{H}}^\infty \in \mathbb{R}^{n \times n}$ as

$$\tilde{\mathbf{H}}^\infty_{i,j} := \frac{p_1}{2d} \cos(\theta^{\mathrm{RF}}_{i,j}) \cdot \frac{\pi - \theta^{\mathrm{RF}}_{i,j}}{2\pi}. \tag{42}$$

The following lemma (restated) is from the proof of Lemma 1 of Satpathi & Srikant (2021), which relates $\min \mathrm{eig}(\tilde{\mathbf{H}}^\infty)$ to $\theta^{\mathrm{RF}}_{\min}$. For reader's convenience, we also provide its proof in Appendix H.1.

**Lemma 30.**

$$\min \mathrm{eig}(\tilde{\mathbf{H}}^\infty) \geq \frac{1}{8\pi} \cdot \frac{p_1}{2d} \cdot \sqrt{\frac{\log(1/\cos \theta^{RF}_{\min})}{\log(2n/\cos \theta^{RF}_{\min})}}.$$

We then focus on estimating $\theta^{\mathrm{RF}}_{\min}$.

**Lemma 31.** *Recall the definition of $C(n, d, q)$ in Eq. (16). For any $q > 0$, when $p_1$ is sufficient large such that*

$$\frac{10dnq\sqrt{2d}}{\sqrt{p_1}} \leq C(n, d, q). \tag{43}$$

*we have*

$$\Pr_{\mathbf{V},\mathbf{X}} \left\{ \cos \theta^{RF}_{\min} \geq 1 - C(n, d, q) \right\} \leq \frac{4}{q^2}.$$

The proof of Lemma 31 is in Appendix H.2. Intuitively, when $n$ becomes larger, some $\mathbf{X}_i$'s (together with $\boldsymbol{h}^{\mathrm{RF}}_{\mathbf{V},\mathbf{X}_i}$'s) will get closer to each other, and thus $\theta^{\mathrm{RF}}_{\min}$ will get closer to zero. Such intuition is captured by Lemma 31 since $C(n, d, q)$ is monotone decreasing with respect to $n$.

The above lemmas study the minimum eigenvalue of $\tilde{\mathbf{H}}^\infty$. We need to relate it to the minimum eigenvalue of $\mathbf{HH}^T$, which is achieved by the following lemma.

**Lemma 32.** *For any $q > 0$,*

$$\Pr_{\mathbf{X},\mathbf{V},\mathbf{W}_0} \left\{ \left| \frac{1}{p_2} \min \mathrm{eig}(\mathbf{HH}^T) - \min \mathrm{eig}(\tilde{\mathbf{H}}^\infty) \right| \geq qn^2\sqrt{2p_1 d} + q^2 n^3 d + \frac{qn^2 p_1}{\sqrt{p_2}} \right\} \leq \frac{3}{q^2}.$$

The proof of Lemma 32 is in Appendix H.3. From the derivation in Appendix B, we know that each element of $\frac{\mathbf{HH}^T}{p_1 p_2}$ will approach the corresponding element of $\frac{1}{p_1}\tilde{\mathbf{H}}^\infty$ as $p_1$ and $p_2$ get larger. Therefore, it is natural to expect that the minimum eigenvalue of those two matrices will also be closer to each other when $p_1$ and $p_2$ becomes larger, which is captured by Lemma 32.

**Lemma 33.** *For any $a \in (0, 1]$, we have $\log \frac{1}{a} \geq 1 - a$.*

*Proof.* Consider the function $h(a) := \log(1/a) - 1 + a$. We have $\frac{\partial h(a)}{\partial a} = -\frac{1}{a} + 1 \leq 0$. Thus, we know $h(a)$ is monotone decreasing in $a \in (0, 1]$. Thus, we have $h(a) \geq h(1) = 0$. The result of this lemma thus follows. $\square$

Now we are ready to prove Proposition 6.

*Proof of Proposition 6.* We define three events

$$\mathcal{J}_1 := \left\{ \cos\theta_{\min}^{\mathrm{RF}} \geq 1 - C(n,d,q) \right\},$$

$$\mathcal{J}_2 := \left\{ \left| \frac{1}{p_2} \min\mathrm{eig}(\mathbf{HH}^T) - \min\mathrm{eig}(\tilde{\mathbf{H}}^\infty) \right| \geq qn^2\sqrt{2p_1 d} + q^2 n^3 d + \frac{qn^2 p_1}{\sqrt{p_2}} \right\},$$

$$\mathcal{J}_3 := \left\{ \frac{1}{p_2} \min\mathrm{eig}(\mathbf{HH}^T) \leq p_1 J(n,p_1,p_2,d,q) \triangleq \frac{p_1}{16\pi d}\sqrt{\frac{C(n,d,q)}{\log(4n)}} \right.$$

$$\left. - \left( qn^2\sqrt{2p_1 d} + q^2 n^3 d + \frac{qn^2 p_1}{\sqrt{p_2}} \right) \right\}.$$

**Step 1: prove $\mathcal{J}_1 \cup \mathcal{J}_2 \supseteq \mathcal{J}_3$.**

In order to prove $\mathcal{J}_1 \cup \mathcal{J}_2 \supseteq \mathcal{J}_3$, it is equivalent to prove $\mathcal{J}_1^c \cap \mathcal{J}_2^c \subseteq \mathcal{J}_3^c$. To that end, suppose $\mathcal{J}_1^c$ and $\mathcal{J}_2^c$ happen. Thus, we have

$$\log(1/\cos\theta_{\min}^{\mathrm{RF}}) \geq 1 - \cos\theta_{\min}^{\mathrm{RF}} \quad \text{(by Lemma 33)}$$
$$\geq C(n,d,q) \quad \text{(by the event } \mathcal{J}_1^c). \tag{44}$$

Thus, we have

$$\min\mathrm{eig}(\tilde{\mathbf{H}}^\infty)$$

$$\geq \frac{1}{8\pi} \cdot \frac{p_1}{2d}\sqrt{\frac{\log(1/\cos\theta_{\min}^{\mathrm{RF}})}{\log(2n/\cos\theta_{\min}^{\mathrm{RF}})}} \quad \text{(by Lemma 30)}$$

$$= \frac{p_1}{16\pi d}\sqrt{\frac{\log(1/\cos\theta_{\min}^{\mathrm{RF}})}{\log(2n) + \log(1/\cos\theta_{\min}^{\mathrm{RF}})}}$$

$$\geq \frac{p_1}{16\pi d}\sqrt{\frac{C(n,d,q)}{\log(2n) + C(n,d,q)}}$$

$$\text{(by Eq. (44) and } \frac{a}{\log(2n) + a} \text{ is monotone increasing with respect to } a)$$

$$\geq \frac{p_1}{16\pi d}\sqrt{\frac{C(n,d,q)}{\log(4n)}} \quad \text{(since } \log(2) \approx 0.7, C(n,d,q) \leq \frac{\pi-1}{4\pi} \cdot \frac{1}{2} \leq \frac{1}{8} \leq \log 2).$$

Thus, we have

$$\frac{1}{p_2}\min\mathrm{eig}(\mathbf{HH}^T) \geq \min\mathrm{eig}(\tilde{\mathbf{H}}^\infty) - \left| \frac{1}{p_2}\min\mathrm{eig}(\mathbf{HH}^T) - \min\mathrm{eig}(\tilde{\mathbf{H}}^\infty) \right|$$

$$\text{(by the triangle inequality)}$$

$$> \frac{p_1}{16\pi d}\sqrt{\frac{C(n,d,q)}{\log(4n)}} - \left( qn^2\sqrt{2p_1 d} + q^2 n^3 d + \frac{qn^2 p_1}{\sqrt{p_2}} \right) \quad \text{(by the event } \mathcal{J}_2^c)$$

$$= p_1 J(n,p_1,p_2,d,q) \quad \text{(by Eq. (17))},$$

i.e., $\mathcal{J}_3^c$ must then occur. Thus, we have shown that $\mathcal{J}_1^c \cap \mathcal{J}_2^c \subseteq \mathcal{J}_3^c$, which implies that $\mathcal{J}_1 \cup \mathcal{J}_2 \supseteq \mathcal{J}_3$.

**Step 2: estimate $\mathcal{J}_3$**

We have

$$\Pr_{\mathbf{X},\mathbf{V},\mathbf{W}_0}[\mathcal{J}_3] \leq \Pr_{\mathbf{X},\mathbf{V},\mathbf{W}_0}[\mathcal{J}_1 \cup \mathcal{J}_2] \quad \text{(because } \mathcal{J}_1 \cup \mathcal{J}_2 \supseteq \mathcal{J}_3)$$

$$\leq \Pr_{\mathbf{X},\mathbf{V},\mathbf{W}_0}[\mathcal{J}_1] + \Pr_{\mathbf{X},\mathbf{V},\mathbf{W}_0}[\mathcal{J}_2] \quad \text{(by the union bound)}$$

$$= \Pr_{\mathbf{X},\mathbf{V}}[\mathcal{J}_1] + \Pr_{\mathbf{X},\mathbf{V},\mathbf{W}_0}[\mathcal{J}_2] \quad \text{(as } \mathcal{J}_1 \text{ is independent of } \mathbf{W}_0)$$

$$\leq \frac{7}{q^2} \quad \text{(by Lemma 31 and Lemma 32)}.$$

The result of Proposition 6 thus follows. $\qquad\square$

In the rest of this section, we prove Lemma 30, Lemma 31, and Lemma 32.

## H.1 Proof of Lemma 30

*Proof.* For simplicity of notation, we define $\boldsymbol{a}_i \in \mathbb{R}^{p_1}$ as

$$\boldsymbol{a}_i := \frac{\boldsymbol{h}_{\mathbf{V},\mathbf{X}_i}^{\mathrm{RF}}}{\|\boldsymbol{h}_{\mathbf{V},\mathbf{X}_i}^{\mathrm{RF}}\|_2} \text{ for all } i = 1, 2, \cdots, n.$$

Let $\boldsymbol{a}_i^{\otimes k} \in \mathbb{R}^{p_1 k}$ (a column vector with $p_1 k$ elements) denote the $k$-time Kronecker product of the vector $\boldsymbol{a}_i$ with itself. We define

$$\mathbf{A} := [\boldsymbol{a}_1 \ \boldsymbol{a}_2 \ \cdots \ \boldsymbol{a}_n] \in \mathbb{R}^{p_1 \times n},$$

$$\mathbf{A}^{(k)} := \left[\boldsymbol{a}_1^{\otimes k} \ \boldsymbol{a}_2^{\otimes k} \ \cdots \ \boldsymbol{a}_n^{\otimes k}\right] \in \mathbb{R}^{(p_1 k) \times n},$$

$$\mathbf{B}^{(k)} := \left(\mathbf{A}^{(k)}\right)^T \mathbf{A}^{(k)}.$$

Thus, we have

$$\cos \theta_{i,j}^{\mathrm{RF}} = \boldsymbol{a}_i^T \boldsymbol{a}_j. \tag{45}$$

By the definition of Kronecker product, we thus have[11]

$$\left(\boldsymbol{a}_i^T \boldsymbol{a}_j\right)^k = \left(\boldsymbol{a}_i^{\otimes k}\right)^T \left(\boldsymbol{a}_j^{\otimes k}\right). \tag{46}$$

Thus, by Lemma 17, we have

$$\cos(\theta_{i,j}^{\mathrm{RF}}) \cdot \frac{\pi - \theta_{i,j}^{\mathrm{RF}}}{2\pi} = \frac{\cos \theta_{i,j}^{\mathrm{RF}}}{4} + \frac{1}{2\pi} \sum_{k=0}^{\infty} \frac{(2k)!}{(k!)^2} \frac{4}{2k+1} \left(\frac{\cos \theta_{i,j}^{\mathrm{RF}}}{2}\right)^{2k+2}$$

$$= \frac{\boldsymbol{a}_i^T \boldsymbol{a}_j}{4} + \frac{1}{2\pi} \sum_{k=0}^{\infty} \frac{(2k)!}{(k!)^2} \frac{4}{2k+1} \left(\frac{\boldsymbol{a}_i^T \boldsymbol{a}_j}{2}\right)^{2k+2}$$

$$= \frac{\boldsymbol{a}_i^T \boldsymbol{a}_j}{4} + \frac{1}{2\pi} \sum_{k=0}^{\infty} \frac{(2k)!}{(k!)^2} \frac{4}{2k+1} \left(\frac{1}{2}\right)^{2k+2} \left(\boldsymbol{a}_i^{\otimes 2k+2}\right)^T \boldsymbol{a}_j^{\otimes 2k+2}.$$

Using Eq. (42), we then have

$$\tilde{\mathbf{H}}^{\infty} = \frac{p_1}{2d} \left(\frac{\mathbf{A}^T \mathbf{A}}{4} + \frac{1}{2\pi} \sum_{k=0}^{\infty} \frac{(2k)!}{(k!)^2} \frac{4}{2k+1} \left(\frac{1}{2}\right)^{2k+2} \mathbf{B}^{(2k+2)}\right).$$

Thus, we have

$$\min \mathrm{eig}(\tilde{\mathbf{H}}^{\infty}) = \min_{\boldsymbol{u}:\, \|\boldsymbol{u}\|_2 = 1} \boldsymbol{u}^T \tilde{\mathbf{H}}^{\infty} \boldsymbol{u}$$

$$\geq \frac{p_1}{2d} \cdot \frac{1}{2\pi} \sum_{k=0}^{\infty} \frac{(2k)!}{(k!)^2} \frac{4}{2k+1} \left(\frac{1}{2}\right)^{2k+2} \min_{\boldsymbol{u}:\, \|\boldsymbol{u}\|_2 = 1} \boldsymbol{u}^T \mathbf{B}^{(2k+2)} \boldsymbol{u}$$

$$= \frac{p_1}{2d} \cdot \frac{1}{2\pi} \sum_{k=0}^{\infty} \frac{(2k)!}{(k!)^2} \frac{4}{2k+1} \left(\frac{1}{2}\right)^{2k+2} \min \mathrm{eig}(\mathbf{B}^{(2k+2)}). \tag{47}$$

---

[11]To help readers understand the correctness of Eq. (46), we give a toy example as follows. We have

$$\left(\begin{bmatrix} a & b \end{bmatrix} \begin{bmatrix} c \\ d \end{bmatrix}\right)^2 = (ac + bd)^2 = a^2 c^2 + 2abcd + b^2 d^2.$$

We also have

$$\begin{bmatrix} a \\ b \end{bmatrix}^{\otimes 2} = \begin{bmatrix} aa \\ ab \\ ba \\ bb \end{bmatrix}, \begin{bmatrix} c \\ d \end{bmatrix}^{\otimes 2} = \begin{bmatrix} cc \\ cd \\ dc \\ dd \end{bmatrix} \implies \left(\begin{bmatrix} a \\ b \end{bmatrix}^{\otimes 2}\right)^T \left(\begin{bmatrix} c \\ d \end{bmatrix}^{\otimes 2}\right) = a^2 c^2 + 2abcd + b^2 d^2.$$

Thus, we have shown that $\left(\begin{bmatrix} a & b \end{bmatrix} \begin{bmatrix} c \\ d \end{bmatrix}\right)^2 = \left(\begin{bmatrix} a \\ b \end{bmatrix}^{\otimes 2}\right)^T \left(\begin{bmatrix} c \\ d \end{bmatrix}^{\otimes 2}\right).$

Notice that all diagonal elements of $\mathbf{B}^{(2k+2)}$ equal to 1. Thus, by Gershgorin circle theorem (Bell, 1965), we have

$$\min\operatorname{eig}(\mathbf{B}^{(2k+2)}) \geq 1 - \max_i \sum_{j\neq i} \mathbf{B}_{ij}^{(2k+2)}, \tag{48}$$

where $\mathbf{B}_{ij}^{(2k+2)}$ denotes the $(i,j)$-th element of $\mathbf{B}^{(2k+2)}$. Notice that

$$
\begin{aligned}
\max_i \sum_{j\neq i} \mathbf{B}_{ij}^{(2k+2)} &= \max_i \sum_{j\neq i} \left(\boldsymbol{a}_i^{\otimes 2k+2}\right)^T \left(\boldsymbol{a}_j^{\otimes 2k+2}\right)\\
&= \max_i \sum_{j\neq i} \left(\cos\theta_{i,j}^{\mathrm{RF}}\right)^{2k+2} \quad\text{(by Eq. (46) and Eq. (45))}\\
&\leq (n-1)\left(\cos\theta_{\min}^{\mathrm{RF}}\right)^{2k+2}.
\end{aligned}
$$

Note that, when $k \geq k^* := \frac{\log(2n-2)}{2\log(1/\cos\theta_{\min}^{\mathrm{RF}})} - 1$, we have

$$
\begin{aligned}
2k+2 &\geq \frac{\log(2n-2)}{\log(1/\cos\theta_{\min}^{\mathrm{RF}})}\\
&\implies (2k+2)\log(1/\cos\theta_{\min}^{\mathrm{RF}}) \geq \log(2n-2)\\
&\implies (n-1)\left(\cos\theta_{\min}^{\mathrm{RF}}\right)^{2k+2} \leq \frac{1}{2}.
\end{aligned}
$$

Therefore, we have

$$\max_i \sum_{j\neq i} \mathbf{B}_{ij}^{(2k+2)} \leq \frac{1}{2}, \quad\text{for all } k\geq k^*.$$

By Eq. (47) and Eq. (48), we thus have

$$
\begin{aligned}
\min\operatorname{eig}(\tilde{\mathbf{H}}^\infty) &\geq \frac{p_1}{8d\pi}\sum_{k\geq k^*}\frac{(2k)!}{(k!)^2}\frac{4}{2k+1}\left(\frac{1}{2}\right)^{2k+2}\\
&= \frac{p_1}{8d\pi}\sum_{k\geq\lceil k^*\rceil}\frac{(2k-1)!!}{(2k)!!}\frac{1}{2k+1}\\
&\geq \frac{p_1}{8d\pi}\sum_{k\geq\lceil k^*\rceil}\frac{1}{\sqrt{\pi\left(k+\frac{4}{\pi}-1\right)}}\frac{1}{2k+1} \quad\text{(by Lemma 16)}\\
&\geq \frac{p_1}{8d\pi}\int_{k^*+1}^\infty \frac{1}{\sqrt{\pi\left(x+\frac{4}{\pi}-1\right)}}\frac{1}{2x+1}\,dx\\
&\geq \frac{p_1}{8d\pi}\int_{k^*+1}^\infty \frac{1}{2\sqrt{\pi}}(x+1)^{-\frac{3}{2}}\,dx \quad\text{(notice that } \frac{4}{\pi}-1\approx 0.27\leq 1)\\
&= \frac{p_1}{8d\pi}\frac{1}{\sqrt{\pi}}\frac{1}{\sqrt{k^*+2}}.
\end{aligned}
$$

Notice that $\frac{1}{\sqrt{\pi}}\geq\frac{1}{2}$ and

$$k^*+2 = \frac{\log(2n-2)}{2\log(1/\cos\theta_{\min}^{\mathrm{RF}})}+1 \leq \frac{\log(2n)}{\log(1/\cos\theta_{\min}^{\mathrm{RF}})}+1 = \frac{\log(2n/\cos\theta_{\min}^{\mathrm{RF}})}{\log(1/\cos\theta_{\min}^{\mathrm{RF}})}.$$

We thus have

$$\min\operatorname{eig}(\tilde{\mathbf{H}}^\infty) \geq \frac{p_1}{16d\pi}\sqrt{\frac{\log(1/\cos\theta_{\min}^{\mathrm{RF}})}{\log(2n/\cos\theta_{\min}^{\mathrm{RF}})}}.$$

$\square$

## H.2  Proof of Lemma 31

We first show some useful lemmas.

**Lemma 34.** *For any $\theta \in [0,\ \pi]$, we have*

$$1 - \frac{\sin\theta + (\pi - \theta)\cos\theta}{\pi} \geq \frac{\pi - 1}{2\pi}\sin^2\theta,$$

$$\lim_{\theta \to 0^+} \frac{1 - \frac{\sin\theta + (\pi - \theta)\cos\theta}{\pi}}{\sin^2\theta} = \frac{1}{2}.$$

*Proof.* To prove the first part, we have

$$\frac{\sin\theta + (\pi - \theta)\cos\theta}{\pi} \leq \frac{\sin\theta + (\pi - \theta)\sqrt{1 - \sin^2\theta}}{\pi}$$

$$\text{(although } \cos\theta \text{ could be negative, we always have } \cos\theta \leq \sqrt{1 - \sin^2\theta})$$

$$\leq \frac{\sin\theta + (\pi - \theta)\sqrt{1 - \sin^2\theta + \frac{1}{4}\sin^4\theta}}{\pi}$$

$$= \frac{\sin\theta + (\pi - \theta)\left(1 - \frac{1}{2}\sin^2\theta\right)}{\pi}$$

$$\leq \frac{\sin\theta + (\pi - \sin\theta)\left(1 - \frac{1}{2}\sin^2\theta\right)}{\pi} \quad \text{(because } \sin\theta \leq \theta)$$

$$= 1 - \frac{\pi - \sin\theta}{2\pi}\sin^2\theta$$

$$\leq 1 - \frac{\pi - 1}{2\pi}\sin^2\theta \quad \text{(because } \sin\theta \leq 1),$$

i.e.,

$$1 - \frac{\sin\theta + (\pi - \theta)\cos\theta}{\pi} \geq \frac{\pi - 1}{2\pi}\sin^2\theta.$$

To prove the second part, we have

$$\lim_{\theta \to 0^+} \frac{1 - \frac{\sin\theta + (\pi - \theta)\cos\theta}{\pi}}{\sin^2\theta} = \lim_{\theta \to 0^+} \frac{\frac{\partial}{\partial\theta}\left(1 - \frac{\sin\theta + (\pi - \theta)\cos\theta}{\pi}\right)}{\frac{\partial \sin^2\theta}{\partial\theta}} \quad \text{(by L'Hospital's rule)}$$

$$= \lim_{\theta \to 0^+} \frac{-\cos\theta + (\pi - \theta)\sin\theta + \cos\theta}{2\pi\sin\theta\cos\theta}$$

$$= \lim_{\theta \to 0^+} \frac{\pi - \theta}{2\pi\cos\theta}$$

$$= \frac{1}{2}.$$

$\square$

**Lemma 35.** *Consider $a \geq 0$ and $b > 0$. Let $\delta := |b - 1|$. If $\delta \in [0,\ 0.5]$, we then have*

$$a - a\delta \leq \frac{a}{b} \leq a + 2a\delta.$$

*Therefore, for any $c \in \mathbb{R}$, we have*

$$\left|\frac{a}{b} - c\right| \leq |a - c| + 2a\delta.$$

*Further, if we know the upper bound of $a$, we have the following conclusion: (i) if $a \leq 1$, we must have $\frac{a}{b} \leq a + 2\delta$; (ii) if $a \leq 1.5$, we must have $\left|\frac{a}{b} - c\right| \leq |a - c| + 3a$.*

*Proof.* We have

$$\frac{a}{b} \leq a\frac{1}{1-\delta} \text{ (because } b \geq 1 - |1-b| = 1-\delta)$$
$$\leq a\frac{1+(1-2\delta)\delta}{1-\delta} \text{ (by } (1-2\delta) \geq 0 \text{ because } \delta \in [0,\ 0.5])$$
$$= a\frac{1+\delta-2\delta^2}{1-\delta}$$
$$= a\frac{(1-\delta)(1+2\delta)}{1-\delta}$$
$$= a + 2a\delta.$$

We also have

$$\frac{a}{b} \geq a\frac{1}{1+\delta} \text{ (because } b \leq 1 + |1-b| = 1+\delta)$$
$$\geq a\frac{1-\delta^2}{1+\delta}$$
$$= a - a\delta.$$

The result of this lemma thus follows. □

**Lemma 36.** *If the condition in Eq.* (43) *is satisfied, then*

$$\frac{1 - 2C(n,d,q) + \frac{2dnq}{\sqrt{p_1}}}{1 - \frac{4dnq\sqrt{2d}}{\sqrt{p_1}} - \frac{4d^2n^2q^2}{p_1}} \leq 1 - C(n,d,q).$$

*Proof.* By Eq. (43) and the definition of $C(n,d,q)$ in Eq. (16), we have

$$\frac{10dnq\sqrt{2d}}{\sqrt{p_1}} \leq \frac{\pi-1}{4\pi} \cdot \frac{1}{2} \leq \frac{1}{8} \leq \frac{1}{2} \tag{49}$$
$$\implies \frac{dnq\sqrt{2d}}{\sqrt{p_1}} \leq \frac{1}{20}, \text{ and } \frac{d^2n^2q^2 \cdot 2d}{p_1} \leq \left(\frac{1}{20}\right)^2 = \frac{1}{400}$$
$$\implies \frac{4dnq\sqrt{2d}}{\sqrt{p_1}} + \frac{4d^2n^2q^2}{p_1} \leq \frac{1}{5} + \frac{1}{200d} \leq 0.5. \tag{50}$$

We also have

$$\frac{2dnq}{\sqrt{p_1}} + \frac{8dnq\sqrt{2d}}{\sqrt{p_1}} + \frac{8d^2n^2q^2}{p_1}$$
$$\leq \frac{dnq\sqrt{2d}}{\sqrt{p_1}} + \frac{8dnq\sqrt{2d}}{\sqrt{p_1}} + \frac{1}{2}\sqrt{\frac{8d^2n^2q^2}{p_1}} \text{ (because } 2 \leq \sqrt{2d} \text{ and } \frac{8d^2n^2q^2}{p_1} \leq \frac{1}{100d} \leq \frac{1}{4})$$
$$= \frac{9dnq\sqrt{2d}}{\sqrt{p_1}} + \frac{dnq\sqrt{2}}{\sqrt{p_1}}$$
$$\leq \frac{10dnq\sqrt{2d}}{\sqrt{p_1}}$$
$$\leq C(n,d,q) \text{ (by Eq. (43)).} \tag{51}$$

Thus, we have

$$\frac{2dnq}{\sqrt{p_1}} \leq C(n,d,q) \text{ (by Eq. (43))}$$
$$\implies 1 - 2C(n,d,q) + \frac{2dnq}{\sqrt{p_1}} \in [0,1] \quad \text{(since } C(n,d,q) \leq \frac{1}{8} \text{ by Eq. (49)).} \tag{52}$$

By Eq. (50), Eq. (52) and applying Lemma 35(i) (where $a = 1 - 2C(n,d,q) + \frac{2dnq}{\sqrt{p_1}}$, $b = 1 - \frac{4dnq\sqrt{2d}}{\sqrt{p_1}} - \frac{4d^2n^2q^2}{p_1}$, and $\delta = \frac{4dnq\sqrt{2d}}{\sqrt{p_1}} + \frac{4d^2n^2q^2}{p_1}$), we thus have

$$\frac{1 - 2C(n,d,q) + \frac{2dnq}{\sqrt{p_1}}}{1 - \frac{4dnq\sqrt{2d}}{\sqrt{p_1}} - \frac{4d^2n^2q^2}{p_1}}$$

$$\leq 1 - 2C(n,d,q) + \frac{2dnq}{\sqrt{p_1}} + \frac{8dnq\sqrt{2d}}{\sqrt{p_1}} + \frac{8d^2n^2q^2}{p_1}$$

$$\leq 1 - C(n,d,q) \text{ (by Eq. (51))}.$$

$\square$

**Lemma 37.** *Given* $\mathbf{X}$*, for any* $m > 0$*,*

$$\Pr_{\mathbf{V}} \left\{ \left| \beta_{i,j} - \frac{p_1}{2d} \right| \geq 2m\sqrt{2p_1 d} + 2m^2 d \right\} \leq \frac{2}{m^2}.$$

*In other words, given any* $\boldsymbol{x}, \boldsymbol{z} \in \mathcal{S}^{d-1}$ *and for any* $q > 0$*,*

$$\Pr_{\mathbf{V}} \left\{ \left| \|\boldsymbol{h}_{\mathbf{V},\boldsymbol{x}}^{RF}\|_2 \cdot \|\boldsymbol{h}_{\mathbf{V},\boldsymbol{z}}^{RF}\|_2 - \frac{p_1}{2d} \right| \geq 2m\sqrt{2p_1 d} + 2m^2 d \right\} \leq \frac{2}{m^2}.$$

*Proof.* Define

$$Q_k^i := (\mathbf{X}_i^T \mathbf{V}[k])(\mathbf{X}_i^T \mathbf{V}[k]) 1_{\{\mathbf{X}_i^T \mathbf{V}[k] > 0\}}.$$

By Eq. (1), we have

$$\frac{1}{p_1} \|\boldsymbol{h}_{\mathbf{V},\mathbf{X}_i}^{RF}\|_2^2 = \frac{1}{p_1} (\boldsymbol{h}_{\mathbf{V},\mathbf{X}_i}^{RF})^T \cdot \boldsymbol{h}_{\mathbf{V},\mathbf{X}_i}^{RF} = \frac{1}{p_1} \sum_{k=1}^{p_1} Q_k^i.$$

Note that

$$|Q_k^i| \leq \|\mathbf{X}_i\|_2 \cdot \|\mathbf{V}[k]\|_2 \cdot \|\mathbf{X}_i\|_2 \cdot \|\mathbf{V}[k]\| = 1 \text{ (by Assumption 1)}.$$

Further, note that

$$\mathbb{E}_{\mathbf{V}}[Q_1^i] = \int_{\mathcal{S}^{d-1}} (\mathbf{X}_i^T \boldsymbol{v})(\mathbf{X}_i^T \boldsymbol{v}) 1_{\{\mathbf{X}_i^T \boldsymbol{v} > 0\}} d\lambda(\boldsymbol{v}) \text{ (by Eq. (1))}$$

$$= \frac{\sin 0 + \pi \cos 0}{2d\pi} \text{ (by Lemma 21)}$$

$$= \frac{1}{2d}.$$

By Lemma 14, we thus have

$$\Pr_{\mathbf{V}} \left\{ \left| \frac{1}{p_1} \|\boldsymbol{h}_{\mathbf{V},\mathbf{X}_i}^{RF}\|_2^2 - \frac{1}{2d} \right| \geq \frac{m}{\sqrt{p_1}} \right\} \leq \frac{1}{m^2}. \tag{53}$$

Notice that

$$\left| \|\boldsymbol{h}_{\mathbf{V},\mathbf{X}_i}^{RF}\|_2^2 - \frac{p_1}{2d} \right| = \left( \|\boldsymbol{h}_{\mathbf{V},\mathbf{X}_i}^{RF}\|_2 + \sqrt{\frac{p_1}{2d}} \right) \cdot \left| \|\boldsymbol{h}_{\mathbf{V},\mathbf{X}_i}^{RF}\|_2 - \sqrt{\frac{p_1}{2d}} \right| \geq \sqrt{\frac{p_1}{2d}} \cdot \left| \|\boldsymbol{h}_{\mathbf{V},\mathbf{X}_i}^{RF}\|_2 - \sqrt{\frac{p_1}{2d}} \right|. \tag{54}$$

Combining Eq. (53) and Eq. (54), we then have

$$\Pr_{\mathbf{V}} \left\{ \left| \|\boldsymbol{h}_{\mathbf{V},\mathbf{X}_i}^{RF}\|_2 - \sqrt{\frac{p_1}{2d}} \right| \geq m\sqrt{2d} \right\} \leq \Pr_{\mathbf{V}} \left\{ \left| \|\boldsymbol{h}_{\mathbf{V},\mathbf{X}_i}^{RF}\|_2^2 - \frac{p_1}{2d} \right| \geq m\sqrt{p_1} \right\} \leq \frac{1}{m^2}.$$

Finally, notice that

$$\left| \beta_{i,j} - \frac{p_1}{2d} \right|$$

$$= \left| \|\boldsymbol{h}_{\mathbf{V},\mathbf{X}_i}^{\mathrm{RF}}\|_2 \left( \|\boldsymbol{h}_{\mathbf{V},\mathbf{X}_j}^{\mathrm{RF}}\|_2 - \sqrt{\frac{p_1}{2d}} \right) + \|\boldsymbol{h}_{\mathbf{V},\mathbf{X}_j}^{\mathrm{RF}}\|_2 \left( \|\boldsymbol{h}_{\mathbf{V},\mathbf{X}_i}^{\mathrm{RF}}\|_2 - \sqrt{\frac{p_1}{2d}} \right) \right.$$

$$\left. - \left( \|\boldsymbol{h}_{\mathbf{V},\mathbf{X}_j}^{\mathrm{RF}}\|_2 - \sqrt{\frac{p_1}{2d}} \right) \left( \|\boldsymbol{h}_{\mathbf{V},\mathbf{X}_i}^{\mathrm{RF}}\|_2 - \sqrt{\frac{p_1}{2d}} \right) \right|$$

$$\leq \sqrt{p_1} \left| \|\boldsymbol{h}_{\mathbf{V},\mathbf{X}_j}^{\mathrm{RF}}\|_2 - \sqrt{\frac{p_1}{2d}} \right| + \sqrt{p_1} \left| \|\boldsymbol{h}_{\mathbf{V},\mathbf{X}_i}^{\mathrm{RF}}\|_2 - \sqrt{\frac{p_1}{2d}} \right|$$

$$+ \left| \|\boldsymbol{h}_{\mathbf{V},\mathbf{X}_j}^{\mathrm{RF}}\|_2 - \sqrt{\frac{p_1}{2d}} \right| \cdot \left| \|\boldsymbol{h}_{\mathbf{V},\mathbf{X}_i}^{\mathrm{RF}}\|_2 - \sqrt{\frac{p_1}{2d}} \right| \text{ (by the triangle inequality and Lemma 11).}$$

Thus, we have

$$\left\{ \left| \|\boldsymbol{h}_{\mathbf{V},\mathbf{X}_i}^{\mathrm{RF}}\|_2 - \sqrt{\frac{p_1}{2d}} \right| \geq m\sqrt{2d} \right\} \cup \left\{ \left| \|\boldsymbol{h}_{\mathbf{V},\mathbf{X}_j}^{\mathrm{RF}}\|_2 - \sqrt{\frac{p_1}{2d}} \right| \geq m\sqrt{2d} \right\}$$

$$\supseteq \left\{ \left| \beta_{i,j} - \frac{p_1}{2d} \right| \geq 2m\sqrt{2p_1 d} + 2m^2 d \right\}.$$

Applying the union bound, we thus have

$$\Pr_{\mathbf{V}} \left\{ \left| \beta_{i,j} - \frac{p_1}{2d} \right| \geq 2m\sqrt{2p_1 d} + 2m^2 d \right\} \leq \frac{2}{m^2}.$$

$\square$

Now we are ready to prove Lemma 31.

*Proof of Lemma 31.* Define three events as

$$\mathcal{J}_{1,i,j} := \left\{ \frac{2d}{p_1} (\boldsymbol{h}_{\mathbf{V},\mathbf{X}_i}^{\mathrm{RF}})^T \cdot \boldsymbol{h}_{\mathbf{V},\mathbf{X}_j}^{\mathrm{RF}} \geq 1 - 2C(n,d,q) + \frac{2dnq}{\sqrt{p_1}} \right\},$$

$$\mathcal{J}_{2,i,j} := \left\{ \beta_{i,j} \leq \frac{p_1}{2d} - 2nq\sqrt{2p_1 d} - 2n^2 q^2 d \right\},$$

$$\mathcal{J}_{3,i,j} := \left\{ \cos\theta_{i,j}^{\mathrm{RF}} \geq 1 - 2C(n,d,q) \right\}.$$

We take a few steps as follows to finish the proof.

**Step 1: estimate $\mathcal{J}_{1,i,j}$.**

Define

$$Q_k^{i,j} := (\mathbf{X}_i^T \mathbf{V}[k])(\mathbf{X}_j^T \mathbf{V}[k]) \mathbf{1}_{\{\mathbf{X}_i^T \mathbf{V}[k]>0, \ \mathbf{X}_j^T \mathbf{V}[k]>0\}}, \quad k = 1, 2, \cdots, p_1.$$

By Eq. (1) and the definition of $Q_k^{i,j}$, we have

$$(\boldsymbol{h}_{\mathbf{V},\mathbf{X}_i}^{\mathrm{RF}})^T \cdot \boldsymbol{h}_{\mathbf{V},\mathbf{X}_j}^{\mathrm{RF}} = \sum_{k=1}^{p_1} Q_k^{i,j}. \tag{55}$$

Note that

$$|Q_k^{i,j}| \leq \|\mathbf{X}_i\|_2 \cdot \|\mathbf{V}[k]\|_2 \cdot \|\mathbf{X}_j\|_2 \cdot \|\mathbf{V}[k]\|_2 = 1 \text{ (by Assumption 1 and Lemma 12).}$$

By Lemma 14, we then have

$$\Pr_{\mathbf{V}} \left\{ \left| \frac{1}{p_1} \sum_{k=1}^{p_1} Q_k^{i,j} - \mathbb{E}_{\mathbf{V}}[Q_1^{i,j}] \right| \geq \frac{m}{\sqrt{p_1}} \right\} \leq \frac{1}{m^2}.$$

Let $\theta_{i,j} = \arccos(\mathbf{X}_i^T \mathbf{X}_j) \in [0, \pi]$ denote the angle between $\mathbf{X}_i$ and $\mathbf{X}_j$, where $i \neq j$ and $i, j \in \{1, 2, \cdots, n\}$. Notice that

$$\underset{\mathbf{V}}{\mathbb{E}}[Q_1^{i,j}] = \int_{\mathcal{S}^{d-1}} (\mathbf{X}_i^T \boldsymbol{v})(\mathbf{X}_j^T \boldsymbol{v}) 1_{\{\mathbf{X}_i^T \boldsymbol{v} > 0, \mathbf{X}_j^T \boldsymbol{v} > 0\}} d\lambda(\boldsymbol{v}) \text{ (by Eq. (1))}$$

$$= \frac{\sin \theta_{i,j} + (\pi - \theta_{i,j}) \cos \theta_{i,j}}{2d\pi} \text{ (by Lemma 21)}$$

$$\leq \frac{1}{2d} \left( 1 - \frac{\pi - 1}{2\pi} \sin^2 \theta_{i,j} \right) \text{ (by Lemma 34)}.$$

Thus, we have

$$\underset{\mathbf{V}}{\Pr} \left\{ \frac{1}{p_1} \sum_{k=1}^{p_1} Q_k^{i,j} \geq \frac{1}{2d} \left( 1 - \frac{\pi - 1}{2\pi} \sin^2 \theta_{i,j} \right) + \frac{m}{\sqrt{p_1}} \right\} \leq \frac{1}{m^2}.$$

For any $\alpha \in [0, 1]$, we have

$$\underset{\mathbf{X}}{\Pr} \left\{ \sin^2 \theta_{i,j} \leq \alpha \right\}$$

$$= \underset{\mathbf{X}}{\Pr} \left\{ \theta_{i,j} \leq \arcsin\left(\sqrt{\alpha}\right) \text{ OR } \pi - \theta_{i,j} \leq \arcsin\left(\sqrt{\alpha}\right) \right\}$$

$$\leq \underset{\mathbf{X}}{\Pr} \left\{ \theta_{i,j} \leq \arcsin\left(\sqrt{\alpha}\right) \right\} + \underset{\mathbf{X}}{\Pr} \left\{ \pi - \theta_{i,j} \leq \arcsin\left(\sqrt{\alpha}\right) \right\} \text{ (by the union bound)}$$

$$= I_\alpha \left( \frac{d-1}{2}, \frac{1}{2} \right) \text{ (area of two caps, by Lemma 8 and Assumption 1)}$$

$$\leq \frac{2\sqrt{d}\alpha^{\frac{d-1}{2}}}{(d-1)\sqrt{1-\alpha}} \text{ (by Lemma 9 and Lemma 10)}.$$

Further, because

$$\left\{ \sin^2 \theta_{i,j} > \alpha \right\} \cap \left\{ \frac{1}{p_1} \sum_{k=1}^{p_1} Q_k^{i,j} < \frac{1}{2d} \left( 1 - \frac{\pi - 1}{2\pi} \sin^2 \theta_{i,j} \right) + \frac{m}{\sqrt{p_1}} \right\}$$

$$\subseteq \left\{ \frac{2d}{p_1} \sum_{k=1}^{p_1} Q_k^{i,j} < 1 - \frac{\pi - 1}{2\pi}\alpha + \frac{2dm}{\sqrt{p_1}} \right\},$$

we have

$$\left\{ \sin^2 \theta_{i,j} \leq \alpha \right\} \cup \left\{ \frac{1}{p_1} \sum_{k=1}^{p_1} Q_k^{i,j} \geq \frac{1}{2d} \left( 1 - \frac{\pi - 1}{2\pi} \sin^2 \theta_{i,j} \right) + \frac{m}{\sqrt{p_1}} \right\}$$

$$\supseteq \left\{ \frac{2d}{p_1} \sum_{k=1}^{p_1} Q_k^{i,j} \geq 1 - \frac{\pi - 1}{2\pi}\alpha + \frac{2dm}{\sqrt{p_1}} \right\}.$$

Thus, by the union bound and Eq. (55), we have

$$\underset{\mathbf{V}, \mathbf{X}}{\Pr} \left\{ \frac{2d}{p_1} (\boldsymbol{h}_{\mathbf{V}, \mathbf{X}_i}^{\text{RF}})^T \cdot \boldsymbol{h}_{\mathbf{V}, \mathbf{X}_j}^{\text{RF}} \geq 1 - \frac{\pi - 1}{2\pi}\alpha + \frac{2dm}{\sqrt{p_1}} \right\} \leq \frac{1}{m^2} + \frac{2\sqrt{d}\alpha^{\frac{d-1}{2}}}{(d-1)\sqrt{1-\alpha}}. \quad (56)$$

By letting

$$\alpha = \min \left\{ \frac{1}{2}, \left( \frac{(d-1)^2}{8d} \right)^{\frac{1}{d-1}} (qn)^{-\frac{4}{d-1}} \right\}, \text{ and } m = qn,$$

we have

$$\frac{2\sqrt{d}\alpha^{\frac{d-1}{2}}}{(d-1)\sqrt{1-\alpha}} \leq \frac{2\sqrt{2}\sqrt{d}\alpha^{\frac{d-1}{2}}}{(d-1)} \text{ (because } \alpha \leq \frac{1}{2})$$

$$\leq \frac{2\sqrt{2}\sqrt{d}\sqrt{\frac{(d-1)^2}{8d} \frac{1}{q^2n^2}}}{d-1} \text{ (because } \alpha \leq \left( \frac{(d-1)^2}{8d} \right)^{\frac{1}{d-1}} (qn)^{-\frac{4}{d-1}})$$

$$= \frac{1}{q^2n^2}.$$

Thus, by Eq. (56), we have

$$\Pr_{\mathbf{V},\mathbf{X}}[\mathcal{J}_{1,i,j}] \le \frac{2}{q^2 n^2}. \tag{57}$$

**Step 2: estimate $\mathcal{J}_{2,i,j}$.** By Lemma 37, we have

$$\Pr_{\mathbf{V}}\left\{\beta_{i,j} \le \frac{p_1}{2d} - 2m\sqrt{2p_1 d} - 2m^2 d\right\} \le \frac{2}{m^2}.$$

Letting $m = qn$, we then have

$$\Pr_{\mathbf{V}}[\mathcal{J}_{2,i,j}] \le \frac{2}{q^2 n^2}. \tag{58}$$

**Step 3: prove $\mathcal{J}_{3,i,j} \subseteq \mathcal{J}_{1,i,j} \cup \mathcal{J}_{2,i,j}$.**

In order to show $\mathcal{J}_{3,i,j} \subseteq \mathcal{J}_{1,i,j} \cup \mathcal{J}_{2,i,j}$, it suffices to show $\mathcal{J}_{3,i,j}^c \supseteq \mathcal{J}_{1,i,j}^c \cap \mathcal{J}_{2,i,j}^c$. When $\mathcal{J}_{1,i,j}^c \cap \mathcal{J}_{2,i,j}^c$ happens, we have

$$\frac{2d}{p_1}\beta_{i,j} > 1 - \frac{2d}{p_1} \cdot 2nq\sqrt{2p_1 d} - \frac{2d}{p_1} \cdot 2n^2 q^2 d = 1 - \frac{4dnq\sqrt{2d}}{\sqrt{p_1}} - \frac{4d^2 n^2 q^2}{p_1},$$

$$\frac{2d}{p_1}(\boldsymbol{h}_{\mathbf{V},\mathbf{X}_i}^{\mathrm{RF}})^T \cdot \boldsymbol{h}_{\mathbf{V},\mathbf{X}_j}^{\mathrm{RF}} < 1 - 2C(n,d,q) + \frac{2dnq}{\sqrt{p_1}}.$$

Thus, we have

$$\begin{aligned}
\cos\theta_{i,j}^{\mathrm{RF}} &= \frac{(\boldsymbol{h}_{\mathbf{V},\mathbf{X}_i}^{\mathrm{RF}})^T \cdot \boldsymbol{h}_{\mathbf{V},\mathbf{X}_j}^{\mathrm{RF}}}{\beta_{i,j}} \\
&< \frac{1 - 2C(n,d,q) + \frac{2dnq}{\sqrt{p_1}}}{1 - \frac{4dnq\sqrt{2d}}{\sqrt{p_1}} - \frac{4d^2 n^2 q^2}{p_1}} \\
&\le 1 - C(n,d,q) \quad \text{(by Lemma 36)}
\end{aligned}$$

i.e., the event $\mathcal{J}_{3,i,j}^c$ happens. To sum up, we have proven that $\mathcal{J}_{3,i,j}^c \supseteq \mathcal{J}_{1,i,j}^c \cap \mathcal{J}_{2,i,j}^c$, which implies $\mathcal{J}_{3,i,j} \subseteq \mathcal{J}_{1,i,j} \cup \mathcal{J}_{2,i,j}$.

**Step 4: estimate $\mathcal{J}_{3,i,j}$.** We have

$$\begin{aligned}
\Pr_{\mathbf{V},\mathbf{X}}[\mathcal{J}_{3,i,j}] &\le \Pr_{\mathbf{V},\mathbf{X}}[\mathcal{J}_{1,i,j}] + \Pr_{\mathbf{V},\mathbf{X}}[\mathcal{J}_{2,i,j}] \quad \text{(by } \mathcal{J}_{3,i,j} \subseteq \mathcal{J}_{1,i,j} \cup \mathcal{J}_{2,i,j} \text{ and the union bound)} \\
&\le \frac{4}{q^2 n^2} \quad \text{(by Eq. (57) and Eq. (58)).}
\end{aligned} \tag{59}$$

**Step 5: estimate $\cos\theta_{\min}^{\mathrm{RF}}$.** We have

$$\begin{aligned}
&\Pr_{\mathbf{V},\mathbf{X}}\left\{\cos\theta_{\min}^{\mathrm{RF}} \ge 1 - C(n,d,q)\right\} \\
&= \Pr_{\mathbf{V},\mathbf{X}}\left[\bigcup_{i \ne j}\mathcal{J}_{3,i,j}\right] \\
&\le n(n-1)\Pr_{\mathbf{V},\mathbf{X}}[\mathcal{J}_{3,i,j}] \quad \text{(by the union bound)} \\
&\le \frac{4}{q^2} \quad \text{(by Eq. (59)).}
\end{aligned}$$

The result of Lemma 31 thus follows. $\qquad\square$

## H.3 Proof of Lemma 32

We first introduce two useful lemmas. Define $\mathbf{H}^\infty \in \mathbb{R}^{n \times n}$ as

$$\mathbf{H}_{i,j}^\infty := \beta_{i,j}\cos(\theta_{i,j}^{\mathrm{RF}}) \cdot \frac{\pi - \theta_{i,j}^{\mathrm{RF}}}{2\pi}.$$

**Lemma 38.** *Given* $\mathbf{X}$ *and* $\mathbf{V}$, *for any* $q > 0$, *we have*

$$\Pr_{\mathbf{W}_0} \left\{ \max_{i,j} \left| \frac{1}{p_2} (\mathbf{H}\mathbf{H}^T)_{i,j} - \mathbf{H}_{i,j}^\infty \right| \geq \frac{qnp_1}{\sqrt{p_2}} \right\} \leq \frac{1}{q^2}.$$

*Thus, we also have*

$$\Pr_{\mathbf{W}_0, \mathbf{X}, \mathbf{V}} \left\{ \max_{i,j} \left| \frac{1}{p_2} (\mathbf{H}\mathbf{H}^T)_{i,j} - \mathbf{H}_{i,j}^\infty \right| \geq \frac{qnp_1}{\sqrt{p_2}} \right\} \leq \frac{1}{q^2}.$$

*Proof.* For notation simplicity, given any $i, j \in \{1, 2, \cdots, n\}$, we define

$$Q_k^{i,j} := (\boldsymbol{h}_{\mathbf{V}, \mathbf{X}_i}^{\mathrm{RF}})^T \boldsymbol{h}_{\mathbf{V}, \mathbf{X}_j}^{\mathrm{RF}} \mathbb{1}_{\{(\boldsymbol{h}_{\mathbf{V}, \mathbf{X}_i}^{\mathrm{RF}})^T \mathbf{W}_0[k] > 0, \ (\boldsymbol{h}_{\mathbf{V}, \mathbf{X}_j}^{\mathrm{RF}})^T \mathbf{W}_0[k]\}} \text{ for all } k \in \{1, 2, \cdots, p_2\}.$$

By Eq. (2), we thus have

$$(\mathbf{H}\mathbf{H}^T)_{i,j} = (\boldsymbol{h}_{\mathbf{V}, \mathbf{W}_0, \mathbf{X}_i}^{\mathrm{Three}})^T \boldsymbol{h}_{\mathbf{V}, \mathbf{W}_0, \mathbf{X}_j}^{\mathrm{Three}} = \sum_{k=1}^{p_2} (\boldsymbol{h}_{\mathbf{V}, \mathbf{X}_i}^{\mathrm{RF}})^T \boldsymbol{h}_{\mathbf{V}, \mathbf{X}_j}^{\mathrm{RF}} \mathbb{1}_{\{(\boldsymbol{h}_{\mathbf{V}, \mathbf{X}_i}^{\mathrm{RF}})^T \mathbf{W}_0[k] > 0, \ (\boldsymbol{h}_{\mathbf{V}, \mathbf{X}_j}^{\mathrm{RF}})^T \mathbf{W}_0[k]\}}$$

$$= \sum_{k=1}^{p_2} Q_k^{i,j}.$$

By Lemma 20 and recalling Eq. (42), we have

$$\mathbb{E}_{\mathbf{W}_0}[Q_k^{i,j}] = \mathbf{H}_{i,j}^\infty.$$

By Lemma 11 and Lemma 12, we have

$$|Q_k^{i,j}| \leq \|\boldsymbol{h}_{\mathbf{V}, \mathbf{X}_i}^{\mathrm{RF}}\|_2 \cdot \|\boldsymbol{h}_{\mathbf{V}, \mathbf{X}_j}^{\mathrm{RF}}\|_2 \leq p_1.$$

Note that $Q_k^{i,j}$ are independent across $k$. By Lemma 14, for any $m > 0$, we have

$$\Pr_{\mathbf{W}_0} \left\{ \left| \frac{1}{p_2} (\mathbf{H}\mathbf{H}^T)_{i,j} - \mathbf{H}_{i,j}^\infty \right| \geq m \frac{p_1}{\sqrt{p_2}} \right\} \leq \frac{1}{m^2}. \tag{60}$$

The result of this lemma thus follows by letting $m = qn$ and the union bound, i.e.,

$$\Pr_{\mathbf{W}_0} \left\{ \max_{i,j} \left| \frac{1}{p_2} (\mathbf{H}\mathbf{H}^T)_{i,j} - \mathbf{H}_{i,j}^\infty \right| \geq \frac{qnp_1}{\sqrt{p_2}} \right\} = \Pr_{\mathbf{W}_0} \left\{ \bigcup_{i,j} \left\{ \left| \frac{1}{p_2} (\mathbf{H}\mathbf{H}^T)_{i,j} - \mathbf{H}_{i,j}^\infty \right| \geq \frac{qnp_1}{\sqrt{p_2}} \right\} \right\}$$

$$\leq \sum_{i,j} \Pr_{\mathbf{W}_0} \left\{ \left| \frac{1}{p_2} (\mathbf{H}\mathbf{H}^T)_{i,j} - \mathbf{H}_{i,j}^\infty \right| \geq \frac{qnp_1}{\sqrt{p_2}} \right\}$$

(by the union bound)

$$\leq \sum_{i,j} \frac{1}{q^2 n^2} \text{ (by letting } m = qn \text{ in Eq. (60))}$$

$$= \frac{1}{q^2}.$$

$\square$

**Lemma 39.** *Given* $\mathbf{X}$, *for any* $q > 0$, *we must have*

$$\Pr_{\mathbf{V}} \left\{ \max_{i,j} \left| \mathbf{H}_{i,j}^\infty - \tilde{\mathbf{H}}_{i,j}^\infty \right| \geq qn\sqrt{2p_1 d} + q^2 n^2 d \right\} \leq \frac{2}{q^2}.$$

*Thus, we also have*

$$\Pr_{\mathbf{V}, \mathbf{X}, \mathbf{W}_0} \left\{ \max_{i,j} \left| \mathbf{H}_{i,j}^\infty - \tilde{\mathbf{H}}_{i,j}^\infty \right| \geq qn\sqrt{2p_1 d} + q^2 n^2 d \right\} \leq \frac{2}{q^2}.$$

*Proof.* We have

$$\left|\mathbf{H}_{i,j}^{\infty} - \tilde{\mathbf{H}}_{i,j}^{\infty}\right| = \left|\left(\beta_{i,j} - \frac{p_1}{2d}\right)\cos(\theta_{i,j}^{\mathrm{RF}}) \cdot \frac{\pi - \theta_{i,j}^{\mathrm{RF}}}{2\pi}\right| \quad \text{(by Eq. (42))}$$

$$\leq \left|\beta_{i,j} - \frac{p_1}{2d}\right| \cdot \left|\cos(\theta_{i,j}^{\mathrm{RF}})\right| \cdot \left|\frac{\pi - \theta_{i,j}^{\mathrm{RF}}}{2\pi}\right| \quad \text{(by Lemma 12)}$$

$$\leq \frac{1}{2}\left|\beta_{i,j} - \frac{p_1}{2d}\right| \quad \text{(since } 0 \leq \theta_{i,j}^{\mathrm{RF}} \leq \frac{\pi}{2}\text{).}$$

The result of this lemma thus follows by letting $m = qn$ in Lemma 37 and the union bound, i.e.,

$$\Pr_{\mathbf{V}}\left\{\max_{i,j}\left|\mathbf{H}_{i,j}^{\infty} - \tilde{\mathbf{H}}_{i,j}^{\infty}\right| \geq qn\sqrt{2p_1 d} + q^2 n^2 d\right\}$$

$$\leq \Pr_{\mathbf{V}}\left\{\max_{i,j}\left|\beta_{i,j} - \frac{p_1}{2d}\right| \geq 2qn\sqrt{2p_1 d} + 2q^2 n^2 d\right\}$$

$$= \Pr_{\mathbf{V}}\left\{\bigcup_{i,j}\left\{\left|\beta_{i,j} - \frac{p_1}{2d}\right| \geq 2qn\sqrt{2p_1 d} + 2q^2 n^2 d\right\}\right\}$$

$$\leq \sum_{i,j}\Pr_{\mathbf{V}}\left\{\left|\beta_{i,j} - \frac{p_1}{2d}\right| \geq 2qn\sqrt{2p_1 d} + 2q^2 n^2 d\right\} \quad \text{(by the union bound)}$$

$$\leq \sum_{i,j}\frac{2}{q^2 n^2} \quad (\text{ by letting } m = qn \text{ in Lemma 37})$$

$$= \frac{2}{q^2}.$$

$\square$

Now we are ready to prove Lemma 32.

*Proof of Lemma 32.* By the triangle inequality, we have

$$\left|\frac{1}{p_2}(\mathbf{HH}^T)_{i,j} - \tilde{\mathbf{H}}_{i,j}^{\infty}\right| \leq \left|\mathbf{H}_{i,j}^{\infty} - \tilde{\mathbf{H}}_{i,j}^{\infty}\right| + \left|\frac{1}{p_2}(\mathbf{HH}^T)_{i,j} - \mathbf{H}_{i,j}^{\infty}\right|.$$

Thus, we have

$$\Pr_{\mathbf{X},\mathbf{V},\mathbf{W}_0}\left\{\max_{i,j}\left|\frac{1}{p_2}(\mathbf{HH}^T)_{i,j} - \tilde{\mathbf{H}}_{i,j}^{\infty}\right| \geq qn\sqrt{2p_1 d} + q^2 n^2 d + \frac{qnp_1}{\sqrt{p_2}}\right\}$$

$$\leq \Pr_{\mathbf{X},\mathbf{V},\mathbf{W}_0}\left\{\left\{\max_{i,j}\left|\mathbf{H}_{i,j}^{\infty} - \tilde{\mathbf{H}}_{i,j}^{\infty}\right| \geq qn\sqrt{2p_1 d} + q^2 n^2 d\right\}\right.$$

$$\left.\cup \left\{\max_{i,j}\left|\frac{1}{p_2}(\mathbf{HH}^T)_{i,j} - \mathbf{H}_{i,j}^{\infty}\right| \geq \frac{qnp_1}{\sqrt{p_2}}\right\}\right\}$$

$$\leq \Pr_{\mathbf{X},\mathbf{V},\mathbf{W}_0}\left\{\max_{i,j}\left|\mathbf{H}_{i,j}^{\infty} - \tilde{\mathbf{H}}_{i,j}^{\infty}\right| \geq qn\sqrt{2p_1 d} + q^2 n^2 d\right\}$$

$$+ \Pr_{\mathbf{X},\mathbf{V},\mathbf{W}_0}\left\{\max_{i,j}\left|\frac{1}{p_2}(\mathbf{HH}^T)_{i,j} - \mathbf{H}_{i,j}^{\infty}\right| \geq \frac{qnp_1}{\sqrt{p_2}}\right\} \quad \text{(by the union bound)}$$

$$\leq \frac{3}{q^2} \quad \text{(by Lemma 38 and Lemma 39).}$$

The result of Lemma 32 thus follows by Lemma 13 (where $k = n$). $\square$

# I  Proof of Proposition 5 and Proposition 7

We first provide some useful lemmas.

**Lemma 40.** *For any $\varphi \in [0, 2\pi]$, we must have $\sin \varphi \leq \varphi$. For any $\varphi \in [0, \pi/2]$, we must have $\varphi \leq \frac{\pi}{2} \sin \varphi$.*

*Proof.* See Lemma 41 of Ju et al. (2021). □

**Lemma 41.** *For any $a_1, a_2 \in [-1, 1]$ that $|a_1 - a_2| \leq 1$, we must have*

$$|\arccos(a_1) - \arccos(a_2)| \leq \frac{\sqrt{2}\pi}{2}\sqrt{|a_1 - a_2|}.$$

*Proof.* Without loss of generality, we assume $a_2 \geq a_1$ and let $\delta := a_2 - a_1 \in [0, 1]$. Because $\frac{\partial \arccos x}{\partial x} = -\frac{1}{\sqrt{1-x^2}}$, we have

$$\frac{\partial(\arccos(a_1) - \arccos(a_1 + \delta))}{\partial a_1} = -\frac{1}{\sqrt{1 - a_1^2}} + \frac{1}{\sqrt{1 - (a_1 + \delta)^2}}$$
$$\begin{cases} \leq 0, & \text{when } a_1 \in [-1, -\frac{\delta}{2}] \\ \geq 0, & \text{when } a_1 \in [-\frac{\delta}{2}, 1 - \delta] \end{cases}.$$

Thus, we know the largest value of $\arccos(a_1) - \arccos(a_1 + \delta)$ can only be achieved at either $a_1 = -1$ or $a_1 = 1 - \delta$, i.e.,

$$\arccos(a_1) - \arccos(a_1 + \delta) \leq \max\{\pi - \arccos(-1 + \delta), \arccos(1 - \delta)\} = \arccos(1 - \delta). \tag{61}$$

(The last equality is because $\arccos(-x) = \pi - \arccos x$.) It remains to show that $\arccos(1 - \delta) \leq \frac{\sqrt{2}\pi}{2}\sqrt{\delta}$. To that end, it suffices to prove $\cos(\frac{\sqrt{2}\pi}{2}\sqrt{\delta}) \leq 1 - \delta$. Let $\theta := \frac{\sqrt{2}\pi}{2}\sqrt{\delta}$, i.e., $\delta = \frac{2}{\pi^2}\theta^2$. When $\theta > \frac{\pi}{2}$, we have $\cos(\frac{\sqrt{2}\pi}{2}\sqrt{\delta}) = \cos\theta < 0 < 1 - \delta$ (since $\delta \in [0, 1]$). When $\theta \in [0, \frac{\pi}{2}]$, we have

$$\cos(\frac{\sqrt{2}\pi}{2}\sqrt{\delta}) = \cos\theta = \sqrt{1 - \sin^2\theta} \leq \sqrt{1 - \sin^2\theta + \frac{1}{4}\sin^4\theta} = 1 - \frac{1}{2}\sin^2\theta$$
$$\leq 1 - \frac{1}{2}(\frac{2}{\pi}\theta)^2 \text{ (by Lemma 40)}$$
$$= 1 - \delta.$$

Therefore, we have proven that $\arccos(1 - \delta) \leq \frac{\sqrt{2}\pi}{2}\sqrt{\delta}$ for all $\delta \in [0, 1]$. By Eq. (61), the result of this lemma thus follows. □

**Lemma 42.** *For any real number $a_1, a_2, \delta_1$, and $\delta_2$ such that $a_2 \in [-1, 1]$, $a_2 + \delta_2 \in [-1, 1]$, and $|\delta_2| \leq 1$, we must have*

$$\left|(a_1 + \delta_1)\frac{\pi - \arccos(a_2 + \delta_2)}{2\pi} - a_1\frac{\pi - \arccos(a_2)}{2\pi}\right| \leq \frac{1}{2}|\delta_1| + \frac{\sqrt{2}|a_1|\sqrt{|\delta_2|}}{4}.$$

*Proof.* Define

$$b := a_1\frac{\pi - \arccos(a_2 + \delta_2)}{2\pi}.$$

we have

$$\left| (a_1 + \delta_1) \frac{\pi - \arccos(a_2 + \delta_2)}{2\pi} - a_1 \frac{\pi - \arccos(a_2)}{2\pi} \right|$$

$$= \left| (a_1 + \delta_1) \frac{\pi - \arccos(a_2 + \delta_2)}{2\pi} - b + b - a_1 \frac{\pi - \arccos(a_2)}{2\pi} \right|$$

$$\leq \left| (a_1 + \delta_1) \frac{\pi - \arccos(a_2 + \delta_2)}{2\pi} - b \right| + \left| b - a_1 \frac{\pi - \arccos(a_2)}{2\pi} \right|$$

$$= |\delta_1| \cdot \left| \frac{\pi - \arccos(a_2 + \delta_2)}{2\pi} \right| + |a_1| \cdot \left| \frac{\arccos(a_2 + \delta_2) - \arccos(a_2)}{2\pi} \right|$$

$$\leq \frac{1}{2} |\delta_1| + \frac{\sqrt{2}|a_1|\sqrt{|\delta_2|}}{4} \text{ (since } \arccos(\cdot) \in [0, \pi] \text{ and by Lemma 41).}$$

$\square$

**Lemma 43.** *For any* $\theta \in [0, \pi]$, *we have*

$$\frac{\sin\theta + (\pi - \theta)\cos\theta}{\pi} \in [0, 1].$$

*Proof.* We have

$$\frac{\partial(\sin\theta + (\pi - \theta)\cos\theta)}{\partial\theta} = -(\pi - \theta)\sin\theta \leq 0.$$

Thus, $\sin\theta + (\pi - \theta)\cos\theta$ is monotone decreasing. The result of this lemma thus follows by plugging $\theta = 0$ and $\theta = \pi$ into the expression. $\square$

**Lemma 44.** *Recall the definition of* $K^{Three}(\cdot)$ *in Eq.* (6) *and the definition of* $Q(p_1, d)$ *in Eq.* (18). *When* $p_1$ *is large enough such that* $9d \cdot Q(p_1, d) \leq 1$, *we must have*

$$\Pr_{\mathbf{V}} \left\{ \max_{\boldsymbol{x}, \boldsymbol{z}} \left| \frac{1}{p_1} (\boldsymbol{h}_{\mathbf{V},\boldsymbol{z}}^{RF})^T \boldsymbol{h}_{\mathbf{V},\boldsymbol{x}}^{RF} \frac{\pi - \arccos\left( \frac{(\boldsymbol{h}_{\mathbf{V},\boldsymbol{z}}^{RF})^T \boldsymbol{h}_{\mathbf{V},\boldsymbol{x}}^{RF}}{\|\boldsymbol{h}_{\mathbf{V},\boldsymbol{z}}^{RF}\|_2 \cdot \|\boldsymbol{h}_{\mathbf{V},\boldsymbol{x}}^{RF}\|_2} \right)}{2\pi} - K^{Three}(\boldsymbol{x}^T \boldsymbol{z}) \right| \geq \sqrt{\frac{Q(p_1, d)}{d}} \right\}$$

$$\leq \frac{d^2}{(p_1 + 1)e^{d+1}}.$$

*Proof.* Because $9d \cdot Q(p_1, d) \leq 1$, we have

$$Q(p_1, d) = \sqrt{Q(p_1, d)}\sqrt{Q(p_1, d)} \leq \sqrt{Q(p_1, d)}\sqrt{\frac{1}{9d}} = \sqrt{\frac{Q(p_1, d)}{9d}}. \tag{62}$$

We also have

$$2d \cdot Q(p_1, d) \leq \frac{2}{9} \leq 0.5. \tag{63}$$

Define two events

$$\mathcal{J}_1 := \left\{ \max_{\boldsymbol{x}, \boldsymbol{z}} \left| \frac{1}{p_1} (\boldsymbol{h}_{\mathbf{V},\boldsymbol{x}}^{RF})^T \boldsymbol{h}_{\mathbf{V},\boldsymbol{z}}^{RF} - K^{RF}(\boldsymbol{x}^T \boldsymbol{z}) \right| \geq Q(p_1, d) \right\},$$

$$\mathcal{J}_2 := \left\{ \max_{\boldsymbol{x}, \boldsymbol{z}} \left| a - K^{Three}(\boldsymbol{x}^T \boldsymbol{z}) \right| \geq \sqrt{\frac{Q(p_1, d)}{d}} \right\}.$$

Notice that the randomness of those events is on $\mathbf{V}$. We first show $\mathcal{J}_1 \supseteq \mathcal{J}_2$, i.e., $\mathcal{J}_1^c \subseteq \mathcal{J}_2^c$. To that end, suppose $\mathcal{J}_1^c$ happens. Because of $\mathcal{J}_1^c$, we have

$$\max_{\boldsymbol{x}, \boldsymbol{z}} \left| \frac{2d}{p_1} (\boldsymbol{h}_{\mathbf{V},\boldsymbol{z}}^{RF})^T \boldsymbol{h}_{\mathbf{V},\boldsymbol{x}}^{RF} - 2d \cdot K^{RF}(\boldsymbol{x}^T \boldsymbol{z}) \right| \leq 2d \cdot Q(p_1, d). \tag{64}$$

By Eq. (4), we have

$$K^{\mathrm{RF}}(\boldsymbol{x}^T\boldsymbol{x}) = K^{\mathrm{RF}}(1) = \frac{1}{2d}. \tag{65}$$

Thus, we have

$$\max_{\boldsymbol{x},\boldsymbol{z}} \left| \frac{2d}{p_1} \left\| \boldsymbol{h}_{\mathbf{V},\boldsymbol{x}}^{\mathrm{RF}} \right\|_2 \cdot \left\| \boldsymbol{h}_{\mathbf{V},\boldsymbol{z}}^{\mathrm{RF}} \right\|_2 - 1 \right|$$

$$= \max_{\boldsymbol{x}} \left| \frac{2d}{p_1} \left\| \boldsymbol{h}_{\mathbf{V},\boldsymbol{x}}^{\mathrm{RF}} \right\|_2^2 - 1 \right| \text{ (the max value is achieved when } \|\boldsymbol{h}_{\mathbf{V},\boldsymbol{x}}^{\mathrm{RF}}\|_2 = \|\boldsymbol{h}_{\mathbf{V},\boldsymbol{z}}^{\mathrm{RF}}\|_2)$$

$$= \max_{\boldsymbol{x}} \left| \frac{2d}{p_1} \left\| \boldsymbol{h}_{\mathbf{V},\boldsymbol{x}}^{\mathrm{RF}} \right\|_2^2 - 2dK^{\mathrm{RF}}(\boldsymbol{x}^T\boldsymbol{x}) \right| \text{ (by Eq. (65))}$$

$$\leq \max_{\boldsymbol{x},\boldsymbol{z}} \left| \frac{2d}{p_1} \left\| \boldsymbol{h}_{\mathbf{V},\boldsymbol{x}}^{\mathrm{RF}} \right\|_2 \left\| \boldsymbol{h}_{\mathbf{V},\boldsymbol{z}}^{\mathrm{RF}} \right\|_2 - 2dK^{\mathrm{RF}}(\boldsymbol{x}^T\boldsymbol{z}) \right| \text{ (since we could set } \boldsymbol{z} = \boldsymbol{x} \text{ on the right hand side)}$$

$$\leq 2d \cdot Q(p_1, d) \text{ (because of } \mathcal{J}_1^c). \tag{66}$$

By Eq. (66), Eq. (64), and Eq. (63), we thus have

$$\left| \frac{2d}{p_1} \left\| \boldsymbol{h}_{\mathbf{V},\boldsymbol{x}}^{\mathrm{RF}} \right\|_2 \cdot \left\| \boldsymbol{h}_{\mathbf{V},\boldsymbol{z}}^{\mathrm{RF}} \right\|_2 - 1 \right| \leq 0.5 \text{ for all } \boldsymbol{x} \text{ and } \boldsymbol{z}. \tag{67}$$

Thus, we then have $\frac{2d}{p_1}(\boldsymbol{h}_{\mathbf{V},\boldsymbol{z}}^{\mathrm{RF}})^T\boldsymbol{h}_{\mathbf{V},\boldsymbol{x}}^{\mathrm{RF}} \leq \frac{2d}{p_1}\|\boldsymbol{h}_{\mathbf{V},\boldsymbol{x}}^{\mathrm{RF}}\|_2 \cdot \|\boldsymbol{h}_{\mathbf{V},\boldsymbol{z}}^{\mathrm{RF}}\|_2 \leq 1.5$. Besides, we have $\frac{2d}{p_1}(\boldsymbol{h}_{\mathbf{V},\boldsymbol{z}}^{\mathrm{RF}})^T\boldsymbol{h}_{\mathbf{V},\boldsymbol{x}}^{\mathrm{RF}} \geq 0$ because all elements of $\boldsymbol{h}_{\mathbf{V},\boldsymbol{x}}^{\mathrm{RF}}$ and $\boldsymbol{h}_{\mathbf{V},\boldsymbol{z}}^{\mathrm{RF}}$ are non-negative by Eq. (1). In other words, we have

$$\frac{2d}{p_1}(\boldsymbol{h}_{\mathbf{V},\boldsymbol{z}}^{\mathrm{RF}})^T\boldsymbol{h}_{\mathbf{V},\boldsymbol{x}}^{\mathrm{RF}} \in [0, 1.5] \text{ for all } \boldsymbol{x} \text{ and } \boldsymbol{z}. \tag{68}$$

Therefore, we then have

$$\max_{\boldsymbol{x},\boldsymbol{z}} \left| \frac{(\boldsymbol{h}_{\mathbf{V},\boldsymbol{z}}^{\mathrm{RF}})^T\boldsymbol{h}_{\mathbf{V},\boldsymbol{x}}^{\mathrm{RF}}}{\|\boldsymbol{h}_{\mathbf{V},\boldsymbol{z}}^{\mathrm{RF}}\|_2 \cdot \|\boldsymbol{h}_{\mathbf{V},\boldsymbol{x}}^{\mathrm{RF}}\|_2} - 2d \cdot K^{\mathrm{RF}}(\boldsymbol{x},\boldsymbol{z}) \right|$$

$$= \max_{\boldsymbol{x},\boldsymbol{z}} \left| \frac{\frac{2d}{p_1}(\boldsymbol{h}_{\mathbf{V},\boldsymbol{z}}^{\mathrm{RF}})^T\boldsymbol{h}_{\mathbf{V},\boldsymbol{x}}^{\mathrm{RF}}}{\frac{2d}{p_1}\|\boldsymbol{h}_{\mathbf{V},\boldsymbol{z}}^{\mathrm{RF}}\|_2 \cdot \|\boldsymbol{h}_{\mathbf{V},\boldsymbol{x}}^{\mathrm{RF}}\|_2} - 2d \cdot K^{\mathrm{RF}}(\boldsymbol{x},\boldsymbol{z}) \right|$$

$$\leq \max_{\boldsymbol{x},\boldsymbol{z}} \left| \frac{2d}{p_1}(\boldsymbol{h}_{\mathbf{V},\boldsymbol{z}}^{\mathrm{RF}})^T\boldsymbol{h}_{\mathbf{V},\boldsymbol{x}}^{\mathrm{RF}} - 2d \cdot K^{\mathrm{RF}}(\boldsymbol{x},\boldsymbol{z}) \right| + 3\max_{\boldsymbol{x},\boldsymbol{z}} \left| \frac{2d}{p_1}\|\boldsymbol{h}_{\mathbf{V},\boldsymbol{x}}^{\mathrm{RF}}\|_2 \cdot \|\boldsymbol{h}_{\mathbf{V},\boldsymbol{z}}^{\mathrm{RF}}\|_2 - 1 \right|$$

(by Lemma 35(ii) where $a = \frac{2d}{p_1}(\boldsymbol{h}_{\mathbf{V},\boldsymbol{z}}^{\mathrm{RF}})^T\boldsymbol{h}_{\mathbf{V},\boldsymbol{x}}^{\mathrm{RF}} \in [0, 1.5]$ by Eq. (68),

$b = \frac{2d}{p_1}\|\boldsymbol{h}_{\mathbf{V},\boldsymbol{x}}^{\mathrm{RF}}\|_2 \cdot \|\boldsymbol{h}_{\mathbf{V},\boldsymbol{z}}^{\mathrm{RF}}\|_2$, and $\delta = \left| \frac{2d}{p_1}\|\boldsymbol{h}_{\mathbf{V},\boldsymbol{x}}^{\mathrm{RF}}\|_2 \cdot \|\boldsymbol{h}_{\mathbf{V},\boldsymbol{z}}^{\mathrm{RF}}\|_2 - 1 \right| \in [0, 0.5]$ by Eq. (67)).

$$\leq 9d \cdot Q(p_1, d) \text{ (by Eq. (64) and Eq. (66)).} \tag{69}$$

Now we apply Lemma 42 by letting $\delta_1 = \frac{1}{p_1}(\boldsymbol{h}_{\mathbf{V},\boldsymbol{x}}^{\mathrm{RF}})^T\boldsymbol{h}_{\mathbf{V},\boldsymbol{z}}^{\mathrm{RF}} - K^{\mathrm{RF}}(\boldsymbol{x}^T\boldsymbol{z})$, $\delta_2 = \frac{(\boldsymbol{h}_{\mathbf{V},\boldsymbol{z}}^{\mathrm{RF}})^T\boldsymbol{h}_{\mathbf{V},\boldsymbol{x}}^{\mathrm{RF}}}{\|\boldsymbol{h}_{\mathbf{V},\boldsymbol{z}}^{\mathrm{RF}}\|_2 \cdot \|\boldsymbol{h}_{\mathbf{V},\boldsymbol{x}}^{\mathrm{RF}}\|_2} - 2d \cdot K^{\mathrm{RF}}(\boldsymbol{x}^T\boldsymbol{z})$, $a_1 = K^{\mathrm{RF}}(\boldsymbol{x}^T\boldsymbol{z})$, and $a_2 = 2d \cdot K^{\mathrm{RF}}(\boldsymbol{x}^T\boldsymbol{z})$. We first check the conditions required by Lemma 42. By Eq. (4) and Lemma 43, we have

$$a_2 = 2d \cdot K^{\mathrm{RF}}(\boldsymbol{x}^T\boldsymbol{z}) \in [0, 1] \subseteq [-1, 1].$$

Because $\left| (\boldsymbol{h}_{\mathbf{V},\boldsymbol{z}}^{\mathrm{RF}})^T\boldsymbol{h}_{\mathbf{V},\boldsymbol{x}}^{\mathrm{RF}} \right| \leq \|\boldsymbol{h}_{\mathbf{V},\boldsymbol{z}}^{\mathrm{RF}}\|_2 \cdot \|\boldsymbol{h}_{\mathbf{V},\boldsymbol{x}}^{\mathrm{RF}}\|_2$, we have

$$a_2 + \delta_2 = \frac{(\boldsymbol{h}_{\mathbf{V},\boldsymbol{z}}^{\mathrm{RF}})^T\boldsymbol{h}_{\mathbf{V},\boldsymbol{x}}^{\mathrm{RF}}}{\|\boldsymbol{h}_{\mathbf{V},\boldsymbol{z}}^{\mathrm{RF}}\|_2 \cdot \|\boldsymbol{h}_{\mathbf{V},\boldsymbol{x}}^{\mathrm{RF}}\|_2} \in [-1, 1].$$

By Eq. (69) and $9d \cdot Q(p_1, d) \leq 1$ (the condition of this lemma), we have

$$|\delta_2| \leq 9d \cdot Q(p_1, d) \leq 1.$$

Therefore, all conditions of Lemma 42 are satisfied. According to Lemma 42, we then have

$$\left| (a_1 + \delta_1) \frac{\pi - \arccos(a_2 + \delta_2)}{2\pi} - a_1 \frac{\pi - \arccos(a_2)}{2\pi} \right| \leq \frac{1}{2}|\delta_1| + \frac{\sqrt{2}|a_1|\sqrt{|\delta_2|}}{4}$$

$$\implies \left| \frac{1}{p_1}(\boldsymbol{h}_{\mathbf{V},z}^{\mathrm{RF}})^T \boldsymbol{h}_{\mathbf{V},x}^{\mathrm{RF}} \frac{\pi - \arccos\left( \frac{(\boldsymbol{h}_{\mathbf{V},z}^{\mathrm{RF}})^T \boldsymbol{h}_{\mathbf{V},x}^{\mathrm{RF}}}{\|\boldsymbol{h}_{\mathbf{V},z}^{\mathrm{RF}}\|_2 \cdot \|\boldsymbol{h}_{\mathbf{V},x}^{\mathrm{RF}}\|_2} \right)}{2\pi} \right.$$

$$\left. - K^{\mathrm{RF}}(\boldsymbol{x}^T\boldsymbol{z}) \frac{\pi - \arccos(2d \cdot K^{\mathrm{RF}}(\boldsymbol{x}^T\boldsymbol{z}))}{2\pi} \right| \leq \frac{1}{2} \left| \frac{1}{p_1}(\boldsymbol{h}_{\mathbf{V},x}^{\mathrm{RF}})^T \boldsymbol{h}_{\mathbf{V},z}^{\mathrm{RF}} - K^{\mathrm{RF}}(\boldsymbol{x}^T\boldsymbol{z}) \right|$$

$$+ \frac{\sqrt{2}|K^{\mathrm{RF}}(\boldsymbol{x}^T\boldsymbol{z})|\sqrt{\left| \frac{(\boldsymbol{h}_{\mathbf{V},z}^{\mathrm{RF}})^T \boldsymbol{h}_{\mathbf{V},x}^{\mathrm{RF}}}{\|\boldsymbol{h}_{\mathbf{V},z}^{\mathrm{RF}}\|_2 \cdot \|\boldsymbol{h}_{\mathbf{V},x}^{\mathrm{RF}}\|_2} - 2d \cdot K^{\mathrm{RF}}(\boldsymbol{x}^T\boldsymbol{z}) \right|}}{4}.$$

By $\mathcal{J}_1^c$ and Eq. (69), we thus have

$$\max_{\boldsymbol{x},\boldsymbol{z}} \left| \frac{1}{p_1}(\boldsymbol{h}_{\mathbf{V},z}^{\mathrm{RF}})^T \boldsymbol{h}_{\mathbf{V},x}^{\mathrm{RF}} \frac{\pi - \arccos\left( \frac{(\boldsymbol{h}_{\mathbf{V},z}^{\mathrm{RF}})^T \boldsymbol{h}_{\mathbf{V},x}^{\mathrm{RF}}}{\|\boldsymbol{h}_{\mathbf{V},z}^{\mathrm{RF}}\|_2 \cdot \|\boldsymbol{h}_{\mathbf{V},x}^{\mathrm{RF}}\|_2} \right)}{2\pi} - K^{\mathrm{Three}}(\boldsymbol{x}^T\boldsymbol{z}) \right|$$

$$\leq \frac{Q(p_1,d)}{2} + \frac{\sqrt{2}|K^{\mathrm{RF}}(\boldsymbol{x},\boldsymbol{z})|\sqrt{9d \cdot Q(p_1,d)}}{4}$$

$$\leq \frac{Q(p_1,d)}{2} + \sqrt{\frac{9}{32d} \cdot Q(p_1,d)} \text{ (because } K^{\mathrm{RF}}(\boldsymbol{x},\boldsymbol{z}) \in \left[0, \frac{1}{2d}\right] \text{ by Lemma 43)}$$

$$\leq \left( \sqrt{\frac{1}{36d}} + \sqrt{\frac{9}{32d}} \right) \sqrt{Q(p_1,d)} \text{ (by Eq. (62))}$$

$$\leq \sqrt{\frac{Q(p_1,d)}{d}} \text{ (since } \sqrt{\frac{1}{36}} + \sqrt{\frac{9}{32}} \approx \frac{1}{6} + 0.53 \leq 1\text{),}$$

i.e., $\mathcal{J}_2^c$ happens. We next estimate the probability of $\mathcal{J}_2$. We have

$$\Pr_{\mathbf{V}}[\mathcal{J}_2] \leq \Pr_{\mathbf{V}}[\mathcal{J}_1] \text{ (because } \mathcal{J}_2 \subseteq \mathcal{J}_1\text{)}$$

$$\leq \frac{d^2}{(p_1 + 1)e^{d+1}} \text{ (by Lemma 25, noticing that } 9d \cdot Q(p_1,d) \leq 1 \implies p_1 \geq 10\text{).}$$

The result of this lemma thus follows. $\qquad\square$

**Lemma 45.** *We have*

$$\|\mathbf{H}^T(\mathbf{H}\mathbf{H}^T)^{-1}\|_2 \leq \frac{1}{\sqrt{\min \mathrm{eig}(\mathbf{H}\mathbf{H}^T)}}.$$

*Proof.* For any $\boldsymbol{a} \in \mathbb{R}^n$, we have

$$\|\mathbf{H}^T(\mathbf{H}\mathbf{H}^T)^{-1}\boldsymbol{a}\|_2 = \sqrt{(\mathbf{H}^T(\mathbf{H}\mathbf{H}^T)^{-1}\boldsymbol{a})^T \mathbf{H}^T(\mathbf{H}\mathbf{H}^T)^{-1}\boldsymbol{a}} = \sqrt{\boldsymbol{a}^T(\mathbf{H}\mathbf{H}^T)^{-1}\boldsymbol{a}}$$

$$\leq \frac{\|\boldsymbol{a}\|_2}{\sqrt{\min \mathrm{eig}(\mathbf{H}\mathbf{H}^T)}}.$$

The result of this lemma thus follows. $\qquad\square$

We are now ready to prove Proposition 5 and Proposition 7.

## I.1 Proof of Proposition 5

*Proof.* For $k = 1, 2, \cdots, p_2$, define

$$K_k = \int_{\mathcal{S}^{d-1}} \frac{(\boldsymbol{h}_{\mathbf{V},\boldsymbol{x}}^{\mathrm{RF}})^T \boldsymbol{h}_{\mathbf{V},\boldsymbol{z}}^{\mathrm{RF}}}{p_1} \mathbb{1}_{\{(\boldsymbol{h}_{\mathbf{V},\boldsymbol{x}}^{\mathrm{RF}})^T \mathbf{W}_0[k] > 0, \ (\boldsymbol{h}_{\mathbf{V},\boldsymbol{z}}^{\mathrm{RF}})^T \mathbf{W}_0[k]\}} g(\boldsymbol{z}) d\mu(\boldsymbol{z}). \tag{70}$$

It is obvious that $K_1, K_2, \cdots, K_{p_2}$ are *i.i.d.* (when randomness is on $\mathbf{W}_0$). By Eq. (19) and Eq. (2), we have

$$f_{\mathbf{V},\mathbf{W}_0}^g(\boldsymbol{x}) = \frac{1}{p_2} \sum_{k=1}^{p_2} K_k. \tag{71}$$

Notice that

$$
\begin{aligned}
|K_k| &\leq \int_{\mathcal{S}^{d-1}} \left| \frac{(\boldsymbol{h}_{\mathbf{V},\boldsymbol{x}}^{\mathrm{RF}})^T \boldsymbol{h}_{\mathbf{V},\boldsymbol{z}}^{\mathrm{RF}}}{p_1} \right| \cdot |g(\boldsymbol{z})| \, d\mu(\boldsymbol{z}) \\
&\leq \int_{\mathcal{S}^{d-1}} |g(\boldsymbol{z})| \, d\mu(\boldsymbol{z}) \text{ (by Lemma 11)} \\
&= \|g\|_1.
\end{aligned}
\tag{72}
$$

Thus, by Lemma 14, we have

$$\Pr_{\mathbf{W}_0} \left\{ \left| \frac{1}{p_2} \sum_{k=1}^{p_2} K_k - \mathop{\mathsf{E}}_{\mathbf{W}_0}[K_1] \right| \geq \frac{q\|g\|_1}{\sqrt{p_2}} \right\} \leq \frac{1}{q^2}. \tag{73}$$

For any $k \in \{1, 2, \cdots, p_2\}$, we have

$$
\begin{aligned}
&\mathop{\mathsf{E}}_{\mathbf{W}_0}[K_k] \\
&= \int_{\mathcal{S}^{d-1}} \mathop{\mathsf{E}}_{\mathbf{W}_0} \left[ \frac{(\boldsymbol{h}_{\mathbf{V},\boldsymbol{x}}^{\mathrm{RF}})^T \boldsymbol{h}_{\mathbf{V},\boldsymbol{z}}^{\mathrm{RF}}}{p_1} \mathbb{1}_{\{(\boldsymbol{h}_{\mathbf{V},\boldsymbol{x}}^{\mathrm{RF}})^T \mathbf{W}_0[k] > 0, \ (\boldsymbol{h}_{\mathbf{V},\boldsymbol{z}}^{\mathrm{RF}})^T \mathbf{W}_0[k]\}} \right] g(\boldsymbol{z}) d\mu(\boldsymbol{z}) \\
&= \int_{\mathcal{S}^{d-1}} \frac{(\boldsymbol{h}_{\mathbf{V},\boldsymbol{x}}^{\mathrm{RF}})^T \boldsymbol{h}_{\mathbf{V},\boldsymbol{z}}^{\mathrm{RF}}}{p_1} \cdot \frac{\pi - \arccos\left( \frac{(\boldsymbol{h}_{\mathbf{V},\boldsymbol{x}}^{\mathrm{RF}})^T \boldsymbol{h}_{\mathbf{V},\boldsymbol{z}}^{\mathrm{RF}}}{\|\boldsymbol{h}_{\mathbf{V},\boldsymbol{x}}^{\mathrm{RF}}\|_2 \cdot \|\boldsymbol{h}_{\mathbf{V},\boldsymbol{z}}^{\mathrm{RF}}\|_2} \right)}{2\pi} g(\boldsymbol{z}) d\mu(\boldsymbol{z}) \text{ (by Lemma 20)} \\
&= f(\boldsymbol{x}) + \int_{\mathcal{S}^{d-1}} \left( \frac{(\boldsymbol{h}_{\mathbf{V},\boldsymbol{x}}^{\mathrm{RF}})^T \boldsymbol{h}_{\mathbf{V},\boldsymbol{z}}^{\mathrm{RF}}}{p_1} \cdot \frac{\pi - \arccos\left( \frac{(\boldsymbol{h}_{\mathbf{V},\boldsymbol{x}}^{\mathrm{RF}})^T \boldsymbol{h}_{\mathbf{V},\boldsymbol{z}}^{\mathrm{RF}}}{\|\boldsymbol{h}_{\mathbf{V},\boldsymbol{x}}^{\mathrm{RF}}\|_2 \cdot \|\boldsymbol{h}_{\mathbf{V},\boldsymbol{z}}^{\mathrm{RF}}\|_2} \right)}{2\pi} - K^{\mathrm{Three}}(\boldsymbol{x}^T \boldsymbol{z}) \right) g(\boldsymbol{z}) d\mu(\boldsymbol{z})
\end{aligned}
$$

(by $f = f_g$ and Eq. (7)).

Thus, we have

$$
\begin{aligned}
&\left| \mathop{\mathsf{E}}_{\mathbf{W}_0}[K_k] - f(\boldsymbol{x}) \right| \\
&\leq \max_{\boldsymbol{x},\boldsymbol{z}} \left| \frac{(\boldsymbol{h}_{\mathbf{V},\boldsymbol{x}}^{\mathrm{RF}})^T \boldsymbol{h}_{\mathbf{V},\boldsymbol{z}}^{\mathrm{RF}}}{p_1} \cdot \frac{\pi - \arccos\left( \frac{(\boldsymbol{h}_{\mathbf{V},\boldsymbol{x}}^{\mathrm{RF}})^T \boldsymbol{h}_{\mathbf{V},\boldsymbol{z}}^{\mathrm{RF}}}{\|\boldsymbol{h}_{\mathbf{V},\boldsymbol{x}}^{\mathrm{RF}}\|_2 \cdot \|\boldsymbol{h}_{\mathbf{V},\boldsymbol{z}}^{\mathrm{RF}}\|_2} \right)}{2\pi} - K^{\mathrm{Three}}(\boldsymbol{x}^T \boldsymbol{z}) \right| \cdot \|g\|_1. \tag{74}
\end{aligned}
$$

Applying Lemma 44, we then have

$$\Pr_{\mathbf{V}} \left\{ \left| \mathop{\mathsf{E}}_{\mathbf{W}_0}[K_k] - f(\boldsymbol{x}) \right| \geq \sqrt{\frac{Q(p_1, d)}{d}} \|g\|_1 \right\} \leq \frac{d^2}{(p_1 + 1)e^{d+1}}. \tag{75}$$

Notice that

$$\left| f_{\mathbf{V},\mathbf{W}_0}^g(\boldsymbol{x}) - f(\boldsymbol{x}) \right| = \left| \frac{1}{p_2} \sum_{k=1}^{p_2} K_k - f(\boldsymbol{x}) \right| \text{ (by Eq. (71))}$$

$$\leq \left| \frac{1}{p_2} \sum_{k=1}^{p_2} K_k - \underset{\mathbf{W}_0}{\mathsf{E}}[K_1] \right| + \left| \underset{\mathbf{W}_0}{\mathsf{E}}[K_1] - f(\boldsymbol{x}) \right| \text{ (by the triangle inequality).}$$

Combining Eq. (73) and Eq. (75) by the union bound, we thus have

$$\underset{\mathbf{V},\mathbf{W}_0}{\mathsf{Pr}} \left\{ \left| f_{\mathbf{V},\mathbf{W}_0}^g(\boldsymbol{x}) - f(\boldsymbol{x}) \right| \geq \frac{q\|g\|_1}{\sqrt{p_2}} + \sqrt{\frac{Q(p_1, d)}{d}} \|g\|_1 \right\}$$

$$\leq \underset{\mathbf{V},\mathbf{W}_0}{\mathsf{Pr}} \left\{ \left| \frac{1}{p_2} \sum_{k=1}^{p_2} K_k - \underset{\mathbf{W}_0}{\mathsf{E}}[K_1] \right| \geq \frac{q\|g\|_1}{\sqrt{p_2}} \right\} + \underset{\mathbf{V},\mathbf{W}_0}{\mathsf{Pr}} \left\{ \left| \underset{\mathbf{W}_0}{\mathsf{E}}[K_k] - f(\boldsymbol{x}) \right| \geq \sqrt{\frac{Q(p_1, d)}{d}} \|g\|_1 \right\}$$

$$\leq \frac{d^2}{(p_1 + 1)e^{d+1}} + \frac{1}{q^2}. \tag{76}$$

□

## I.2 Proof of Proposition 7

*Proof.* For $k = 1, 2, \cdot, p_2$, define $\mathbf{K}_k \in \mathbb{R}^n$ whose $i$-th element is

$$\mathbf{K}_{k,i} := \int_{\mathcal{S}^{d-1}} \frac{(\boldsymbol{h}_{\mathbf{V},\mathbf{X}_i}^{\mathrm{RF}})^T \boldsymbol{h}_{\mathbf{V},\boldsymbol{z}}^{\mathrm{RF}}}{p_1} \mathbf{1}_{\{(\boldsymbol{h}_{\mathbf{V},\mathbf{X}_i}^{\mathrm{RF}})^T \mathbf{W}_0[k]>0,\ (\boldsymbol{h}_{\mathbf{V},\boldsymbol{z}}^{\mathrm{RF}})^T \mathbf{W}_0[k]\}} g(\boldsymbol{z}) d\mu(\boldsymbol{z}).$$

Note that $\mathbf{K}_{k,i}$ is similar to $K_k$ in Eq. (70), with the only difference that the former is defined with respect to $\mathbf{X}_i$ and the latter is defined with respect to $\boldsymbol{x}$. Thus, we use a similar strategy to work with $\mathbf{K}_{k,i}$. By Eq. (20) and Eq. (19), we have

$$\mathbf{F}_{\mathbf{V},\mathbf{W}_0}^g(\mathbf{X}) = \frac{1}{p_2} \sum_{k=1}^{p_2} \mathbf{K}_k.$$

Similar to Eq. (72), we have

$$|\mathbf{K}_{k,i}| \leq \|g\|_1, \text{ for all } i = 1, 2, \cdots, n.$$

Thus, we have

$$\|\mathbf{K}_k\|_2 = \sqrt{\sum_{i=1}^n \mathbf{K}_{k,i}^2} \leq \sqrt{n} \|g\|_1.$$

By Lemma 14, we thus have

$$\underset{\mathbf{W}_0}{\mathsf{Pr}} \left\{ \left\| \frac{1}{p_2} \sum_{k=1}^{p_2} \mathbf{K}_k - \underset{\mathbf{W}_0}{\mathsf{E}}[\mathbf{K}_1] \right\|_2 \geq \frac{q\sqrt{n}\|g\|_1}{\sqrt{p_2}} \right\} \leq \frac{1}{q^2}.$$

Similar to Eq. (74), we have

$$\left\| \underset{\mathbf{W}_0}{\mathsf{E}}[\mathbf{K}_1] - \mathbf{F}(\mathbf{X}) \right\|_2$$

$$\leq \sqrt{n} \max_{\boldsymbol{x},\boldsymbol{z}} \left| \frac{(\boldsymbol{h}_{\mathbf{V},\boldsymbol{x}}^{\mathrm{RF}})^T \boldsymbol{h}_{\mathbf{V},\boldsymbol{z}}^{\mathrm{RF}}}{p_1} \cdot \frac{\pi - \arccos \left( \frac{(\boldsymbol{h}_{\mathbf{V},\boldsymbol{x}}^{\mathrm{RF}})^T \boldsymbol{h}_{\mathbf{V},\boldsymbol{z}}^{\mathrm{RF}}}{\|\boldsymbol{h}_{\mathbf{V},\boldsymbol{x}}^{\mathrm{RF}}\|_2 \cdot \|\boldsymbol{h}_{\mathbf{V},\boldsymbol{z}}^{\mathrm{RF}}\|_2} \right)}{2\pi} - K^{\mathrm{Three}}(\boldsymbol{x},\boldsymbol{z}) \right| \cdot \|g\|_1.$$

Thus, similar to Eq. (76), we have

$$\underset{\mathbf{V},\mathbf{W}_0}{\mathsf{Pr}} \left\{ \left\| \mathbf{F}(\mathbf{X}) - \mathbf{F}_{\mathbf{V},\mathbf{W}_0}^g(\mathbf{X}) \right\|_2 \geq \frac{q\sqrt{n}\|g\|_1}{\sqrt{p_2}} + \sqrt{\frac{Q(p_1, d)}{d}} \sqrt{n}\|g\|_1 \right\} \leq \frac{d^2}{(p_1 + 1)e^{d+1}} + \frac{1}{q^2}. \tag{77}$$

We note that

term D + term B of Eq. (22)

$$\leq \left\| \boldsymbol{h}_{\mathbf{V},\mathbf{W}_0,\boldsymbol{x}}^{\text{Three}} \right\|_2 \left\| \mathbf{H}^T(\mathbf{H}\mathbf{H}^T)^{-1} \right\|_2 \cdot \left( \|\mathbf{F}(\mathbf{X}) - \mathbf{F}_{\mathbf{V},\mathbf{W}_0}^g(\mathbf{X})\|_2 + \|\boldsymbol{\epsilon}\|_2 \right) \text{ (by Lemma 12)}$$

$$\leq \frac{\sqrt{p_1 p_2} \cdot \left( \|\mathbf{F}(\mathbf{X}) - \mathbf{F}_{\mathbf{V},\mathbf{W}_0}^g(\mathbf{X})\|_2 + \|\boldsymbol{\epsilon}\|_2 \right)}{\sqrt{\min \text{eig}(\mathbf{H}\mathbf{H}^T)}} \text{ (by Lemma 45 and Lemma 11)}.$$

Combining Eq. (77) and Proposition 6 by the union bound, we thus have

$$\Pr_{\mathbf{X},\mathbf{V},\mathbf{W}_0} \left\{ \text{term D + term B of Eq. (22)} \geq \frac{\sqrt{n}\|g\|_1 \left( \frac{q}{\sqrt{p_2}} + \sqrt{\frac{Q(p_1,d)}{d}} \right) + \|\boldsymbol{\epsilon}\|_2}{\sqrt{J(n,p_1,p_2,d,q)}} \right\}$$

$$\leq \frac{d^2}{(p_1+1)e^{d+1}} + \frac{8}{q^2}.$$

$\square$

## J   Details Related to Learnable Set

In this part, we first restate Proposition 2 in a more precise way, i.e., Proposition 46 in Appendix J.1 and Proposition 47 in Appendix J.2. Then, in Appendix J.3 we discuss the generalization performance of ground-truth functions outside the learnable set.

### J.1   $\mathcal{F}_{(3)}^{\ell_2}$ contains all polynomials with finite degree

By the following proposition, we show that $\mathcal{F}_{(3)}^{\ell_2}$ contains all polynomials with finite degree. We formally state it in the following proposition.

**Proposition 46.** *Let $k$ be a finite non-negative integer. For any $f(\boldsymbol{x}) = \sum_{i=0}^k c_i(\boldsymbol{x}^T\boldsymbol{a}_i)^i$ where $c_i \in \mathbb{R}$ and $\boldsymbol{a}_i \in \mathbb{R}^d$, we must have $f \in \mathcal{F}_{(3)}^{\ell_2}$.*

We prove Proposition 46 in Appendix K. Although Proposition 46 is only for no-bias situation of 3-layer NTK, we can easily prove the similar results for the biased 3-layer NTK with the same proof technique.

### J.2   $\mathcal{F}_{(3)}^{\ell_2}$ is a superset of $\mathcal{F}_{(2),b}^{\ell_2}$ (recall the definition of $\mathcal{F}_{(2),b}^{\ell_2}$ in Section 4.2)

The learnable sets of both 3-layer and 2-layer NTK models also contain polynomials with *infinite* degree. Notice that not all infinite-degree polynomials belong to the learnable sets, because the norm of the corresponding function $g$ may not be finite. As we mentioned in footnote 4, the constrain $\|g\|_\infty < \infty$ can be relaxed to $\|g\|_1 < \infty$. However, with $\|g\|_1 < \infty$, the comparison among those learnable sets becomes more difficult. For convenience, we just relax the constraint to $\|g\|_2 < \infty$ (instead of $\|g\|_1 < \infty$) in the following result.

**Proposition 47.** *Under the constraint of $\|g\|_2 < \infty$, the learnable set of the 3-layer NTK (no bias) is at least as large as the 2-layer NTK (both with and without bias) ,i.e., $\mathcal{F}_{(2)}^{\ell_2} \cup \mathcal{F}_{(2),b}^{\ell_2} \subseteq \mathcal{F}_{(3)}^{\ell_2}$. The learnable set of 2-layer NTK with bias is larger than that of 2-layer NTK without bias i.e., $\mathcal{F}_{(2)}^{\ell_2} \subset \mathcal{F}_{(2),b}^{\ell_2}$. The learnable sets of 2-layer NTK with different bias settings are the same i.e., $\mathcal{F}_{(2),b_1}^{\ell_2} = \mathcal{F}_{(2),b_2}^{\ell_2}$ for any $b_1, b_2 \in (0,1)$.*

We prove Proposition 47 in Appendix L. An important message conveyed by Proposition 47 is that, 3-layer NTK can at least learn all learnable functions for 2-layer NTK under the constraint $\|g\|_2 < \infty$. We conjecture that the same result may also hold for $\|g\|_1 < \infty$, which we leave for future work.

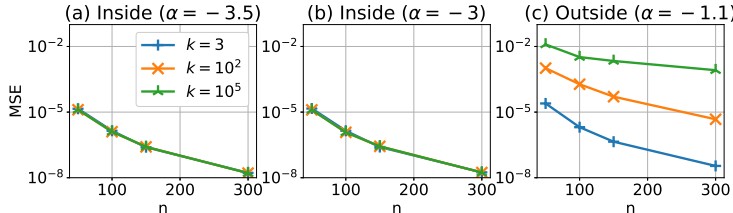

Figure 9: Curves of test MSE of 2-layer NTK with normal bias with respect to $n$ for the ground-truth functions $f_{k,\alpha}(\boldsymbol{x})$ where $p_1, p_2 \to \infty$, $d = 2$, and $\boldsymbol{\epsilon} = \boldsymbol{0}$. Every curve is the average of 10 simulation runs.

### J.3 Generalization performance of ground-truth functions outside the learnable set

One may wonder what happens to the generalization performance for functions outside the learnable set. Notice that although we have proven that ground-truth functions inside the learnable set can be learned, it is possible that some functions outside the learnable set could still be learnable. For 2-layer NTK models without bias, Ju et al. (2021) shows that if a ground-truth function has a positive distance away from the learnable set, then such distance becomes the lower bound of the generalization error. Such ground-truth functions with positive distance exist for 2-layer NTK, e.g., $(\boldsymbol{x}^T \boldsymbol{e}_1)^3$, because $\mathcal{F}^{\ell_2}_{(2)}$ does not contain odd power polynomials except linear functions. However, for 2-layer NTK with bias or 3-layer NTK, there do not exist such ground-truth functions with a positive distance away from the learnable set. In other words, functions outside the learnable set is still in the closure of the corresponding learnable set. Thus, it is unclear whether or not those functions have a very different generalization performance compared with functions inside the learnable set.

We now use simulation results in Fig. 9 to show that functions outside the learnable set may indeed exhibit qualitatively different generalization performance (and thus Proposition 47 will be meaningful in capturing ground-truth functions with good generalization performance). We construct an example of functions inside and outside the learnable set (in the sense of finite $\|g\|_2$, consistent with Proposition 47). For simplicity, we focus on $\mathcal{F}^{\ell_2}_{(2),\text{NLB}}$, which is the learnable set for the 2-layer NTK with normal bias. We then consider a specific type of normalized ground-truth functions $f_{k,\alpha} := \bar{f}_{k,\alpha}/\|\bar{f}_{k,\alpha}\|_2$ where $\bar{f}_{k,\alpha}(\boldsymbol{x}) := \sum_{i=1}^{k} i^\alpha \left(\boldsymbol{x}^T \boldsymbol{e}_d\right)^i$. By previous discussion, we have already known that if $k$ is finite, then $f_{k,\alpha} \in \mathcal{F}^{\ell_2}_{(2),\text{NLB}}$. However, when $k = \infty$, then whether $f_{\infty,\alpha} \in \mathcal{F}^{\ell_2}_{(2),\text{NLB}}$ or not is determined by the value of $\alpha$. We let $d = 2$ and choose the value of $\alpha$ to be $-3.5$, $-3$, and $-1.1$, respectively. It can be verified that $f_{k,\alpha} \in \mathcal{F}^{\ell_2}_{(2),\text{NLB}}$ when $\alpha = -3.5$ or $\alpha = -3$, while $f_{k,\alpha} \notin \mathcal{F}^{\ell_2}_{(2),\text{NLB}}$ when $\alpha = -1.1$. In numerical experiments, it is difficult to directly calculate $f_{\infty,\alpha}$, as we do not know the close form of $f_{\infty,\alpha}$. Therefore, we use $f_{k,\alpha}$ to approach $f_{\infty,\alpha}$ by increasing $k$. In Fig. 9(a), we let $\alpha = -3.5$ and plot the test MSE with respect to $n$ when $k = 3$ (blue curve), $k = 10^2$ (orange curve), and $k = 10^5$ (green curve), respectively. We can see that these three curves almost overlap with each other, which implies that increasing $k$ does not alter the test error significantly. (Similar phenomenon also appears in Fig. 9(b) where $\alpha = -3$.) In contrast, when we let $\alpha = -1.1$ in Fig. 9(c), larger $k$ leads to a much flatter curve. This phenomenon suggests that when $k \to \infty$, providing more training data becomes less effective in lowering the test error. Besides, by comparing the curve of $k = 10^5$ in Fig. 9(a) and (c), we can see that the curve in Fig. 9(c) is higher than the one in Fig. 9(a) by several orders of magnitude. Therefore, we can tell that the functions inside and outside the learnable set could have very different generalization performance.

The setup of Fig. 10 is the same as that of Fig. 9 except that here we let x-axis be $k$. In Fig. 10, we can see that the curves of $\alpha = -3.5$ and $\alpha = -3$ (finite $\|g\|_2$) in all sub-figures (a)(b)(c)(d) are almost flat with respect to $k$. In contrast, the curves of $\alpha = -1.1$ (infinite $\|g\|_2$) keep increasing with respect to $k$, and have much higher generalization error when $k$ is large than those with finite $\|g\|_2$. This also validates our conjecture that the functions inside and outside the learnable set could have very different generalization performance.

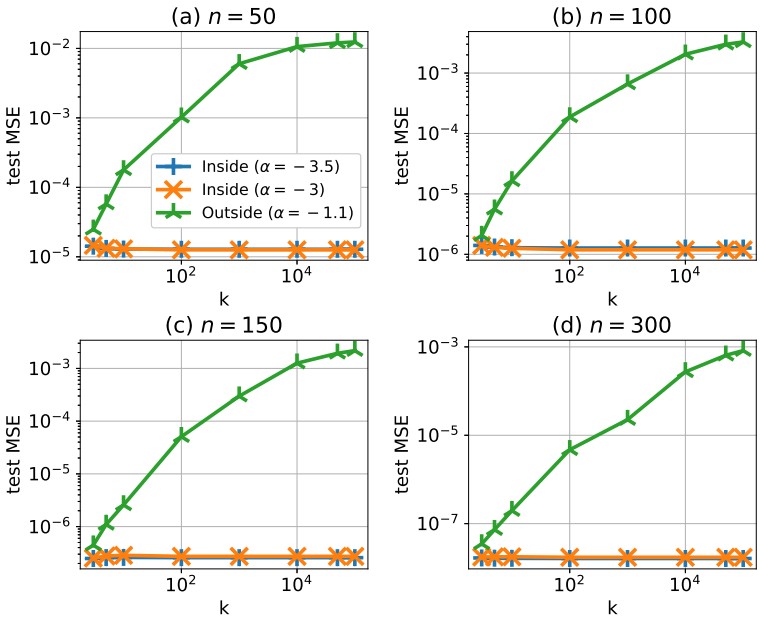

Figure 10: Curves of test MSE of 2-layer NTK with normal bias with respect to $k$ for the ground-truth functions $f_{k,\alpha}(\boldsymbol{x})$ where $p_1, p_2 \to \infty$, $d = 2$, and $\boldsymbol{\epsilon} = \boldsymbol{0}$. Every curve is the average of 10 simulation runs.

# K  Proof of Proposition 46

*Proof.* We prove Proposition 46 by using similar methods as in Ju et al. (2021). For any $f_g \in \mathcal{F}_{(3)}^{\ell_2}$, we have

$$f_g(\boldsymbol{x}) = g \circledast h^{(3)}(\boldsymbol{x}) := \int_{\mathsf{SO}(d)} g(\mathbf{S}\boldsymbol{e}_d) h^{(3)}(\mathbf{S}^{-1}\boldsymbol{x}) d\mathbf{S}, \tag{78}$$

$$h^{(3)}(\boldsymbol{x}) := K^{\mathrm{Three}}(\boldsymbol{x}^T \boldsymbol{e}_d), \tag{79}$$

where $\boldsymbol{e}_d := [0\ 0\ \cdots\ 0\ 1]^T \in \mathbb{R}^d$, and $\mathbf{S}$ is a $d \times d$ orthogonal matrix that denotes a rotation in $\mathcal{S}^{d-1}$, chosen from the set $\mathsf{SO}(d)$ of all rotations. An important property of the convolution Eq. (78) is that it corresponds to multiplication in the frequency domain, similar to Fourier coefficients. To define such a transformation to the frequency domain, we use a set of hyper-spherical harmonics $\Xi_{\mathbf{K}}^l$ (Vilenkin, 1968; Dokmanic & Petrinovic, 2009) when $d \geq 3$, which forms an orthonormal basis for functions on $\mathcal{S}^{d-1}$. These harmonics are indexed by $l$ and $\mathbf{K}$, where $\mathbf{K} = (k_1, k_2, \cdots, k_{d-2})$ and $l = k_0 \geq k_1 \geq k_2 \geq \cdots \geq k_{d-2} \geq 0$ (those $k_i$'s and $l$ are all non-negative integers). Any function $f \in L^2(\mathcal{S}^{d-1} \mapsto \mathbb{R})$ (including even $\delta$-functions (Li & Wong, 2013)) can be decomposed uniquely into these harmonics, i.e., $f(\boldsymbol{x}) = \sum_l \sum_{\mathbf{K}} c_f(l, \mathbf{K}) \Xi_{\mathbf{K}}^l(\boldsymbol{x})$, where $c_f(\cdot, \cdot)$ are projections of $f$ onto the basis function.

In Eq. (78), let $c_g(\cdot, \cdot)$ and $c_h(\cdot, \cdot)$ denote the coefficients corresponding to the decompositions of $g$ and $h$, respectively. Then, we must have (Dokmanic & Petrinovic, 2009)

$$c_{f_g}(l, \mathbf{K}) = \Lambda \cdot c_g(l, \mathbf{K}) c_{h^{(3)}}(l, \boldsymbol{0}), \tag{80}$$

where $\Lambda$ is some normalization constant.

Eq. (80) describes an interesting "filtering" interpretation on $\mathcal{F}_{(3)}^{\ell_2}$. Specifically, $h^{(3)}$ and $c_{h^{(3)}}$ work like a channel or a filter in a wireless communication system, where $c_g$ denotes the transmitted signal and $c_{f_g}$ denotes the received signal. Therefore, for any basis function $f(\boldsymbol{x}) = \Xi_{\mathbf{K}}^l(\boldsymbol{x})$, as long as $c_{h^{(3)}}(l, \boldsymbol{0}) \neq 0$, we must have $f = f_g \in \mathcal{F}_{(3)}^{\ell_2}$ where the corresponding $g(\cdot)$ can simply be chosen as $g(\boldsymbol{z}) = \frac{\Xi_{\mathbf{K}}^l(\boldsymbol{z})}{\Lambda c_{h^{(3)}}(l, \boldsymbol{0})}$. Indeed, we have the following proposition about values of $c_{h^{(3)}}(l, \boldsymbol{0})$.

**Proposition 48.** $c_{h^{(3)}}(l, \mathbf{0}) > 0$ *for all* $l = 0, 1, 2, \cdots$.

We provide its proof in Appendix K.1.

By Proposition 48, we know that all harmonics $\Xi_{\mathbf{K}}^l \in \mathcal{F}_{(3)}^{\ell_2}$. Notice that the set $\mathcal{F}_{(3)}^{\ell_2}$ is invariant under addition and scale operation[12]. Therefore, any finite sum of $\Xi_{\mathbf{K}}^l$ belongs to $\mathcal{F}_{(3)}^{\ell_2}$. Notice that for any non-negative integer $i$ and a real-valued vector $\mathbf{a} \in \mathbb{R}^d$, a polynomial $(\mathbf{x}^T \mathbf{a})^i$ consists of a finite sum of harmonic basis. Thus, $\mathcal{F}_{(3)}^{\ell_2}$ contains any polynomials $(\mathbf{x}^T \mathbf{a})^l$ for all $l = 0, 1, 2, \cdots$. Proposition 46 thus follows. $\qquad\square$

### K.1 Proof of Proposition 48

It is relatively easy to prove the result when $d = 2$, which is omitted here. We focus on the general case when $d \geq 3$. By Eq. (115) of Ju et al. (2021), the harmonics $\Xi_{\mathbf{0}}^l$ can be expressed by

$$\Xi_{\mathbf{0}}^l(\mathbf{x}) = A_{\mathbf{0}}^l \sum_{k=0}^{\lfloor \frac{l}{2} \rfloor} (-1)^k \frac{\Gamma(l - k + \frac{d-2}{2})}{\Gamma(\frac{d-2}{2}) k! (l - 2k)!} (2\mathbf{x}^T \mathbf{e}_d)^{l-2k}, \tag{81}$$

where $A_{\mathbf{0}}^l$ is a positive number as the normalization factor of $\Xi_{\mathbf{0}}^l$. We give a few examples of $\Xi_{\mathbf{0}}^l$ as follows.

$$\begin{aligned}
\Xi_{\mathbf{0}}^0(\mathbf{x}) &= A_{\mathbf{0}}^0, \\
\Xi_{\mathbf{0}}^1(\mathbf{x}) &= A_{\mathbf{0}}^1 (d - 2) \mathbf{x}^T \mathbf{e}_d, \\
\Xi_{\mathbf{0}}^2(\mathbf{x}) &= A_{\mathbf{0}}^2 \frac{d-2}{2} \left( d(\mathbf{x}^T \mathbf{e}_d)^2 - 1 \right), \\
\Xi_{\mathbf{0}}^3(\mathbf{x}) &= A_{\mathbf{0}}^3 \frac{d-2}{2} \cdot d \cdot \left( (\mathbf{x}^T \mathbf{e}_d)^3 - \mathbf{x}^T \mathbf{e}_d \right).
\end{aligned}$$

Recalling Eq. (6), we perform a Taylor expansion of $K^{\text{Three}}(\cdot)$. Let $u_0, u_1, \cdots$ denote the Taylor expansion coefficients of $2d \cdot K^{\text{Three}}$, i.e.,

$$2d \cdot K^{\text{Three}}(a) = \sum_{k=0}^{\infty} u_k a^k. \tag{82}$$

The following lemma shows that all coefficients in Eq. (82) are positive.

**Lemma 49.** *For all* $k = 0, 1, 2, \cdots$, *we have* $u_k > 0$ *in Eq.* (82).

*Proof.* By Lemma 17, for any $a, b \in [0, 1]$, we have

$$2d \cdot K^{\text{RF}}(a) = \frac{1}{\pi} \left( 1 + \frac{\pi}{2} a + \sum_{k=0}^{\infty} \frac{2(2k)!}{(k+1)(2k+1)(k!)^2} \left( \frac{a}{2} \right)^{2k+2} \right), \tag{83}$$

$$K^{\text{Two}}(b) = \frac{b}{4} + \frac{1}{2\pi} \sum_{k=0}^{\infty} \frac{(2k)!}{(k!)^2} \frac{4}{2k+1} \left( \frac{b}{2} \right)^{2k+2}. \tag{84}$$

By Lemma 43, we know that $2d \cdot K^{\text{RF}}(a) \in [0, 1]$. Thus, we can let $b = 2d \cdot K^{\text{RF}}(a)$ in Eq. (84) and then apply Eq. (83), i.e.,

$$K^{\text{Two}}(2d \cdot K^{\text{RF}}(a))$$

$$= \frac{1}{4\pi} \left( 1 + \frac{\pi}{2} a + \sum_{k=0}^{\infty} \frac{2(2k)!}{(k+1)(2k+1)(k!)^2} \left( \frac{a}{2} \right)^{2k+2} \right)$$

$$+ \frac{1}{2\pi} \sum_{l=0}^{\infty} \frac{(2l)!}{(l!)^2} \frac{4}{2l+1} \left( \frac{1}{2\pi} \left( 1 + \frac{\pi}{2} a + \sum_{k=0}^{\infty} \frac{2(2k)!}{(k+1)(2k+1)(k!)^2} \left( \frac{a}{2} \right)^{2k+2} \right) \right)^{2l+2}. \tag{85}$$

---

[12]Specifically, if $f_{g_1}, f_{g_2} \in \mathcal{F}_{(3)}^{\ell_2}$, then $f_{g_1+g_2} := f_{g_1} + f_{g_2} \in \mathcal{F}_{(3)}^{\ell_2}$ and $f_{\alpha g_1} := \alpha f_{g_1} \in \mathcal{F}_{(3)}^{\ell_2}$.

By Eq. (6) and Eq. (82), we know that $u_k$ is the coefficient of $a^k$ in Eq. (85). In order to know the sign of $u_k$, it remains to combine similar terms in Eq. (85). To that end, we apply Lemma 27 and have

$$u_0 = \frac{1}{4\pi} + \frac{1}{2\pi}\sum_{l=0}^{\infty}\frac{(2l)!}{(l!)^2}\frac{4}{2l+1}\left(\frac{1}{2\pi}\right)^{2l+2},$$

$$u_1 = \frac{1}{8} + \frac{1}{2\pi}\sum_{l=0}^{\infty}\frac{(2l)!}{(l!)^2}\frac{4}{2l+1}\left(\frac{1}{2\pi}\right)^{2l+2}(2l+2)\left(\frac{\pi}{2}\right)^{2l+1},$$

$$u_{2i+1} = \frac{1}{2\pi}\sum_{l=0}^{\infty}\frac{(2l)!}{(l!)^2}\frac{4}{2l+1}\left(\frac{1}{2\pi}\right)^{2l+2}\cdot\sum_{\substack{k_0+k_1+k_2+k_4+\cdots+k_{2i}=2l+2 \\ k_1+2k_2+4k_4+\cdots+2ik_{2i}=2i+1}}$$

$$(k_0,k_1,k_2,k_4,\cdots,k_{2i})!\left(\frac{\pi}{2}\right)^{k_1}\prod_{j=0}^{i-1}\left(\frac{2(2j)!}{(j+1)(2j+1)(j!)^2}\left(\frac{1}{2}\right)^{2j+2}\right)^{k_{2j+2}},$$

$$u_{2i+2} = \frac{1}{4\pi}\frac{2(2i)!}{(i+1)(2i+1)(i!)^2}\left(\frac{1}{2}\right)^{2i+2}$$

$$+ \frac{1}{2\pi}\sum_{l=0}^{\infty}\frac{(2l)!}{(l!)^2}\frac{4}{2l+1}\left(\frac{1}{2\pi}\right)^{2l+2}\cdot\sum_{\substack{k_0+k_1+k_2+k_4+\cdots+k_{2i}=2l+2 \\ k_1+2k_2+4k_4+\cdots+2ik_{2i}=2i+2}}$$

$$(k_0,k_1,k_2,k_4,\cdots,k_{2i})!\left(\frac{\pi}{2}\right)^{k_1}\prod_{j=0}^{i-1}\left(\frac{2(2j)!}{(j+1)(2j+1)(j!)^2}\left(\frac{1}{2}\right)^{2j+2}\right)^{k_{2j+2}}.$$

As we can see, every term in those expressions of $u_0, u_1, \cdots$ is positive, which implies that $u_k > 0$ for all $k = 0, 1, \cdots$. $\qquad\square$

From Eq. (82), we have $2d \cdot K^{\text{Three}}(\boldsymbol{x}^T\boldsymbol{e}_d) = \sum_{k=0}^{\infty}u_k(\boldsymbol{x}^T\boldsymbol{e}_d)^k$. We now consider the decomposition of each $(\boldsymbol{x}^T\boldsymbol{e}_d)^k$ into harmonics.

**Lemma 50.** *Let $a$ and $b$ be two non-negative integers. Define the function*

$$Q(a,b) := \int_{\mathcal{S}^{d-1}}(\boldsymbol{x}^T\boldsymbol{e}_d)^a \cdot \Xi_{\boldsymbol{0}}^b(\boldsymbol{x})d\mu(\boldsymbol{x}). \tag{86}$$

*We must have*

$$Q(2k, 2m+1) = Q(2k+1, 2m) = 0, \tag{87}$$

$$Q(2k, 2m)\begin{cases} > 0, & \text{if } m \le k, \\ = 0, & \text{if } m > k, \end{cases} \tag{88}$$

*and*

$$Q(2k+1, 2m+1)\begin{cases} > 0, & \text{if } m \le k, \\ = 0, & \text{if } m > k. \end{cases} \tag{89}$$

*Proof.* By Eq. (81), we have $\Xi_{\boldsymbol{0}}^b(-\boldsymbol{x}) = (-1)^b\Xi_{\boldsymbol{0}}^b(\boldsymbol{x})$. Thus, when $a + b$ is odd, the function $(\boldsymbol{x}^T\boldsymbol{e}_d)^a \cdot \Xi_{\boldsymbol{0}}^b(\boldsymbol{x})$ is an odd function with respect to $\boldsymbol{x}$. By symmetry of $\mathcal{S}^{d-1}$, we then have $Q(a,b) = 0$ when $a + b$ is odd, i.e., Eq. (87) holds. Eq. (88) has been proved in Lemma 53 of Ju et al. (2021) by mathematical induction. Here we prove Eq. (89).

By Eq. (118) of Ju et al. (2021), for any $a$, we have

$$Q(a+1, l+1) = q_{l,1} \cdot Q(a, l+2) + q_{l,2} \cdot Q(a, l), \tag{90}$$

where $q_{l,1} > 0$ and $q_{l,2} > 0$. Applying Eq. (90) for $a = 2k$ and $l = 2m$, we have

$$Q(2k+1, 2m+1) = q_{2m,1} \cdot Q(2k, 2m+2) + q_{2m,2} \cdot Q(2k, 2m).$$

By Eq. (88), the result of Eq. (89) thus follows. $\qquad\square$

Notice that

$$
\begin{aligned}
c_{h^{(3)}}(l, \mathbf{0}) &= \int_{\mathcal{S}^{d-1}} h^{(3)}(\boldsymbol{x}) \Xi_{\mathbf{0}}^{l}(\boldsymbol{x}) d\mu(\boldsymbol{x}) \\
&= \frac{1}{2d} \int_{\mathcal{S}^{d-1}} 2d \cdot K^{\text{Three}}(\boldsymbol{x}^{T} \boldsymbol{e}_d) \Xi_{\mathbf{0}}^{l}(\boldsymbol{x}) d\mu(\boldsymbol{x}) \text{ (by Eq. (79))} \\
&= \frac{1}{2d} \sum_{k=0}^{\infty} u_k \cdot Q(k, l) \text{ (by Eq. (82) and Eq. (86))} \\
&> 0 \text{ (by Lemma 49 and Lemma 50).}
\end{aligned}
$$

The result of Proposition 48 thus follows.

# L  Proof of Proposition 47

*Proof.* Using similar decomposition in Eq. (78), we define filter functions $h^{(2)}$ (for 2-layer NTK, no-bias) and $h_b^{(2)}$ (for 2-layer NTK, with bias). The corresponding harmonic coefficients are denoted by $c_{h^{(2)}}$ and $c_{h_b^{(2)}}$. We have the following result about the magnitude of those harmonic coefficients.

**Lemma 51.** *For any $b_1, b_2 \in (0, 1)$, we must have $c_{h_{b_1}^{(2)}}(2k, \mathbf{0}) = \Theta\left(c_{h^{(2)}}(2k, \mathbf{0})\right)$, $c_{h_{b_1}^{(2)}}(k, \mathbf{0}) = \Theta\left(c_{h_{b_2}^{(2)}}(k, \mathbf{0})\right)$, and $c_{h^{(3)}}(k, \mathbf{0}) = \Omega\left(c_{h_{b_1}^{(2)}}(k, \mathbf{0})\right)$. Here, $\Theta(\cdot)$ and $\Omega(\cdot)$ denote the orders as $k$ becomes large.*

We prove Lemma 51 in Appendix M.

Lemma 51 has the following implications for the magnitude of the harmonics coefficients when the leading index of harmonics is large (i.e., $k$ in Lemma 51 is large). The first statement states that, for 2-layer NTK, the setting with bias and the setting without bias have the same order of harmonics coefficients for even terms. (For odd terms, recall that for 2-layer NTK without bias, the coefficients of odd terms except linear term are zero. In contrast, for 2-layer NTK with bias, the coefficients of odd terms are not zero (Ju et al., 2021). Hence, the first statement does not hold for odd terms). The second statement states that, the coefficients of harmonics for 2-layer NTK have the same order with respect to $k$ for all non-zero bias. The third statement states that, the coefficients for 3-layer NTK even without bias is not smaller (in order) than 2-layer NTK with bias.

By comparing the magnitude of these filter coefficients, we can then compare whether polynomials with infinite degree belong to each of the learnable sets. Specifically, consider an infinite-degree polynomial with the form $f_g(\boldsymbol{x}) = \sum_{l, \mathbf{K}} \alpha_{l, \mathbf{K}} \cdot \Xi_{\mathbf{K}}^{l}(\boldsymbol{x})$. By Eq. (80), we have $g(\boldsymbol{z}) = \sum_{l, \mathbf{K}} \frac{\alpha_{l, \mathbf{K}} \cdot \Xi_{\mathbf{K}}^{l}(\boldsymbol{z})}{\Lambda \cdot c_h(l, \mathbf{0})}$, where $h$ can be $h^{(3)}$, $h^{(2)}$, or $h_b^{(2)}$. Thus, the magnitude of $c_h(l, \mathbf{0})$ determines the norm of $g$. Specifically, we have $\|g\|_2 = \sum_{l, \mathbf{K}} \frac{\alpha_{l, \mathbf{K}}}{\Lambda \cdot c_h(l, \mathbf{0})}$ due to the orthogonality of harmonics $\Xi_{\mathbf{K}}^{l}$. Note that $c_{h_{b_1}^{(2)}}(k, \mathbf{0}) = \Theta\left(c_{h_{b_2}^{(2)}}(k, \mathbf{0})\right)$ by Lemma 51. Thus, if $\|g\|_2$ is finite for 2-layer NTK with bias $b_1 > 0$, then it must also be finite for 2-layer NTK with a different bias $b_2 > 0$. This implies that The learnable sets of 2-layer NTK with different bias settings are the same i.e., $\mathcal{F}_{(2), b_1}^{\ell_2} = \mathcal{F}_{(2), b_2}^{\ell_2}$ for any $b_1, b_2 \in (0, 1)$. Similarly, for 3-layer NTK, by Lemma 51, we can also show that $\mathcal{F}_{(2)}^{\ell_2} \cup \mathcal{F}_{(2), b}^{\ell_2} \subseteq \mathcal{F}_{(3)}^{\ell_2}$ and $\mathcal{F}_{(2)}^{\ell_2} \subset \mathcal{F}_{(2), b}^{\ell_2}$. Therefore, the result of Proposition 47 follows. □

# M  Proof of Lemma 51

The following lemma shows the relationship between harmonic coefficients and Taylor coefficients.

**Lemma 52.** *Consider two polynomial functions $h_\alpha(\boldsymbol{x}) := \sum_{k=0}^{\infty} u_{\alpha, k} (\boldsymbol{x}^T \boldsymbol{e}_d)^k$ and $h_\beta(\boldsymbol{x}) := \sum_{k=0}^{\infty} u_{\beta, k} (\boldsymbol{x}^T \boldsymbol{e}_d)^k$ where $u_{\alpha, k} \geq 0$ and $u_{\beta, k} \geq 0$ for all $k$. Let $c_{h_\alpha}$ and $c_{h_\beta}$ denote their harmonic coefficients. If $u_{\alpha, k} = O(u_{\beta, k})$ (where $O(\cdot)$ denotes the order when $k$ is large), then $c_{h_\alpha}(l, \mathbf{0}) =$*

$O(c_{h_\beta}(l, \mathbf{0}))$ *for large $l$. The same is true if we restrict to only even harmonics, i.e., if $u_{\alpha,2k} = O(u_{\beta,2k})$, then $c_{h_\alpha}(2l, \mathbf{0}) = O(c_{h_\beta}(2l, \mathbf{0}))$ for large $l$. This lemma also holds if $O(\cdot)$ is replaced by $\Omega(\cdot)$ or $\Theta(\cdot)$.*

*Proof.* Notice that

$$c_{h_\alpha}(l, \mathbf{0}) = \int_{\mathcal{S}^{d-1}} h(\boldsymbol{x}) \Xi_{\mathbf{0}}^{l}(\boldsymbol{x}) d\mu(\boldsymbol{x})$$
$$= \sum_{k=0}^{\infty} u_{\alpha,k} \cdot Q(k, l) \text{ (by Eq. (86))}$$
$$= \sum_{k=l}^{\infty} u_{\alpha,k} \cdot Q(k, l) \text{ (by Lemma 50)}.$$

and

$$c_{h_\alpha}(2l, \mathbf{0}) = \int_{\mathcal{S}^{d-1}} h(\boldsymbol{x}) \Xi_{\mathbf{0}}^{2l}(\boldsymbol{x}) d\mu(\boldsymbol{x})$$
$$= \sum_{k=0}^{\infty} u_{\alpha,k} \cdot Q(k, 2l) \text{ (by Eq. (86))}$$
$$= \sum_{k=l}^{\infty} u_{\alpha,2k} \cdot Q(2k, 2l)$$

($Q(k, 2l)$ is non-zero only when $k$ is even and not smaller than $2l$ by Lemma 50).

Similarly, we have

$$c_{h_\beta}(l, \mathbf{0}) = \sum_{k=l}^{\infty} u_{\beta,k} \cdot Q(k, l),$$
$$c_{h_\beta}(2l, \mathbf{0}) = \sum_{k=l}^{\infty} u_{\beta,2k} \cdot Q(2k, 2l).$$

Notice that all $Q(\cdot, \cdot)$ and $u_{.,.}$ are all non-negative. Thus, we have

$$\frac{c_{h_\alpha}(l, \mathbf{0})}{c_{h_\beta}(l, \mathbf{0})} \in \left[ \min_{k \geq l} \frac{u_{\alpha,k}}{u_{\beta,k}}, \max_{k \geq l} \frac{u_{\alpha,k}}{u_{\beta,k}} \right], \qquad \frac{c_{h_\alpha}(2l, \mathbf{0})}{c_{h_\beta}(2l, \mathbf{0})} \in \left[ \min_{k \geq l} \frac{u_{\alpha,2k}}{u_{\beta,2k}}, \max_{k \geq l} \frac{u_{\alpha,2k}}{u_{\beta,2k}} \right].$$

The result of this lemma thus follows. $\qquad\square$

With Lemma 52, in order to show Lemma 51, it is equivalent to compare Taylor coefficients of the expression of different kernels $h^{(3)}$, $h^{(2)}$, and $h_b^{(2)}$. Specifically, we are looking at the Taylor coefficients of the following expression:

$$T(x) := K(x) \frac{\pi - \arccos(K(x))}{2\pi}. \tag{91}$$

For 2-layer NTK, $K(x) = (1 - a)x + a$ where $a \in [0, 1)$ corresponds to different choices of bias ($a = 0$ corresponds to no bias). For 3-layer NTK (no bias), we have $K(x) = 2d \cdot K^{\mathrm{RF}}(x)$ (we neglect the constant $1/(2d)$ in $K^{\mathrm{Three}}$, which does not change its order.) In other words, we have

$$T(x) = \begin{cases} K^{\mathrm{Two}}(x), & \text{if } K(x) = x \text{ (i.e., 2-layer NTK no bias)}, \\ K^{\mathrm{Two}}((1 - a) + a), & \text{if } K(x) = (1 - a)x + a \text{ (i.e., 2-layer NTK with bias } a > 0), \\ 2d \cdot K^{\mathrm{Three}}(x), & \text{if } K(x) = 2d \cdot K^{\mathrm{RF}}(x) \text{ (i.e., 3-layer NTK no bias)}. \end{cases}$$

When $K(x) = x$, we already have the exact form of the Taylor expansion of $T(x)$ by Lemma 17. However, when $K(x)$ is a polynomial, it is not easy to get the close form of Taylor coefficients. We will first estimate the Taylor coefficients when $K(x) = (1 - a)x + a$ in Appendix M.1. Second, we will estimate the Taylor coefficients when $K(x)$ is a polynomial with finite degree in Appendix M.2. Last, we will estimate the case of $K(x) = 2d \cdot K^{\mathrm{RF}}(x)$ in Appendix M.3. (By Lemma 17, we know that $K^{\mathrm{RF}}(x)$ is a polynomial with infinite degree.) The result of Lemma 51 then follows from these estimates.

## M.1 Harmonic coefficients for 2-layer NTK with bias

After adding bias, the kernel of 2-layer NTK changes from $x \frac{\pi - \arccos x}{2\pi}$ to $((1-a)x + a) \frac{\pi - \arccos((1-a)x+a)}{2\pi}$. Here $a > 0$ denotes the bias setting ($a = 0$ corresponds to the no-bias setting). By Lemma 52, we only need to investigate the relationship between the Taylor coefficients. We define $u_{a,m}$ as the Taylor coefficients under the bias setting, i.e.,

$$((1-a)x + a) \frac{\pi - \arccos((1-a)x+a)}{2\pi} = \sum_{m=0}^{\infty} u_{a,m} \cdot x^m.$$

When $a = 0$, $u_{a,m}$ becomes $u_{0,m}$ and corresponds to the no-bias setting.

**Lemma 53.** *For any $k \in \{2, 3, \cdots\}$ and any $a \in [0, 1)$, we must have $\frac{u_{a,2k}}{u_{0,2k}} \in$*

$$\left[ \frac{1}{\left(2 + 2\left(\frac{2a}{1-a} + 1\right)\right)^2} \frac{1}{1 + \frac{\left(\frac{1+a}{2}\right)^2}{1 - \left(\frac{1+a}{2}\right)^2}} \frac{1}{1 - a^2} \cdot \frac{1 + \left(\frac{1-a}{1+a}\right)^{2k+1}}{1 + \frac{1-a}{1+a}}, \; \frac{1}{1 - a^2} \cdot \frac{1 + \left(\frac{1-a}{1+a}\right)^{2k+1}}{1 + \frac{1-a}{1+a}} \right],$$

*and $\frac{u_{a,2k-1}}{u_{0,2k}} \in$*

$$\left[ \frac{1}{\left(2 + 2\left(\frac{2a}{1-a} + 1\right)\right)^2} \frac{1}{1 + \frac{\left(\frac{1+a}{2}\right)^2}{1 - \left(\frac{1+a}{2}\right)^2}} \frac{1}{1 - a^2} \cdot \frac{1 - \left(\frac{1-a}{1+a}\right)^{2k}}{1 + \frac{1-a}{1+a}}, \; \frac{1}{1 - a^2} \cdot \frac{1 - \left(\frac{1-a}{1+a}\right)^{2k}}{1 + \frac{1-a}{1+a}} \right].$$

We prove Lemma 53 in Appendix M.1.1.

Note that when $k \to \infty$, the terms that depend on $k$ (i.e., $\left(\frac{1-a}{1+a}\right)^{2k+1}$ and $\left(\frac{1-a}{1+a}\right)^{2k}$) all approach 0. In other words, as $k$ becomes larger, $u_{a,2k}$ (as well as $u_{a,2k+1}$) approaches (approximately) a constant (that only depends on $a$) multiple of $u_{0,2k}$. Therefore, by Lemma 53, we can then conclude that $u_{b_1,2k} = \Theta(u_{0,2k})$ and $u_{b_1,k} = \Theta(u_{b_2,k})$ when $k$ is large for any $b_1 \in (0, 1)$ and $b_2 \in (0, 1)$. By Lemma 52, it immediately implies that $c_{h_{b_1}^{(2)}}(2k, \mathbf{0}) = \Theta\left(c_{h^{(2)}}(2k, \mathbf{0})\right)$,

$c_{h_{b_1}^{(2)}}(k, \mathbf{0}) = \Theta\left(c_{h_{b_2}^{(2)}}(k, \mathbf{0})\right)$. This proves the first and second statements of Lemma 51.

### M.1.1 Proof of Lemma 53

We first write the form of $u_{0,l}$, i.e., the Taylor coefficients under no-bias setting. By Lemma 17, we have

$$x \frac{\pi - \arccos x}{2\pi} = \frac{x}{4} + \frac{1}{2\pi} \sum_{k=0}^{\infty} \frac{(2k)!}{(k!)^2} \frac{4}{2k+1} \left(\frac{x}{2}\right)^{2k+2}.$$

Thus, for $k \geq 1$, we have

$$u_{0,2k} = \frac{1}{2\pi} \frac{(2k-2)!}{((k-1)!)^2} \frac{4}{2k-1} \frac{1}{2^{2k}}, \tag{92}$$

$$u_{0,2k+1} = 0. \tag{93}$$

Next, we write the expression of $u_{a,l}$. To that end, we define

$$d_{a,2k,i} := \binom{2k+2i}{2k}(1-a)^{2k}a^{2i}, \tag{94}$$

$$d_{a,2k+1,i} := \binom{2k+2i+2}{2k+1}(1-a)^{2k+1}a^{2i+1}. \tag{95}$$

The following lemma provides the expression of $u_{a,l}$.

**Lemma 54.** *For any $k \geq 1$, we must have*

$$u_{a,2k} = \sum_{i=0}^{\infty} u_{0,2(k+i)} d_{a,2k,i},$$

$$u_{a,2k+1} = \sum_{i=0}^{\infty} u_{0,2(k+i+1)} d_{a,2k+1,i}.$$

We prove Lemma 54 in Appendix M.1.2.

Although we have the expression of $u_{a,k}$ by Lemma 54, it is not easy to directly estimate its value because terms like $u_{0,2(k+i)} d_{a,2k,i}$ have a very complicated form. Fortunately, some properties of $u_{0,2k}$ is very helpful. Specifically, by Eq. (92), we have $\frac{u_{0,2(k+1)}}{u_{0,2k}} = \frac{(2k-1)^2 2k}{k^2(2k+1)\cdot 4} = \frac{(2k-1)^2}{2k(2k+1)}$, whose value approaches 1 when $k \to \infty$. In other words, $u_{0,2k}$ has a very slow changing speed when $k$ is large. Therefore, we can approximate the tail of $\sum_{i=0}^{\infty} u_{0,2(k+i)} d_{a,2k,i}$ by treating $u_{0,2(k+i)}$ as a constant. This allows us to focus our attention on estimating $\sum_{i=0}^{\infty} d_{a,2k,i}$ (and its tail $\sum_{i=l}^{\infty} d_{a,2k,i}$), whose value can be calculated by examining the coefficients of the Taylor expansion of $\frac{1}{1-((1-a)x+a)^2}$ (i.e., the sum of a geometric sequence $1, ((1-a)x+a)^2, , ((1-a)x+a)^4, \cdots$). The latter is much easier to study. We show these steps in detail as the following lemmas.

Define

$$l := \max\left\{k, \left\lceil \frac{2a}{1-a} k \right\rceil\right\}. \tag{96}$$

The following lemma estimates the target ratio $\frac{u_{a,2k}}{u_{0,2k}}$ and $\frac{u_{a,2k-1}}{u_{0,2k}}$ in terms of $\sum_{i=0}^{l} d_{a,2k,i}$ and $\sum_{i=0}^{\infty} d_{a,2k,i}$.

**Lemma 55.** *For any $k \in \{2, 3, \cdots\}$, we must have*

$$\frac{u_{a,2k}}{u_{0,2k}} \in \left[ \frac{1}{\left(2 + 2\left(\frac{2a}{1-a}+1\right)\right)^2} \sum_{i=0}^{l} d_{a,2k,i}, \ \sum_{i=0}^{\infty} d_{a,2k,i} \right],$$

$$\frac{u_{a,2k-1}}{u_{0,2k}} \in \left[ \frac{1}{\left(2 + 2\left(\frac{2a}{1-a}+1\right)\right)^2} \sum_{i=0}^{l} d_{a,2k-1,i}, \ \sum_{i=0}^{\infty} d_{a,2k-1,i} \right].$$

We prove Lemma 55 in Appendix M.1.3.

In order to finish the proof of Lemma 53, it only remains to estimate $\sum_{i=0}^{l} d_{a,2k,i}$, $\sum_{i=0}^{l} d_{a,2k-1,i}$, $\sum_{i=0}^{\infty} d_{a,2k,i}$, $\sum_{i=0}^{\infty} d_{a,2k-1,i}$, which are shown by the following two lemmas.

**Lemma 56.** *For any $k \in \{2, 3, \cdots\}$, we must have*

$$\sum_{i=0}^{\infty} d_{a,2k,i} = \frac{1}{1-a^2} \cdot \frac{1 + \left(\frac{1-a}{1+a}\right)^{2k+1}}{1 + \frac{1-a}{1+a}}, \quad \sum_{i=0}^{\infty} d_{a,2k-1,i} = \frac{1}{1-a^2} \cdot \frac{1 - \left(\frac{1-a}{1+a}\right)^{2k}}{1 + \frac{1-a}{1+a}}.$$

We prove Lemma 56 in Appendix M.1.4

**Lemma 57.** *Recall that $l$ is defined in Eq. (96). For any $k \in \{2, 3, \cdots\}$, we must have*

$$\frac{\sum_{i=0}^{l} d_{a,2k,i}}{\sum_{i=0}^{\infty} d_{a,2k,i}}, \ \frac{\sum_{i=0}^{l} d_{a,2k-1,i}}{\sum_{i=0}^{\infty} d_{a,2k-1,i}} \in \left[ \frac{1}{1 + \frac{\left(\frac{1+a}{2}\right)^2}{1-\left(\frac{1+a}{2}\right)^2}}, \ 1 \right].$$

We prove Lemma 57 in Appendix M.1.5

The result of Lemma 53 follows by combining Lemma 56, Lemma 57, and Lemma 55.

### M.1.2 Proof of Lemma 54

*Proof.* We have

$$
((1-a)x+a)\,\frac{\pi-\arccos\left((1-a)x+a\right)}{2\pi}
$$

$$
=\sum_{m=0}^{\infty} u_{0,m}\left((1-a)x+a\right)^m
$$

$$
=\sum_{m=0}^{\infty} u_{0,m}\sum_{i=0}^{m}\binom{m}{i}(1-a)^i a^{m-i}x^i
$$

$$
=\sum_{i=0}^{\infty}\sum_{j=0}^{\infty} u_{0,i+j}\binom{i+j}{i}(1-a)^i a^j x^i \text{ (replace } m \text{ by } i+j \text{ and reorganize terms).}
$$

Thus, we have

$$
u_{a,i}=\sum_{j=0}^{\infty} u_{0,i+j}\binom{i+j}{i}(1-a)^i a^j.
$$

Letting $i=2k$, we have

$$
u_{a,2k}=\sum_{j=0}^{\infty} u_{0,2k+j}\binom{2k+j}{2k}(1-a)^{2k}a^j
$$

$$
=\sum_{i=0}^{\infty} u_{0,2(k+i)}\binom{2k+2i}{2k}(1-a)^{2k}a^{2i} \text{ (by Eq. (93) and letting } j=2i)
$$

$$
=\sum_{i=0}^{\infty} u_{0,2(k+i)}d_{a,2k,i} \text{ (by Eq. (94)).}
$$

Similarly, we have

$$
u_{a,2k+1}=\sum_{j=0}^{\infty} u_{0,2k+1+j}\binom{2k+1+j}{2k+1}(1-a)^{2k+1}a^j
$$

$$
=\sum_{i=0}^{\infty} u_{0,2(k+i+1)}\binom{2k+2i+2}{2k+1}(1-a)^{2k+1}a^{2i+1} \text{ (by Eq. (93) and letting } j=2i+1)
$$

$$
=\sum_{i=0}^{\infty} u_{0,2(k+i+1)}d_{a,2k+1,i} \text{ (by Eq. (95)).}
$$

The result of this lemma thus follows. $\qquad\square$

### M.1.3 Proof of Lemma 55

*Proof.* By Eq. (92), we have

$$
\frac{u_{0,2(k+1)}}{u_{0,2k}}=\frac{(2k-1)^2 2k}{k^2(2k+1)\cdot 4}=\frac{(2k-1)^2}{2k(2k+1)} \tag{97}
$$

$$
\leq 1.
$$

By iterating the above inequality, we have $u_{0,2(k+i)}\leq u_{0,2k}$ for all $i\geq 0$. By Lemma 54, we thus have

$$
\frac{u_{a,2k}}{u_{0,2k}}=\frac{\sum_{i=0}^{\infty} u_{0,2(k+i)}d_{a,2k,i}}{u_{0,2k}}\leq\sum_{i=0}^{\infty} d_{a,2k,i}.
$$

Similarly, we have

$$
\frac{u_{a,2k-1}}{u_{0,2k}}=\frac{\sum_{i=0}^{\infty} u_{0,2(k+i)}d_{a,2k-1,i}}{u_{0,2k}}\leq\sum_{i=0}^{\infty} d_{a,2k-1,i}.
$$

These prove the upper bounds in Lemma 55. To prove the lower bounds, note that for any $m \in \{0, 1, \cdots, l\}$ (recall that $l$ is defined in Eq. (96)), we must have

$$
\begin{aligned}
\frac{u_{0,2(k+m)}}{u_{0,2k}} &= \prod_{i=0}^{m-1} \frac{(2k+2i-1)^2}{(2k+2i)(2k+2i+1)} \quad \text{(by Eq. (97))} \\
&\geq \prod_{i=0}^{m-1} \frac{(2k+2i-1)^2}{(2k+2i+1)^2} \\
&= \frac{(2k-1)^2}{(2k+2m-1)^2} \\
&\geq \frac{k^2}{(2k+2m)^2} \quad \text{(using } 2k-1 \geq k, \text{ which is true because } k \geq 1) \\
&= \frac{1}{\left(2 + \frac{2m}{k}\right)^2} \\
&\geq \frac{1}{\left(2 + 2\left(\frac{2a}{1-a} + 1\right)\right)^2} \quad \text{(because } m \leq l \leq \frac{2a}{1-a}k + k \text{ for } k \geq 2).
\end{aligned}
$$

Thus, we have

$$
\begin{aligned}
\frac{u_{a,2k}}{u_{0,2k}} &= \frac{\sum_{i=0}^{\infty} u_{0,2(k+i)} d_{a,2k,i}}{u_{0,2k}} \quad \text{(by Lemma 54)} \\
&\geq \frac{\sum_{i=0}^{l} u_{0,2(k+i)} d_{a,2k,i}}{u_{0,2k}} \\
&\geq \frac{1}{\left(2 + 2\left(\frac{2a}{1-a} + 1\right)\right)^2} \sum_{i=0}^{l} d_{a,2k,i}.
\end{aligned}
$$

Similarly, we have

$$
\frac{u_{a,2k-1}}{u_{0,2k}} \geq \frac{1}{\left(2 + 2\left(\frac{2a}{1-a} + 1\right)\right)^2} \sum_{i=0}^{l} d_{a,2k-1,i}.
$$

The result of this lemma thus follows. $\qquad\square$

### M.1.4 Proof of Lemma 56

We first state a useful fact.

**Lemma 58.** *For any $|r| < 1$, we have*

$$
\frac{1}{1-r} = \sum_{i=0}^{\infty} r^i.
$$

*Proof.* The result of this lemma directly follows the sum of a geometric series (noticing that $\lim_{i \to \infty} r^i = 0$ when $|r| < 1$). $\qquad\square$

*Proof of Lemma 56.* The proof idea is to express the coefficients of the Taylor expansion of $\frac{1}{1-((1-a)x+a)^2}$ (where $|x| < 1$) in two different ways. On the one hand, we have

$$\frac{1}{1 - ((1-a)x + a)^2}$$

$$= \sum_{m=0}^{\infty} ((1-a)x + a)^{2m} \quad \text{(by letting } r = (1-a)x + a \text{ in Lemma 58)}$$

$$= \sum_{m=0}^{\infty} \sum_{j=0}^{2m} \binom{2m}{j} (1-a)^j a^{2m-j} x^j$$

$$= \sum_{k=0}^{\infty} \left( \left( \sum_{i=0}^{\infty} \binom{2k+2i}{2k} (1-a)^{2k} a^{2i} \right) x^{2k} + \left( \sum_{i=0}^{\infty} \binom{2k+2i+2}{2k+1} (1-a)^{2k+1} a^{2i+1} \right) x^{2k+1} \right)$$

(by letting $j = 2k$, $2m = 2k + 2i$ for $x^{2k}$ and letting $j = 2k + 1$, $2m = 2k + 2i + 2$ for $x^{2k+1}$)

$$= \sum_{k=0}^{\infty} \left( \left( \sum_{i=0}^{\infty} d_{a,2k,i} \right) x^{2k} + \left( \sum_{i=0}^{\infty} d_{a,2k+1,i} \right) x^{2k+1} \right) \quad \text{(by Eq. (94) and Eq. (95)).} \tag{98}$$

One the other hand, we have

$$\frac{1}{1 - ((1-a)x + a)^2}$$

$$= \frac{1}{1 - ((1-a)x + a)} \cdot \frac{1}{1 + ((1-a)x + a)}$$

$$= \frac{1}{1-a} \cdot \frac{1}{1-x} \cdot \frac{1}{1+a} \cdot \frac{1}{1 + \frac{1-a}{1+a}x}$$

$$= \frac{1}{1 - a^2} \left( \sum_{i=0}^{\infty} x^i \right) \left( \sum_{j=0}^{\infty} \left( -\frac{1-a}{1+a}x \right)^j \right) \quad \text{(by Lemma 58)}$$

$$= \frac{1}{1 - a^2} \sum_{i=0}^{\infty} \sum_{j=0}^{\infty} \left( -\frac{1-a}{1+a} \right)^j x^{i+j}$$

$$= \frac{1}{1 - a^2} \sum_{m=0}^{\infty} \left( \sum_{j=0}^{m} \left( -\frac{1-a}{1+a} \right)^j \right) x^m \quad \text{(combine terms of } x^{i+j} \text{ with } i + j = m)$$

$$= \frac{1}{1 - a^2} \sum_{m=0}^{\infty} \frac{1 - \left( -\frac{1-a}{1+a} \right)^{m+1}}{1 + \frac{1-a}{1+a}} x^m. \tag{99}$$

By comparing the coefficients in Eq. (98) and Eq. (99), the result of this lemma thus follows. $\quad \square$

### M.1.5   Proof of Lemma 57

We first prove a useful lemma.

**Lemma 59.** *For any $a > b > c > 0$, we have*

$$\frac{a}{b} < \frac{a - c}{b - c}.$$

*Proof.* Because $a > b > c > 0$, we have

$$b < a \implies bc < ac \implies ab - ac < ab - bc \implies a(b - c) < b(a - c) \implies \frac{a}{b} < \frac{a - c}{b - c}.$$

$\quad \square$

Now we are ready to prove Lemma 57.

*Proof of Lemma 57.* Recall that $l$ is defined in Eq. (96). For any $i \geq l$, we have

$$\frac{k+i}{i} = \frac{k}{i} + 1 \leq \frac{k}{l} + 1 \leq \frac{1+a}{2a} \text{ (because } l \geq \frac{2a}{1-a}k \text{ by Eq. (96)).} \qquad (100)$$

Thus, by Eq. (94), we have

$$\frac{d_{a,2k,i+1}}{d_{a,2k,i}} = \frac{\binom{2k+2i+2}{2k}}{\binom{2k+2i}{2k}}a^2 = \frac{(2k+2i+1)(2k+2i+2)}{(2i+1)(2i+2)}a^2$$
$$\leq \frac{(2k+2i)(2k+2i)}{(2i)\cdot(2i)}a^2 \text{ (by Lemma 59)}$$
$$\leq \left(\frac{1+a}{2}\right)^2 \text{ (by Eq. (100)).}$$

Similarly, by Eq. (95), we have

$$\frac{d_{a,2k-1,i+1}}{d_{a,2k-1,i}} = \frac{\binom{2k+2i+2}{2k-1}}{\binom{2k+2i}{2k-1}}a^2 = \frac{(2k+2i+2)(2k+2i+1)}{(2i+2)(2i+3)}a^2$$
$$\leq \frac{(2k+2i+2)(2k+2i+3)}{(2i+2)(2i+3)}a^2$$
$$\leq \frac{(2k+2i)(2k+2i)}{(2i)\cdot(2i)}a^2 \text{ (by Lemma 59)}$$
$$\leq \left(\frac{1+a}{2}\right)^2 \text{ (by Eq. (100)).}$$

Iterating the above inequalities, we have

$$\frac{d_{a,2k,l+j}}{d_{a,2k,l}} \leq \left(\frac{1+a}{2}\right)^{2j}, \quad \text{and} \quad \frac{d_{a,2k-1,l+j}}{d_{a,2k-1,l}} \leq \left(\frac{1+a}{2}\right)^{2j}.$$

Thus, we have

$$\frac{\sum_{i=l+1}^{\infty} d_{a,2k,i}}{\sum_{i=0}^{l} d_{a,2k,i}} \leq \frac{\sum_{i=l+1}^{\infty} d_{a,2k,i}}{d_{a,2k,l}} = \sum_{j=1}^{\infty} \frac{d_{a,2k,l+j}}{d_{a,2k,l}} \leq \sum_{j=1}^{\infty} \left(\frac{1+a}{2}\right)^{2j} = \frac{\left(\frac{1+a}{2}\right)^2}{1-\left(\frac{1+a}{2}\right)^2}.$$

We then have

$$\frac{\sum_{i=0}^{\infty} d_{a,2k,i}}{\sum_{i=0}^{l} d_{a,2k,i}} = \frac{\sum_{i=0}^{l} d_{a,2k,i} + \sum_{i=l+1}^{\infty} d_{a,2k,i}}{\sum_{i=0}^{l} d_{a,2k,i}} \leq 1 + \frac{\left(\frac{1+a}{2}\right)^2}{1-\left(\frac{1+a}{2}\right)^2}.$$

Therefore, we conclude that

$$\frac{\sum_{i=0}^{l} d_{a,2k,i}}{\sum_{i=0}^{\infty} d_{a,2k,i}} \in \left[\frac{1}{1 + \frac{\left(\frac{1+a}{2}\right)^2}{1-\left(\frac{1+a}{2}\right)^2}}, 1\right].$$

Similarly, we have

$$\frac{\sum_{i=0}^{l} d_{a,2k-1,i}}{\sum_{i=0}^{\infty} d_{a,2k-1,i}} \in \left[\frac{1}{1 + \frac{\left(\frac{1+a}{2}\right)^2}{1-\left(\frac{1+a}{2}\right)^2}}, 1\right].$$

$\square$

## M.2 Expansion for a finite-degree polynomial

We plan to show the third statement of Lemma 51, which was presented in Appendix L. This is more difficult because $K^{\mathrm{RF}}(x)$ is an infinite-degree polynomial (in contrast, Appendix M.1 deals with $K(x) = (1-a)x + a$, which is much simpler). To make progress, we first consider a finite-degree polynomial, and study the expansion. Then, in Appendix M.3, we will extend to $K^{\mathrm{RF}}(x)$ which has infinite degree. Note that Appendix M.1 is a special case of Appendix M.2. However, since Appendix M.1 is much simpler and easy to understand, we retain the proof there, and use the result in Appendix M.2 only as a preparation for Appendix M.3.

Recall the definition of $T(x)$ in Eq. (91). We denote $K(x)$ as a polynomial, i.e.,

$$K(x) = \sum_{i=0}^{\infty} a_i x^i, \tag{101}$$

where $a_i$ denote the coefficient of $x^i$ in $K(x)$.

Define $u_m(\cdot)$ as a function that projects a polynomial in $x$ to a real value such that

$$\tilde{K}(x) \frac{\pi - \arccos\left(\tilde{K}(x)\right)}{2\pi} = \sum_{m=0}^{\infty} u_m\left(\tilde{K}(x)\right) \cdot x^m, \tag{102}$$

where $\tilde{K}(x)$ is any polynomial of $x$. In other words, $u_m(\tilde{K}(x))$ is the Taylor coefficient of $x^m$ in $T(x)$ when $K(x) = \tilde{K}(x)$.

In this subsection, we let the number of terms of $K(x)$ be finite, i.e., there exists $s$ such that $a_i = 0$ for all $i > s$. Further, we impose the following conditions.

*Condition* 2. (i) All coefficients of $K(x)$ are non-negative, i.e., $a_i \geq 0$ for all $i \in \mathbb{Z}_{\geq 0}$. (ii) The sum of all coefficients equals to 1, i.e., $\sum_{i=0}^{\infty} a_i = K(1) = 1$. (iii) $a_0 > 0$ and $a_1 > 0$.

The following lemma shows that when $K(x)$ is a polynomial with finite terms, the Taylor coefficients are on the same order as that of the even-power Taylor coefficients when $K(x) = x$. Note that, according to Eq. (102), when $K(x) = x$, $u_m(x)$ recovers the Taylor coefficients of the polynomial expansion of the function $x \frac{\pi - \arccos(x)}{2\pi}$.

**Lemma 60.** *Under Condition 2 and when $a_i = 0$ for all $i > s$, we must have*

$$\frac{u_j\left(K(x)\right)}{u_{2\lceil j/2 \rceil}(x)} \in \left[\underline{C}, \overline{C}\right], \text{ for all } j = 1, 2, \cdots,$$

*where $\overline{C} > \underline{C} > 0$ are constants that only depends on $K(x)$ and are independent of $j$.*

We prove Lemma 60 in Appendix M.2.1. Note that Lemma 60 can be seen as a generalization of Lemma 53, since $K(x) = (1-a) + a$ satisfies Condition 2 when $a \in (0,1)$.

### M.2.1 Proof of Lemma 60

We introduce some extra notations. Let $b_i$ be the coefficients of $x^i$ of $(K(x))^2$, i.e.,

$$(K(x))^2 = \sum_{i=0}^{2s} b_i x^i, \text{ which implies that } b_i = \sum_{j+k=i} a_j a_k \text{ for all } i \in \mathbb{Z}_{\geq 0}. \tag{103}$$

As in Lemma 27, for all $j \in \mathbb{Z}_{\geq 0}$, we define

$$t(m_0, m_1, \cdots, m_j) := (m_0, m_1, \cdots, m_j)! \cdot a_0^{m_0} a_1^{m_1} \cdots a_j^{m_j} \text{ (we let } a_i = 0 \text{ if } i > s), \tag{104}$$

$$\mathcal{T}_{i,j} := \left\{ (m_0, m_1, \cdots, m_j) \,\middle|\, \begin{array}{c} m_0 + m_1 + \cdots + m_j = i \\ m_1 + 2m_2 + \cdots + j \cdot m_j = j \\ m_0, m_1, \cdots, m_j \in \mathbb{Z}_{\geq 0} \end{array} \right\},$$

$$d_{i,j} := \sum_{(m_0, m_1, \cdots, m_j) \in \mathcal{T}_{i,j}} t(m_0, m_1, \cdots, m_j). \tag{105}$$

| Notation | Description | Definition/Expression |
|:---:|:---:|:---:|
| $a_i$ | coefficients of $x^i$ in $K(x)$ | Eq. (101) |
| $b_i$ | coefficients of $x^i$ in $(K(x))^2$ | Eq. (103) |
| $u_i(\tilde{K}(x))$ | coefficients of $x^i$ in $\tilde{K}(x)(\pi - \arccos(\tilde{K}(x)))/(2\pi)$ | Eq. (102) and Eq. (115) |
| $d_{i,j}$ | the coefficient of $x^j$ in $(K(x))^i$ | Eq. (105) |
| $e_i$ | the coefficient of $x^i$ in $1/(1 - (K(x))^2)$ | Eq. (106) and Eq. (107) |

Table 2: Summary of the notations of various coefficients.

By Lemma 27, we have

$$\frac{1}{1 - (K(x))^2} = \sum_{i=0}^{\infty} (K(x))^{2i} = \sum_{i=0}^{\infty} \left( \sum_{j=0}^{\infty} d_{2i,j} x^j \right)$$

$$= \sum_{j=0}^{\infty} \left( \sum_{i=\lceil \frac{j}{2s} \rceil}^{\infty} d_{2i,j} \right) x^j \text{ (since } d_{2i,j} = 0 \text{ when } j > 2i \cdot s\text{)}.$$

Define

$$e_j \text{ for all } j \in \mathbb{Z}_{\geq 0} \text{ such that } \frac{1}{1 - (K(x))^2} = \sum_{j=0}^{\infty} e_j x^j, \tag{106}$$

i.e.,

$$e_j = \sum_{i=\lceil \frac{j}{2s} \rceil}^{\infty} d_{2i,j}. \tag{107}$$

We summarize those definitions in Table 2.

**Lemma 61.** *Under Condition 2, we must have*

$$\sum_{i=0}^{2s} b_i = 1, \tag{108}$$

$b_i \in [0, 1]$ *for all* $i \in \{0, 1, \cdots, 2s\}$, $b_0 < 1$, *and* $b_1 > 0$.

*Proof.* By Eq. (103) and Condition 2, we have $\sum_{i=0}^{2s} b_i = (K(1))^2 = 1$. Because $a_1 > 0$ and $\sum_{i=0}^{s} a_i = 1$, we have $a_0 < 1$. Thus, we have $b_0 = a_0^2 < 1$ and $b_1 = 2a_0 a_1 > 0$ (as $a_0$ is also positive). $\qquad \square$

**Lemma 62.** *There exist* $\bar{c} \geq \underline{c} > 0$ *such that for all* $k \in \mathbb{Z}_{\geq 0}$, *we must have* $e_k \in [\underline{c}, \bar{c}]$.

*Proof.* Because

$$\frac{1}{1 - (K(x))^2} = 1 + (K(x))^2 \cdot \frac{1}{1 - (K(x))^2},$$

we have

$$\sum_{j=0}^{\infty} e_j x^j = 1 + \left( \sum_{i=0}^{s} a_i x^i \right)^2 \cdot \left( \sum_{j=0}^{\infty} e_j x^j \right) = 1 + \left( \sum_{i=0}^{2s} b_i x^i \right) \cdot \left( \sum_{j=0}^{\infty} e_j x^j \right). \tag{109}$$

Comparing the coefficient of $x^{j+2s}$ on both sides, we have

$$e_{j+2s} = \sum_{i=0}^{2s} e_{j+i}b_{2s-i} \text{ for all } j = 0, 1, \cdots.$$

This is equivalent to

$$e_{j+2s} = b_0 e_{j+2s} + \sum_{i=0}^{2s-1} e_{j+i}b_{2s-i} \text{ for all } j = 0, 1, \cdots.$$

Thus, we have

$$e_{j+2s} = \sum_{i=0}^{2s-1} \frac{b_{2s-i}}{1-b_0} e_{j+i} \text{ for all } j = 0, 1, \cdots.$$

It implies that

$$e_{j+2s} \in \left[ \left( \min_{i\in\{0,1,\cdots,2s-1\}} e_{j+i} \right) \cdot \sum_{k=1}^{2s} \frac{b_k}{1-b_0}, \left( \max_{i\in\{0,1,\cdots,2s-1\}} e_{j+i} \right) \cdot \sum_{k=1}^{2s} \frac{b_k}{1-b_0} \right]$$
$$\text{for all } j = 0, 1, \cdots.$$

By Eq. (108), we have $\sum_{k=1}^{2s} b_k = 1 - b_0$, which implies that $\sum_{k=1}^{2s} \frac{b_k}{1-b_0} = 1$. Thus, we have

$$e_{j+2s} \in \left[ \min_{i\in\{0,1,\cdots,2s-1\}} e_{j+i}, \max_{i\in\{0,1,\cdots,2s-1\}} e_{j+i} \right], \text{ for all } j = 0, 1, \cdots.$$

Iteratively applying the above bounds, we then have

$$e_{2s}, e_{2s+1}, \cdots, e_{4s-1} \in \left[ \min_{i\in\{0,1,\cdots,2s-1\}} e_i, \max_{i\in\{0,1,\cdots,2s-1\}} e_i \right],$$
$$e_{4s}, e_{4s+1}, \cdots, e_{6s-1} \in \left[ \min_{i\in\{2s,2s+1,\cdots,4s-1\}} e_i, \max_{i\in\{2s,2s+1,\cdots,4s-1\}} e_i \right] \in \left[ \min_{i\in\{0,1,\cdots,2s-1\}} e_i, \right.$$
$$\left. \max_{i\in\{0,1,\cdots,2s-1\}} e_i \right],$$
$$\vdots$$
$$e_{2ks}, e_{2ks+1}, \cdots, e_{2ks+2s-1} \in \cdots \in \left[ \min_{i\in\{0,1,\cdots,2s-1\}} e_i, \max_{i\in\{0,1,\cdots,2s-1\}} e_i \right].$$

In other words,

$$e_k \in \left[ \min_{i\in\{0,1,\cdots,2s-1\}} e_i, \max_{i\in\{0,1,\cdots,2s-1\}} e_i \right], \text{ for all } k = 2s, 2s+1, \cdots. \tag{110}$$

By Eq. (109), we have

$$e_0 = 1 + b_0 e_0,$$
$$e_1 = b_1 e_0 + b_0 e_1,$$
$$e_2 = b_2 e_0 + b_1 e_1 + b_0 e_2,$$
$$\vdots$$
$$e_{2s} = b_{2s} e_0 + b_{2s-1} e_1 + \cdots + b_0 e_{2s}.$$

Thus, we have

$$e_0 = \frac{1}{1 - b_0},$$

$$e_1 = \frac{b_1 e_0}{1 - b_0},$$

$$e_2 = \frac{b_2 e_0 + b_1 e_1}{1 - b_0},$$

$$\vdots$$

$$e_{2s} = \frac{b_{2s} e_0 + b_{2s-1} e_1 + \cdots + b_1 e_{2s-1}}{1 - b_0}.$$

By Lemma 61 and using induction, we thus have

$$e_i > 0, \text{ for all } i = 0, 1, \cdots, 2s - 1.$$

Thus, by Eq. (110), we have

$$e_k \in \left[ \min_{i \in \{0,1,\cdots,2s-1\}} e_i, \ \max_{i \in \{0,1,\cdots,2s-1\}} e_i \right], \text{ for all } k = 0, 1, \cdots.$$

$\square$

**Lemma 63.** *When $i \geq j$, there exists a bijection between $\mathcal{T}_{i,j}$ and $\mathcal{T}_{i+1,j}$. Specifically, this bijection is $\mathcal{T}_{i,j} \longleftrightarrow \mathcal{T}_{i+1,j}$: $(m_0, m_1, \cdots, m_j) \longleftrightarrow (m_0 + 1, m_1, \cdots, m_j)$.*

*Proof.* It suffices to show that for any $(m_0, m_1, \cdots, m_j) \in \mathcal{T}_{i+1,j}$, we must have $m_0 \geq 1$. To that end, note that when $i \geq j$, for any $(m_0, m_1, \cdots, m_s) \in \mathcal{T}_{i+1,j}$, we have

$$\sum_{k=0}^{j} m_k = i + 1, \quad \sum_{k=1}^{j} k m_k = j.$$

Thus, we have

$$m_0 = \sum_{k=0}^{j} m_k - \sum_{k=1}^{j} k m_k + \sum_{k=1}^{j} (k-1) m_k = (i + 1 - j) + \sum_{k=1}^{j} (k-1) m_k \geq 1 \text{ (because } i \geq j).$$

The result of this lemma thus follows. $\square$

**Lemma 64.** *If $2i \geq j$, then*

$$\frac{d_{2i+2,j}}{d_{2i,j}} \leq \left( \frac{2i}{2i - j} a_0 \right)^2.$$

*(Notice that when $2i = j$, the right hand side is infinite. Nonetheless, this lemma still holds.)*

*Proof.* We have

$$\frac{d_{2i+2,j}}{d_{2i,j}} = \frac{\sum_{\boldsymbol{m} \in \mathcal{T}_{2i+2,j}} t(\boldsymbol{m})}{\sum_{\boldsymbol{m} \in \mathcal{T}_{2i,j}} t(\boldsymbol{m})}.$$

Let $(m_0^{(k)}, m_1^{(k)}, \cdots, m_j^{(k)})$ denote the $k$-th element in $\mathcal{T}_{2i,j}$. Because $m_0 + m_1 + \cdots + m_j = 2i$ and $m_1 + 2m_2 + \cdots + j \cdot m_j = j$, we have $m_0 \geq 2i - j$. Thus, using the definition of $t(\cdots)$ in Eq. (104), we have

$$\frac{t\left( m_0^{(k)} + 2, m_1^{(k)}, \cdots, m_j^{(k)} \right)}{t\left( m_0^{(k)}, m_1^{(k)}, \cdots, m_j^{(k)} \right)}$$

$$= \frac{(2i + 1)(2i + 2)}{(m_0 + 1)(m_0 + 2)} a_0^2$$

$$\leq \frac{(2i + 1)(2i + 2)}{(2i - j + 1)(2i - j + 2)} a_0^2 \text{ (because } m_0 \geq 2i - j)$$

$$\leq \left( \frac{2i}{2i - j} a_0 \right)^2 \text{ (because } \frac{2i + 1}{2i - j + 1} \leq \frac{2i}{2i - j} \text{ and } \frac{2i + 2}{2i - j + 2} \leq \frac{2i}{2i - j}).$$

By Lemma 63 and $2i \geq j$, we thus have

$$\frac{d_{2i+2,j}}{d_{2i,j}} = \frac{t\left(m_0^{(1)}+2, m_1^{(1)}, \cdots, m_j^{(1)}\right) + \cdots + t\left(m_0^{(|\mathcal{T}_{2i,j}|)}+2, m_1^{(|\mathcal{T}_{2i,j}|)}, \cdots, m_j^{(|\mathcal{T}_{2i,j}|)}\right)}{t\left(m_0^{(1)}, m_1^{(1)}, \cdots, m_j^{(1)}\right) + \cdots + t\left(m_0^{(|\mathcal{T}_{2i,j}|)}, m_1^{(|\mathcal{T}_{2i,j}|)}, \cdots, m_j^{(|\mathcal{T}_{2i,j}|)}\right)}$$

$$\leq \left(\frac{2i}{2i-j}a_0\right)^2.$$

$\square$

**Lemma 65.** *For any $j = 1, 2, \cdots$, we must have*

$$\sum_{i=\lceil \frac{j}{2s} \rceil}^{i^*} d_{2i,j} \geq \frac{1 - \left(\frac{1+a_0}{2}\right)^2}{2 - \left(\frac{1+a_0}{2}\right)^2} \sum_{i=\lceil \frac{j}{2s} \rceil}^{\infty} d_{2i,j},$$

*where*

$$i^* := \left\lceil \frac{1+a_0}{2(1-a_0)} j \right\rceil.$$

*Proof.* For all $i \geq i^*$, we have $2i \geq j$ and

$$\frac{2i}{2i-j}a_0 \leq \frac{2i^*}{2i^*-j}a_0 \leq \frac{\frac{1+a_0}{1-a_0}j}{\frac{1+a_0}{1-a_0}j - j}a_0 = \frac{1+a_0}{2} \text{ (by Lemma 59 and } 2i \geq 2i^* \geq \frac{1+a_0}{1-a_0}j).$$

By Lemma 64, we thus have

$$\frac{d_{2i+2,j}}{d_{2i,j}} \leq \left(\frac{1+a_0}{2}\right)^2 \text{ for all } i \geq i^*.$$

Because $d_{2i,j} \geq 0$ for all $i$ and $j$, we have

$$\sum_{i=i^*+1}^{\infty} d_{2i,j} \leq \sum_{i=i^*}^{\infty} d_{2i,j} \leq d_{2i^*,j} \sum_{k=0}^{\infty} \left(\frac{1+a_0}{2}\right)^{2k} = \frac{d_{2i^*,j}}{1 - \left(\frac{1+a_0}{2}\right)^2} \leq \frac{1}{1 - \left(\frac{1+a_0}{2}\right)^2} \sum_{i=\lceil \frac{j}{2s} \rceil}^{i^*} d_{2i,j}.$$

Therefore, we have

$$\left(1 + \frac{1}{1 - \left(\frac{1+a_0}{2}\right)^2}\right) \sum_{i=\lceil \frac{j}{2s} \rceil}^{i^*} d_{2i,j} \geq \sum_{i=0}^{i^*} d_{2i,j} + \sum_{i=i^*+1}^{\infty} d_{2i,j} = \sum_{i=\lceil \frac{j}{2s} \rceil}^{\infty} d_{2i,j},$$

i.e.,

$$\sum_{i=\lceil \frac{j}{2s} \rceil}^{i^*} d_{2i,j} \geq \frac{1 - \left(\frac{1+a_0}{2}\right)^2}{2 - \left(\frac{1+a_0}{2}\right)^2} \sum_{i=\lceil \frac{j}{2s} \rceil}^{\infty} d_{2i,j}.$$

$\square$

Recall that $u_m(x)$ denotes the Taylor coefficients of $x^{\frac{\pi - \arccos(x)}{2\pi}}$. The following lemma states that $u_{2k}(x)$ is monotone decreasing with respect to $k$. Further, it estimates the decreasing speed. We draw the curve of $u_k(x)$ with respect to $k$ in Fig. 11.

**Lemma 66.** *When $k \geq 1$, we have*

$$u_{2i}(x) \geq u_{2k}(x) \text{ for all } i \in \{1, 2, \cdots, k\},$$

*and*

$$\frac{u_{2(k+m)}(x)}{u_{2k}(x)} \geq \frac{1}{\left(2 + \frac{2m}{k}\right)^2} \text{ for all } m \in \mathbb{Z}_{\geq 0}.$$

*Proof.* By Lemma 17, we have $u_0(x) = 0$, $u_1(x) = \frac{1}{4}$, and for all $k \geq 1$, we have

$$u_{2k}(x) = \frac{1}{2\pi} \frac{(2k-2)!}{((k-1)!)^2} \frac{4}{2k-1} \frac{1}{2^{2k}}, \tag{111}$$

$$u_{2k+1}(x) = 0. \tag{112}$$

(We also plot the curve of $u_k(x)$ with respect to $k$ in Fig. 11, so that we can observe the general trend that is consistent with the statement of this lemma. We continue with the precise proof of this lemma below.)

By Eq. (111), we have

$$\frac{u_{2k+2}(x)}{u_{2k}(x)} = \frac{2k(2k-1)(2k-1)}{k^2(2k+1)} \cdot \frac{1}{4} = \frac{(2k-1)^2}{2k \cdot (2k+1)}. \tag{113}$$

Because $\frac{(2k-1)^2}{2k \cdot (2k+1)} \leq 1$, we know that $u_{2k}(x)$ is monotone decreasing with respect to $k$. Therefore, we have

$$u_{2i}(x) \geq u_{2k}(x) \text{ for all } i \in \{1, 2, \cdots, k\}.$$

Iterating Eq. (113), we have

$$\begin{aligned}
\frac{u_{2(k+m)}(x)}{u_{2k}(x)} &= \prod_{i=0}^{m-1} \frac{(2k+2i-1)^2}{(2k+2i)(2k+2i+1)} \\
&\geq \prod_{i=0}^{m-1} \frac{(2k+2i-1)^2}{(2k+2i+1)^2} \\
&= \frac{(2k-1)^2}{(2k+2m-1)^2} \\
&\geq \frac{k^2}{(2k+2m)^2} \quad \text{(because } 2k-1 \geq k \text{ due to } k \geq 1) \\
&= \frac{1}{\left(2+\frac{2m}{k}\right)^2}.
\end{aligned}$$

$\square$

**Lemma 67.** *Under Condition 2, for any $j = 1, 2, \cdots$, we must have*

$$\frac{u_j(K(x))}{u_{2\lceil j/2 \rceil}(x)} \geq \left(\frac{1}{2} \cdot \frac{1-a_0}{3-a_0}\right)^2 \frac{1-\left(\frac{1+a_0}{2}\right)^2}{2-\left(\frac{1+a_0}{2}\right)^2} e_j.$$

*Proof.* Consider $i^*$ defined in Lemma 65, i.e., $i^* = \left\lceil \frac{1+a_0}{2(1-a_0)} j \right\rceil$. Let $k = \left\lceil \frac{j}{2} \right\rceil$ and $m = i^* - k$. We have

$$\begin{aligned}
\frac{2m}{k} = \frac{2i^*}{k} - 2 &= \frac{2\left\lceil \frac{1+a_0}{2(1-a_0)} j \right\rceil}{\left\lceil \frac{j}{2} \right\rceil} - 2 \\
&\leq \frac{2\left(\frac{1+a_0}{2(1-a_0)} j + 1\right)}{\frac{j}{2}} - 2 \quad \text{(because } \lceil \alpha \rceil \in [\alpha, \alpha+1]) \\
&= \frac{2(1+a_0)}{1-a_0} + \frac{4}{j} - 2 \\
&\leq \frac{2(1+a_0)}{1-a_0} + 4 - 2 \quad \text{(because } j \geq 1) \\
&= \frac{4}{1-a_0}.
\end{aligned}$$

By Lemma 66, we then have

$$\frac{u_{2i^*}(x)}{u_{2\lceil j/2\rceil}(x)} = \frac{u_{2(k+m)}(x)}{u_{2k}(x)} \geq \frac{1}{\left(2+\frac{2m}{k}\right)^2} \geq \frac{1}{\left(2+\frac{4}{1-a_0}\right)^2} = \left(\frac{1}{2}\cdot\frac{1-a_0}{3-a_0}\right)^2.$$

By the first part of Lemma 66, $u_{2i} \geq u_{2i^*}$ for all $i = 1, 2, \cdots, i^*$. We thus have

$$\frac{u_{2i}(x)}{u_{2\lceil j/2\rceil}(x)} \geq \left(\frac{1}{2}\cdot\frac{1-a_0}{3-a_0}\right)^2 \text{ for all } i = 1, 2, \cdots, i^*. \tag{114}$$

Notice that

$$K(x)\frac{\pi - \arccos\left(K(x)\right)}{2\pi}$$

$$=u_1(x)K(x) + \sum_{i=1}^{\infty} u_{2i}(x)\left(K(x)\right)^{2i}$$

$$=u_1(x)K(x) + \sum_{i=1}^{\infty} u_{2i}(x)\sum_{j=0}^{\infty} d_{2i,j}x^j \text{ (by Lemma 27)}$$

$$=\frac{1}{4}K(x) + \sum_{j=0}^{\infty}\left(\sum_{i=\lceil\frac{j}{2s}\rceil}^{\infty} u_{2i}(x)\cdot d_{2i,j}\right)x^j \text{ (since } d_{2i,j} = 0 \text{ when } j > 2i\cdot s).$$

Therefore,

$$u_j\left(K(x)\right) = \frac{a_j}{4} + \sum_{i=\lceil\frac{j}{2s}\rceil}^{\infty} u_{2i}(x)\cdot d_{2i,j} \text{ for all } j \in \mathbb{Z}_{\geq 0}. \tag{115}$$

By Eq. (115), we thus have

$$\frac{u_j\left(K(x)\right)}{u_{2\lceil j/2\rceil}(x)} \geq \frac{1}{u_{2\lceil j/2\rceil}(x)}\sum_{i=\lceil\frac{j}{2s}\rceil}^{\infty} u_{2i}(x)d_{2i,j} \text{ (notice that } a_i \geq 0 \text{ for all } i \in \mathbb{Z}_{\geq 0} \text{ by Condition 2)}$$

$$\geq \frac{1}{u_{2\lceil j/2\rceil}(x)}\sum_{i=\lceil\frac{j}{2s}\rceil}^{i^*} u_{2i}(x)d_{2i,j}$$

$$\geq \left(\frac{1}{2}\cdot\frac{1-a_0}{3-a_0}\right)^2\sum_{i=\lceil\frac{j}{2s}\rceil}^{i^*} d_{2i,j} \text{ (by Eq. (114))}$$

$$\geq \left(\frac{1}{2}\cdot\frac{1-a_0}{3-a_0}\right)^2\frac{1-\left(\frac{1+a_0}{2}\right)^2}{2-\left(\frac{1+a_0}{2}\right)^2}\sum_{i=\lceil\frac{j}{2s}\rceil}^{\infty} d_{2i,j} \text{ (by Lemma 65)}$$

$$= \left(\frac{1}{2}\cdot\frac{1-a_0}{3-a_0}\right)^2\frac{1-\left(\frac{1+a_0}{2}\right)^2}{2-\left(\frac{1+a_0}{2}\right)^2}e_j \text{ (by Eq. (107))}.$$

$\square$

By Lemma 67 and Lemma 62, we can conclude the lower bound in Lemma 60. Next we will prove the upper bound in Lemma 60.

**Lemma 68.** *Under Condition 2, for any* $j = 1, 2, \cdots$, *we must have*

$$\frac{u_j\left(K(x)\right) - \frac{a_j}{4}}{u_{2\lceil j/2\rceil}(x)} \leq (2+2s)^2 e_j.$$

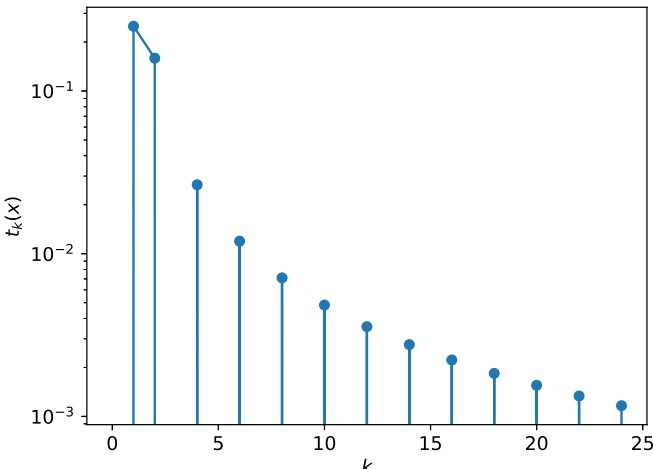

Figure 11: The curve of $u_k(x)$ with respect to $k$. Notice that $u_k(x) = 0$ when $k = 0, 3, 5, 7, \cdots$.

*Proof.* Let $k = \left\lceil \frac{j}{2s} \right\rceil$ and $m = \left\lceil \frac{j}{2} \right\rceil - \left\lceil \frac{j}{2s} \right\rceil$. Thus, we have

$$\frac{m}{k} = \frac{\left\lceil \frac{j}{2} \right\rceil}{\left\lceil \frac{j}{2s} \right\rceil} - 1 \leq \frac{\frac{j}{2} + 1}{\left\lceil \frac{j}{2s} \right\rceil} - 1 = \frac{\frac{j}{2}}{\left\lceil \frac{j}{2s} \right\rceil} + \left( \frac{1}{\left\lceil \frac{j}{2s} \right\rceil} - 1 \right) \leq s$$

$$\text{(noting that } \left\lceil \frac{j}{2s} \right\rceil \geq 1 \text{ since } j \geq 1).$$

By Lemma 66, we then have

$$\frac{u_{2\lceil j/2 \rceil}(x)}{u_{2\lceil j/(2s) \rceil}(x)} = \frac{u_{2(k+m)}(x)}{u_{2k}(x)} \geq \frac{1}{\left( 2 + \frac{2m}{k} \right)^2} \text{ (by Lemma 66)}$$

$$\geq \frac{1}{(2 + 2s)^2}.$$

By the first part of Lemma 66, we further have

$$u_{2i}(x) \leq (2 + 2s)^2 u_{2\lceil j/2 \rceil}(x) \text{ for all } 2i \geq 2 \left\lceil \frac{j}{2s} \right\rceil.$$

Thus, we have

$$u_j(K(x)) - \frac{a_j}{4} = \sum_{i=\left\lceil \frac{j}{2s} \right\rceil}^{\infty} u_{2i}(x) \cdot d_{2i,j} \text{ (by Eq. (115))}$$

$$\leq (2 + 2s)^2 u_{2\lceil j/2 \rceil}(x) \sum_{i=\left\lceil \frac{j}{2s} \right\rceil}^{\infty} d_{2i,j}$$

$$= (2 + 2s)^2 u_{2\lceil j/2 \rceil}(x) e_j \text{ (by Eq. (107))}.$$

The result of the lemma thus follows. $\qquad \square$

Combining Lemma 67, Lemma 68, and Lemma 62, the result of Lemma 60 thus follows.

## M.3    Expansion for an infinite-degree polynomial

We now return to the proof of the third statement of Lemma 51, where the polynomial $K(x)$ has infinite terms. We inherit notations $a_i$, $b_i$, $u_i$, $d_i$, $e_i$ for the finite polynomial case in Appendix M.2. Further, we introduce some additional notations as follows.

Define
$$\tilde{b}_i := \frac{b_i}{1 - b_0}, \text{ for all } i \in \mathbb{Z}_{\geq 0}, \tag{116}$$

and
$$T_k := \sum_{i=k+1}^{\infty} \tilde{b}_i, \quad k \in \mathbb{Z}_{\geq 0}. \tag{117}$$

By Condition 2, Eq. (103), and Eq. (116), we have
$$T_k \geq 0 \text{ for all } k \in \mathbb{Z}_{\geq 0}. \tag{118}$$

Define for any $k \in \{1, 2, \cdots\}$,
$$L_k := \min \left\{ e_j \mid j = 0, 1, \cdots, k-1 \right\} \cup \{1\} - \sum_{i=k}^{\infty} T_i. \tag{119}$$

**Lemma 69.** *For any $i \in \mathbb{Z}_{\geq 0}$ and any $k \in \{1, 2, \cdots\}$, we must have $e_i \geq L_k$. Notice that the indices $i$ and $k$ are not required to be equal.*

We prove Lemma 69 in Appendix M.3.1.

In order to prove the third statement of Lemma 51, by Lemma 52, we only need to lower bound $\frac{u_j(K(x))}{u_{2\lceil j/2 \rceil}(x)}$. Notice that Lemma 67 still holds when $s \to \infty$ (the proof of it will be exactly the same after replacing $\lceil j/2s \rceil$ by zero). Therefore, we only need to prove that $e_i$ is lower bounded by a positive constant. By Lemma 69, if we can find $L_k > 0$ for some $k$, then we are done. By Eq. (119), In order to calculate the exact value of $L_k$, we need to find a way to calculate the exact value of $\sum_{k=0}^{\infty} T_k$, which is provided by the following lemma.

**Lemma 70.** $\sum\limits_{k=0}^{\infty} T_k = \frac{2}{1-a_0^2} \frac{\partial K(x)}{\partial x}\Big|_{x=1}.$

*Proof.* We have
$$\begin{aligned}
\sum_{k=0}^{\infty} T_k &= \sum_{k=0}^{\infty} \sum_{i=k+1}^{\infty} \tilde{b}_i \text{ (by Eq. (117))} \\
&= \sum_{i=1}^{\infty} i \cdot \tilde{b}_i \\
&= \frac{1}{1-b_0} \sum_{i=1}^{\infty} i \cdot b_i \text{ (by Eq. (116))} \\
&= \frac{1}{1-a_0^2} \sum_{i=1}^{\infty} i \cdot b_i \text{ (notice that } b_0 = a_0^2 \text{ by Eq. (103))} \\
&= \frac{1}{1-a_0^2} \frac{\partial \left( \sum_{i=0}^{\infty} b_i x^i \right)}{\partial x} \Big|_{x=1} \\
&= \frac{1}{1-a_0^2} \frac{\partial \left( K(x) \right)^2}{\partial x} \Big|_{x=1} \text{ (by Eq. (103))} \\
&= \frac{2K(1)}{1-a_0^2} \frac{\partial K(x)}{\partial x} \Big|_{x=1} \\
&= \frac{2}{1-a_0^2} \frac{\partial K(x)}{\partial x} \Big|_{x=1} \text{ (by Condition 2).}
\end{aligned}$$
$\square$

Now we consider the case of 3-layer without bias, i.e., the case when the polynomial $K(x)$ is $2d \cdot K^{\text{RF}}(x) = \frac{\sqrt{1-x^2}+(\pi-\arccos(x))x}{\pi}$. After calculation (details in Appendix M.3.2), we have $L_3 \approx 0.069 > 0$, which completes the proof of Proposition 47.

### M.3.1 Proof of Lemma 69

We first prove some useful lemmas.

**Lemma 71.** *(i)* $\sum_{i=1}^{\infty} \tilde{b}_i = 1$. *(ii)* $e_k = \sum_{i=0}^{k-1} \tilde{b}_{k-i} e_i$ *for all* $k \in \{1, 2, \cdots\}$. *(iii)* $e_k \leq e_0 = \frac{1}{1-b_0}$ *for all* $k \in \mathbb{Z}_{\geq 0}$.

*Proof.* Note that

$$\sum_{i=1}^{\infty} \tilde{b}_i = \frac{\sum_{i=1}^{\infty} b_i}{1 - b_0} \text{ (by Eq. (116))}$$

$$= 1 \quad \text{(since } \sum_{i=1}^{\infty} b_i = 1 - b_0 \text{ because } b_0 + \sum_{i=1}^{\infty} b_i = (K(1))^2 = 1\text{).}$$

Because

$$\frac{1}{1 - (K(x))^2} = 1 + (K(x))^2 \cdot \frac{1}{1 - (K(x))^2},$$

we have

$$\sum_{j=0}^{\infty} e_j x^j = 1 + \left(\sum_{i=0}^{\infty} a_i x^i\right)^2 \cdot \left(\sum_{j=0}^{\infty} e_j x^j\right) = 1 + \left(\sum_{i=0}^{\infty} b_i x^i\right) \cdot \left(\sum_{j=0}^{\infty} e_j x^j\right).$$

Comparing the coefficient of $x^j$ on both sides, we have

$$e_0 = 1 + b_0 e_0,$$
$$e_1 = b_1 e_0 + b_0 e_1,$$
$$e_2 = b_2 e_0 + b_1 e_1 + b_0 e_2,$$
$$\vdots$$

We thus have

$$
\begin{cases}
e_0 = \frac{1}{1-b_0}, \\
e_1 = \tilde{b}_1 e_0, \\
e_2 = \tilde{b}_2 e_0 + \tilde{b}_1 e_1 = \left(\tilde{b}_2 + \tilde{b}_1^2\right) e_0, \\
e_3 = \tilde{b}_3 e_0 + \tilde{b}_2 e_1 + \tilde{b}_1 e_2 = \left(\tilde{b}_3 + 2\tilde{b}_2\tilde{b}_1 + \tilde{b}_1^3\right) e_0, \\
e_4 = \tilde{b}_4 e_0 + \tilde{b}_3 e_1 + \tilde{b}_2 e_2 + \tilde{b}_1 e_3 = \left(\tilde{b}_4 + 2\tilde{b}_1\tilde{b}_3 + \tilde{b}_2^2 + 3\tilde{b}_1^2\tilde{b}_2 + \tilde{b}_1^4\right) e_0, \\
\vdots \\
e_k = \sum_{i=0}^{k-1} \tilde{b}_{k-i} e_i, \\
\vdots
\end{cases}
\tag{120}
$$

We now prove that $e_k \leq e_0$ for all $k \in \mathbb{Z}_{\geq 0}$ by mathematical induction. Suppose that for all $i \leq k$, we already have $e_i \leq e_0$ (which is obviously true when $k = 0$). Thus, we have

$$e_{k+1} = \sum_{i=0}^{k} \tilde{b}_{k+1-i} e_i \leq e_0 \sum_{i=0}^{k} \tilde{b}_{k+1-i} \leq e_0 \sum_{i=0}^{\infty} \tilde{b}_i = e_0.$$

By mathematical induction, we thus have $e_i \leq e_0$ for all $i \in \mathbb{Z}_{\geq 0}$. $\square$

For any given real number sequence $\boldsymbol{\Delta} := (\Delta_1, \Delta_2, \cdots)$, we define a sequence $(e_0^{\boldsymbol{\Delta}}, e_1^{\boldsymbol{\Delta}}, e_2^{\boldsymbol{\Delta}}, \cdots)$ by

$$e_0^{\boldsymbol{\Delta}} := e_0, \quad e_k^{\boldsymbol{\Delta}} := \Delta_k + \sum_{i=0}^{k-1} \tilde{b}_{k-i} e_i^{\boldsymbol{\Delta}}, \ k \in \{1, 2, \cdots\}. \tag{121}$$

**Lemma 72.** *For any $k \in \{1, 2, \cdots\}$ and any $\boldsymbol{\Delta}$, we must have*

$$e_k^{\boldsymbol{\Delta}} - e_k = \sum_{i=1}^{k} \Delta_i \cdot \frac{e_{k-i}}{e_0}. \tag{122}$$

*Proof.* We prove Eq. (122) by mathematical induction. When $k = 1$, we have $e_1^{\boldsymbol{\Delta}} = \Delta_1 + \tilde{b}_1 e_0$ (by Eq. (121)) and $e_1 = \tilde{b}_1 e_0$ (by Lemma 71). Thus, Eq. (113) holds when $k = 1$. Suppose that for all $k \in \{1, 2, \cdots, l\}$, Eq. (122) holds. We thus have

$$\begin{aligned}
e_{l+1}^{\boldsymbol{\Delta}} - e_{l+1} &= \Delta_{l+1} + \sum_{i=0}^{l} \tilde{b}_{l+1-i}(e_i^{\boldsymbol{\Delta}} - e_i) \text{ (by Eq. (121) and Lemma 71)} \\
&= \Delta_{l+1} + \sum_{i=1}^{l} \tilde{b}_{l+1-i}(e_i^{\boldsymbol{\Delta}} - e_i) \text{ (notice that } e_0^{\boldsymbol{\Delta}} = e_0) \\
&= \Delta_{l+1} + \sum_{i=1}^{l} \tilde{b}_{l+1-i} \sum_{j=1}^{i} \Delta_j \frac{e_{i-j}}{e_0} \text{ (applying Eq. (122) by induction hypothesis).}
\end{aligned} \tag{123}$$

Notice that

$$\begin{aligned}
\sum_{i=1}^{l} \sum_{j=1}^{i} \tilde{b}_{l+1-i} \cdot \Delta_j \cdot e_{i-j} &= \sum_{j=1}^{l} \Delta_j \sum_{i=j}^{l} \tilde{b}_{l+1-i} e_{i-j} \text{ (by re-organizing terms)} \\
&= \sum_{j=1}^{l} \Delta_j \sum_{i=0}^{l-j} \tilde{b}_{l+1-i-j} e_i \text{ (replacing } i - j \text{ by } i).
\end{aligned}$$

Plugging the above equation into Eq. (123), we then have

$$\begin{aligned}
e_{l+1}^{\boldsymbol{\Delta}} - e_{l+1} &= \Delta_{l+1} + \frac{1}{e_0} \sum_{j=1}^{l} \Delta_j \sum_{i=0}^{l-j} \tilde{b}_{l-j+1-i} e_i \\
&= \Delta_{l+1} + \frac{1}{e_0} \sum_{j=1}^{l} \Delta_j \cdot e_{l+1-j} \text{ (by Lemma 71)} \\
&= \frac{1}{e_0} \sum_{j=1}^{l+1} \Delta_j \cdot e_{l+1-j},
\end{aligned}$$

i.e., Eq. (122) also holds for $k = l+1$. Thus, the mathematical induction is completed and the result of this lemma thus follows. $\square$

Now we are ready to prove Lemma 69.

*Proof of Lemma 69.* Let

$$\Delta_i = \begin{cases} 0, & \text{if } i < k, \\ T_i, & \text{if } i \geq k. \end{cases}$$

We first prove by mathematical induction that

$$e_i^{\boldsymbol{\Delta}} \geq \min\{e_j \mid j = 0, 1, \cdots, k-1\} \cup \{1\} \text{ for all } i \in \mathbb{Z}_{\geq 0}. \tag{124}$$

Towards this end, note that because $\Delta_i = 0$ for all $i < k$, we know from Lemma 72 that $e_i^{\boldsymbol{\Delta}} = e_i$. Hence, Eq. (124) trivially holds for all $i \in \{0, 1, \cdots, k-1\}$. Suppose that Eq. (124) holds for all $i \leq l \in \mathbb{Z}_{\geq 0}$, where $l \geq k-1$ denotes the index of the induction hypothesis. In order to finish the

mathematical induction, we only need to prove that Eq. (124) holds for $i = l + 1$. To this end, we have

$$
\begin{aligned}
e_{l+1}^{\Delta} &= T_{l+1} + \sum_{i=0}^{l} \tilde{b}_{l+1-i} e_i^{\Delta} \text{ (by Eq. (121))} \\
&= \sum_{i=l+2}^{\infty} \tilde{b}_i + \sum_{i=1}^{l+1} \tilde{b}_i e_{l+1-i}^{\Delta} \text{ (by Eq. (117))} \\
&\geq \left( \min\{e_j^{\Delta} \mid j = 0, 1, \cdots, l\} \cup \{1\} \right) \cdot \sum_{i=1}^{\infty} \tilde{b}_i \\
&= \min\{e_j^{\Delta} \mid j = 0, 1, \cdots, l\} \cup \{1\} \text{ (by Lemma 71)} \\
&\geq \min\{e_j \mid j = 0, 1, \cdots, k-1\} \cup \{1\} \text{ (by induction hypothesis).}
\end{aligned}
$$

Thus, Eq. (124) holds by mathematical induction. We thus have

$$
\begin{aligned}
e_i &= e_i^{\Delta} - \sum_{j=1}^{i} \Delta_j \cdot \frac{e_{i-j}}{e_0} \text{ (by Lemma 72)} \\
&\geq e_i^{\Delta} - \sum_{j=1}^{i} \Delta_j \text{ (since } e_{i-j} \leq e_0 \text{ by Lemma 71)} \\
&= e_i^{\Delta} - \sum_{j=k}^{i} T_i \\
&\geq \min\{e_j \mid j = 0, 1, \cdots, k-1\} \cup \{1\} - \sum_{j=k}^{\infty} T_j \text{ (by Eq. (124) and Eq. (118))} \\
&= L_k.
\end{aligned}
$$

The result of this lemma thus follows. $\qquad\square$

### M.3.2 Calculate $L_3$ for 3-layer without bias

Coefficients of Taylor expansion of $K(x) = 2d \cdot K^{\mathrm{RF}}(x) = \frac{\sqrt{1-x^2} + (\pi - \arccos(x))x}{\pi}$ can be derived from Lemma 17, i.e.,

$$
K(x) = \frac{1}{\pi} \left( 1 + \frac{\pi}{2} x + \sum_{k=0}^{\infty} \frac{2(2k)!}{(k+1)(2k+1)(k!)^2} \left( \frac{x}{2} \right)^{2k+2} \right). \tag{125}
$$

By Eq. (125), We can calculate values of $a_i$, $b_i$, and $\tilde{b}_i$ for $i = 0, 1, 2$ by their definitions.

$$
\begin{aligned}
&a_0 = \frac{1}{\pi}, \ a_1 = \frac{1}{2}, \ a_2 = \frac{1}{2\pi}, \\
&b_0 = a_0^2 = \frac{1}{\pi^2}, \ b_1 = 2a_0 a_1 = \frac{1}{\pi}, \ b_2 = 2a_2 a_0 + a_1^2 = \frac{1}{\pi^2} + \frac{1}{4}, \\
&\tilde{b}_0 = \frac{1}{\pi^2 - 1}, \ \tilde{b}_1 = \frac{\pi}{\pi^2 - 1}, \ \tilde{b}_2 = \frac{1}{\pi^2 - 1} + \frac{\pi^2}{4(\pi^2 - 1)}.
\end{aligned}
$$

Then, we calculate the values of $e_i$ by Eq. (120).

$$
e_0 = \frac{\pi^2}{\pi^2 - 1} \approx 1.11, \ e_1 = \frac{\pi^3}{(\pi^2 - 1)^2} \approx 0.39, \ e_2 = \frac{(2\pi^2 - 1)\pi^2}{(\pi^2 - 1)^3} + \frac{\pi^4}{4(\pi^2 - 1)^2} \approx 0.57.
$$

Next, we calculate $\sum_{i=0}^{\infty} T_i$ by Lemma 70.

$$\sum_{i=0}^{\infty} T_i = \frac{2}{1-a_0^2} \frac{\partial \frac{\sqrt{1-x^2}+(\pi-\arccos(x))x}{\pi}}{\partial x} \bigg|_{x=1}$$

$$= \frac{2}{1-a_0^2} \frac{1}{\pi} \left( -\frac{x}{\sqrt{1-x^2}} + (\pi - \arccos(x)) + \frac{x}{\sqrt{1-x^2}} \right)$$

$$\text{(notice that } \frac{\partial \sqrt{1-x^2}}{\partial x} = -\frac{x}{\sqrt{1-x^2}} \text{ and } \frac{\partial}{\partial x} \arccos(x) = -\frac{1}{\sqrt{1-x^2}} \text{)}$$

$$= \frac{2}{1-a_0^2} \frac{\pi - \arccos(x)}{\pi} \bigg|_{x=1} = \frac{2\pi^2}{\pi^2 - 1}.$$

By Eq. (117) and Lemma 71(i), we thus have $T_j = \sum_{i=1}^{\infty} \tilde{b}_i - \sum_{i=1}^{j} \tilde{b}_i = 1 - \sum_{i=1}^{j} \tilde{b}_i$. Therefore, we have

$$T_0 = 1,\ T_1 = 1 - \frac{\pi}{\pi^2 - 1},\ T_2 = 1 - \frac{\pi}{\pi^2 - 1} - \left( \frac{1}{\pi^2 - 1} + \frac{\pi^2}{4(\pi^2 - 1)} \right).$$

Now we are ready to calculate $L_3$ by Eq. (119).

$$L_3 = e_1 + T_0 + T_1 + T_2 - \sum_{i=0}^{\infty} T_i$$

$$= \frac{\pi^3}{(\pi^2 - 1)^2} + 3 - \frac{2\pi}{\pi^2 - 1} - \left( \frac{1}{\pi^2 - 1} + \frac{\pi^2}{4(\pi^2 - 1)} \right) - \frac{2\pi^2}{\pi^2 - 1}$$

$$\approx 0.069.$$