# OpenReview forum: "On the Generalization Power of the Overfitted Three-Layer Neural Tangent Kernel Model"
_NeurIPS.cc/2022/Conference — NeurIPS 2022 Accept_

### Official Review · Reviewer_2rdZ · 2022-07-09

**Rating:** 7
**Confidence:** 3
**Soundness:** 3 good
**Presentation:** 3 good
**Contribution:** 3 good

**Summary:**

This paper studies the generalization performance of overparameterized 3-layer NTK models. A learnable set of ground-truth functions is defined for characterizing the generalization error with regard to the number of samples and the number of two hidden layers. The results provide insights into which hidden layer plays a more important role and how the bias affects the size of the learnable set. Some interesting comparisons are made between 3-layer NTK and 2-layer NTK. Overall the paper is good, especially in its excellent theoretical work, but I still have some concerns. I am willing to increase the score if the responses can solve my questions.

**Questions:**

Q1: I would like to know more about the comparison between this paper and the papers I mentioned in Weakness 1. What does NTK with/without overfitting mean?

Q2: Is the trained model irrelevant to the step size and the number of iterations? If so, why is that?

**Limitations:**

The authors adequately addressed the limitations and potential negative societal impact of their work.

**Strengths And Weaknesses:**

Strengths:
1. The insights are interesting, especially the discussion about how the bias affects the size of the learnable set. The results are also verified by the experiments.
2. The proof seems to be solid.
3. From my understanding and my knowledge about NTK, the assumptions in this paper are mild and the size of the learnable set is large.

Weaknesses:
1. Although the experiment verifies this, the conclusion that the second hidden layer is more important than the first hidden layer seems weak since only the second hidden layer is trained. More theoretical explanations are required. In line 164, it states that it is to "avoid violating NTK assumption" to do so, but papers such as "Learning and generalization in overparameterized neural networks, going beyond two layers" by Z. Allen-Zhu et al. (Neurips 2019) and "Generalization Guarantee of Training Graph Convolutional Networks with Graph Topology Sampling" by H. Li et al. (ICML 2022) consider training both two hidden layers in 3-layer NTK. Some discussions and comparisons might be needed here with regard to these papers.
2. Some parts of writing can be improved. For example, from equations 1 to 3, I think the authors mainly want to give a result of the trained model. I think it is better to make this part a subsection to let people know what this part is intended to cover. Otherwise, readers may feel strange and difficult to follow because the authors suddenly start to derive many equations when introducing the problem formulations and the setup.

---

> ### Author Response · Authors · 2022-08-02
> **Response to Reviewer 2rdZ**
>
> Thanks for your comments. We address your questions as follows.
>
> ---
>
> > *Although the experiment verifies this, the conclusion that the second hidden layer is more important than the first hidden layer seems weak since only the second hidden layer is trained. More theoretical explanations are required.*
>
> We agree that our current analysis is for the specific model where middle layer is trained. However, even in this setting, the result that $p\_2$ is more important than $p\_1$ is not trivial/obvious. The reason is that the number of weights in the middle layer (i.e., its width) is determined by the product of $p\_1$ and $p\_2$ (as it is the layer connected the first hidden-layer and the second hidden-layer, see Fig. 1). Therefore, without our analysis, it is not easy to tell whether $p\_1$ or $p\_2$ is more important even if only the middle layer is trained.
>
> It would indeed be interesting and important to explore whether similar implications hold for other model setups. Nonetheless, to the best of our knowledge, our work is still the first to study this implication for a 3-layer model. Thanks for pointing out the connection between our claims and the choice of model. We will add discussion in the final version of the paper.
>
> ---
>
> > *In line 164, it states that it is to "avoid violating NTK assumption" to do so, but papers such as "Learning and generalization in overparameterized neural networks, going beyond two layers" by Z. Allen-Zhu et al. (Neurips 2019) and "Generalization Guarantee of Training Graph Convolutional Networks with Graph Topology Sampling" by H. Li et al. (ICML 2022) consider training both two hidden layers in 3-layer NTK. Some discussions and comparisons might be needed herewith regard to these papers.*
>
> > *I would like to know more about the comparison between this paper and the papers I mentioned in Weakness 1. What does NTK with/without overfitting mean?*
>
> One major difference is that we considered the overfitted solution without any explicit regularization. Note that overfitting means that the training error for the solution is exactly zero. In contrast, if one uses GD/SGD with regularization, uses early stopping, or uses new training sample in every iteration, then the training error will not be driven to zero (i.e., without overfitting). We note that the work [Z. Allen-Zhu et al 19] provides the training dynamics. However, the difference is that they do not consider the overfitted solution (since they use regularization in training). Hence, their solution cannot be written in an explicit form as ours. This difference also applies to the comparison with [H. Li et al '22]. Besides, [H. Li et al '22] focus on the Graph Convolutional Networks for classification, while ours focus on fully connected network on regression.
>
> ---
>
> > *Some parts of writing can be improved. For example, from equations 1 to 3, I think the authors mainly want to give a result of the trained model. I think it is better to make this part a subsection to let people know what this part is intended to cover. Otherwise, readers may feel strange and difficult to follow because the authors suddenly start to derive many equations when introducing the problem formulations and the setup.*
>
> Thanks for the suggestion. In the revision, we will add a subsection as follows.
>
> **Overfitted NTK solution**
>
> In this subsection, we will derive the overfitted solution for this 3-layer neural network using the NTK approximation. ...
>
> ---
>
> > *Is the trained model irrelevant to the step size and the number of iterations? If so, why is that?*
>
> We do not need to be concerned about the training dynamics because we consider the overfitted solution, whose expression can be written down exactly in Eq. (3). Generally speaking, if we do gradient descent training on a linear model from zero initial point until the training error is zero (i.e., overfitting), then the solution will be exactly the min $\ell\_2$-norm solution. (This fact is also mentioned in the literature, e.g., [Satpathi et al 2021].) To see this, consider a simple example that uses gradient descent to minimize the training error $\lVert y-Xa\rVert\_2^2$, where each row of $X$ is one training sample. Because the initial point is zero, after each iteration/update the solution vector $a$ must always be in the row space of $X$ (since the gradient is in the row space of $X$), i.e., $a=X^Tb$ for some vector $b$. Therefore, when overfitting occurs (i.e., training error becomes zero), we have $y=Xa=XX^Tb$, which implies that $b=(X^TX)^{-1}y$ and thus $a=Xb=X^T(XX^T)^{-1}y=X^{\dagger}y$. As we can see, this is exactly the min $\ell\_2$-norm solution.
>
> [Satpathi, Siddhartha, and Rayadurgam Srikant. "The Dynamics of Gradient Descent for Overparametrized Neural Networks." Learning for Dynamics and Control. PMLR, 2021.]

---

> > ### Comment · Reviewer_2rdZ · 2022-08-02
> > **Response to the authors**
> >
> > Thank the authors for the response. It has solved my major concerns. I have increased my score to 7 due to the solid theoretical analysis and contributions. I may have two more suggestions.
> >
> > 1. In the revision version, I suggest that the authors add the above discussion in a brief way, especially for the comparison with non-overfitted NTK, including but not limited to those references provided by me and the authors. This is because I think it is important to clarify the notions. There is a line of research that studies non-overfitted NTK.
> >
> > 2. Although I feel that this paper has a good technical contribution. I still think it is better to briefly summarize the technical contributions (maybe at the end of the Introduction) to make this paper clear to readers.

---

> > > ### Author Response · Authors · 2022-08-03
> > > **Response to Reviewer 2rdZ**
> > >
> > > Thank you for raising the score and providing those valuable suggestions. We will add the above discussion on the comparison with non-overfitted NTK, and summarize our technical contribution in the final version.

---

### Official Review · Reviewer_6CH3 · 2022-07-09

**Rating:** 7
**Confidence:** 3
**Soundness:** 4 excellent
**Presentation:** 4 excellent
**Contribution:** 3 good

**Summary:**

This paper studies the generalization performance of a neural network with two hidden layers when only the intermediate hidden layer is trained under the NTK assumption (Trained weights are not too far from initialized weights and the hidden layer activation pattern does not change). They show that the the position of hidden layer is crucial for the generalization performance by comparing existing results of 2-NTK with their 3-NTK model.

**Questions:**

Regarding lower bounds of the eigenvalue of $HH^T$: I see that it depends on $\mathcal{J}$. When is this bound useful? I see the discussion about it when $p_1p_2 \to \infty$. In a quantitative sense, how big should $p_1, p_2$ be? How would that compare to your experiments in Fig 2?

Also, can you relate these experiments to the effect of noise?

**Ethics Review Area:**

["I don’t know"]

**Strengths And Weaknesses:**

The paper offers fresh and quantitative insights on the effect of the hidden position of hidden layers on the performance of the neural network. While there is a long list of works exclusively looking at single hidden layer networks, the effect of the intermediate hidden layer has not been well studied in the literature.

The effect of noise is somewhat counterintuitive. As of now, it seems that generalization worsens as the number of samples increases in presence of noise. This might suggest that upper bound on generalization has some room for improvement.

---

> ### Author Response · Authors · 2022-08-02
> **Response to Reviewer 6CH3**
>
> Thanks for your positive feedback. We address your questions as below.
>
> ---
>
> > *The effect of noise is somewhat counter intuitive. As of now, it seems that generalization worsens as the number of samples increases in presence of noise. This might suggest that upper bound on generalization has some room for improvement.*
>
> For the overfitted solutions for linear models, it is not uncommon that the noise effect increases when $n$ becomes larger (see, e.g., [Muthukumar et al '20]). Our experiments shows consistent behavior, i.e., Fig. 8(a) in Appendix D shows that (in the case of finite $p\_1$ and $p\_2$) the noise effect increases when the number of samples increases. An intuitive explanation is that the overfitted solution needs to completely fit the noise component of all training samples, and thus more data also means more noise to fit (and thus contributes to larger model error).
> Nonetheless, we do agree that upper bound on generalization error may have some room for improvement for the noise effect part, as stated in Line 576.
>
> [V. Muthukumar, K. Vodrahalli, V. Subramanian and A. Sahai, "Harmless Interpolation of Noisy Data in Regression," in IEEE Journal on Selected Areas in Information Theory, vol. 1, no. 1, pp. 67-83, May 2020, doi: 10.1109/JSAIT.2020.2984716.]
>
> ---
>
> > *Regarding lower bounds of the eigenvalue of $\mathbf{HH^T}$, I see that it depends on $J$. When is this bound useful? I see the discussion about it when $p\_1,p\_2\to\infty$. In a quantitative sense, how big should be? How would that compare to your experiments in Fig 2?*
>
> In Condition 1 (Line 528, Appendix C), we do provide a quantitative characterization on how large $p\_1$ and $p\_2$ should be to make our upper bound useful. Notice that such a condition is for high probability estimation and may be conservative in reality. Thus, our experiments use $p\_1$ and $p\_2$ without this restriction, and they in fact show consistent decent behavior even for $p\_1$ and $p\_2$ outside this range (as can be observed in Fig 2).
>
> ---
>
> > *Also, can you relate these experiments to the effect of noise?*
>
> Our experiments show consistent behavior, i.e., Fig. 8(a) in Appendix D shows that (in the case of finite $p\_1$ and $p\_2$) the noise effect increases when the number of samples increases. We have also discussed other aspects of noise effect in detail in Appendix D.

---

> > ### Comment · Reviewer_6CH3 · 2022-08-06
> > **Response to the authors**
> >
> > I thank the authors for their response. It has resolved some of my major questions. I will stick to my score.

---

### Official Review · Reviewer_RBFS · 2022-07-11

**Rating:** 6
**Confidence:** 4
**Soundness:** 2 fair
**Presentation:** 3 good
**Contribution:** 2 fair

**Summary:**

In this work, the authors investigate three-layer ReLU networks, in the regression setting to shed greater light on the behavior of practical DNNs, by capturing the hidden layer interactions. Specifically, the work focuses on determining how the (hidden) layer interactions influence the phenomenon of _double-descent_ and consequently investigate any improvement in the generalization performance with the increase in depth over a similar two-layer model.

The authors showcase that in the case of having only the middle-layer trained, the test error of the setup in concern is upper-bounded by the sample complexity $n$, and the hidden layer widths $p_1$ and $p_2$ and analyze the relevance of these entities in maximally affecting the test performance.

In this particular effort, the work demonstrates that while equivalent two-layer networks (with bias), possess the same expressiveness as the three-layer network (without bias) in focus, its generalization performance is dependent on the bias choice itself. Following this demonstration, the authors then subsequently put forth a novel bias choice in terms of the input vector norm which can alleviate the robustness issue of the two-layer model.

**Questions:**

- In Line 141, how do the authors justify the assumption that the movement of the middle-layer weights during training, is indeed infinitesimal given the network's weights do _not_ $\to\infty$ and the model parameterization is not accordingly chosen to be in the lazy regime

- While understood that it was performed for the sake of mathematical simplicity, the authors assume a uniform distribution of the inputs, bottom, and middle layer weights. Can the authors provide a reasoning and qualitative formulation of what might happen for the more standard Gaussian assumption?

- While stated in **Definition 1** to be the **learnable set of 3-layer NTK**, it is not totally evident that the stated set encompasses the entire learnable function space of the chosen model

- In Theorem 1, the authors claim that the result holds with a **high probability**? Can they provide a formal estimation of the probability in question?

- Can the authors explain the observation of essentially zero reduction in the test error for $\sigma^2 = 0.1, 0.4$ in Fig 2 (b)?

**Limitations:**

Please see **Questions** and **Weaknesses**

**Strengths And Weaknesses:**

**Strengths**:
- This work takes an important step toward gaining an understanding of the behavior of DNNs by theoretically analyzing three layer NN models

- It provides a formal bound, albeit under specific input data & fixed layer weights' distribution choice, on the test performance of the concerned three-layer model in terms of the training set size and the widths of the hidden layers, and consequently concludes that the width of the 2$^\text{nd}$ hidden-layer plays the greatest role in the reduction of the test error within the chosen setup

- While possibly indicative due to the presence of the extra-layer, it establishes that the three-layer network _without_ a bias term and with only the middle-layer trained is at least as expressive as its two-layer _biased_ counterpart and strictly more expressive as the two-layer unbiased one

**Weaknesses**:
- It is unclear to me why the authors chose to use the **NTK** terminology in their title and the paper itself for classifying the operational regime of their network given the fact that the hidden-layer widths of their model do not $\to\infty$ as what is strictly required to be in the NTK regime

- While it is acknowledged that a primary objective for the authors has been to understand the concerned network setup in the _practical_ finite-width regime, a truer to the title endeavor would have been to study and compare the NT Kernel for the two models in the asymptotic regime

- It appears that the setup to train the middle layer only was chosen on the seemingly weak premise stated in _Remark 1_, rather than through a formal argument based upon _lazy_ parameterization or taking the widths $\to\infty$

- Variable definitions seem to be incomplete. For e.g. in Eq. (7), $z$ and $g$ seem to be not defined. Same for the caption in Figure 2.

- One of the main claims by the authors is that the middle layer width plays a more central role in decreasing the model test error based upon their upper bound result. It is not justified why such behavior cannot be an artifact of their very choice of the model setup i.e. the middle layer itself is trained

---

> ### Author Response · Authors · 2022-08-02
> **Response to Reviewer RBFS (Part 1)**
>
> Thanks for your comments. We address your questions as follows.
>
> ---
>
> > *It is unclear to me why the authors chose to use the NTK terminology in their title and the paper itself for classifying the operational regime of their network given the fact that the hidden-layer widths of their model do not $\to\infty$ as what is strictly required to be in the NTK regime.*
>
> > *While it is acknowledged that a primary objective for the authors has been to understand the concerned network setup in the practical finite-width regime, a truer to the title endeavor would have been to study and compare the NT Kernel for the two models in the asymptotic regime.*
>
> We agree that the initial introduction of NTK is defined for infinite width networks. However, more recently NTK with finite width has also been defined and considered in literature, e.g., see the following paper and some of its references:
>
> [Novak, Roman, Jascha Sohl-Dickstein, and Samuel S. Schoenholz. "Fast finite width neural tangent kernel." International Conference on Machine Learning. PMLR, 2022.]
>
> Further, although our focus has been on the analysis of finite width NTK, our results apply to both finite and infinite width (we can just let $p\_1,p\_2\to\infty$ in our upper bound). To avoid confusion, we plan to change the title to "finite-width 3-layer NTK models", and clarify this difference between infinite width and finite width in the final version of the paper.
>
> ---
>
> > *It appears that the setup to train the middle layer only was chosen on the seemingly weak premise stated in Remark 1, rather than through a formal argument based upon lazy parameterization or taking the widths $\to \infty$.*
>
> > *In Line 141, how do the authors justify the assumption that the movement of the middle-layer weights during training, is indeed infinitesimal given the network's weights do not $\to\infty$ and the model parameterization is not accordingly chosen to be in the lazy regime.*
>
> Since we only train the middle-layer weights, the weight-change dynamics should be similar to that of a two-layer network whose input is the (untrained) output of the first hidden-layer (which can be viewed as a preprocessing of input $x$). It has been shown in [Du et al 2018] that in a two-layer neural network, when the (finite) width is larger than a threshold and the training step size is small, the gradient descent training keeps the weights in the lazy regime, and thus the NTK model can approximate the actual neural network during training dynamics. We simply assume that similar lazy regime holds, instead of quantifying this effect separately. We then focus on the generalization performance of the finite-width NTK model in this regime.
>
> [Du, Simon S., et al. "Gradient Descent Provably Optimizes Over-parameterized Neural Networks." International Conference on Learning Representations. 2018.]
>
> ---
>
> > *Variable definitions seem to be incomplete. For e.g. in Eq. (7), $z$ and $g$ seem to be not defined. Same for the caption in Figure 2.*
>
> Sorry for the confusion! $\\boldsymbol{z}$ is a dummy variable and is part of the integral variable $d\mu(\\boldsymbol{z})$ where $\mu(\cdot)$ is defined in Line 154. $g(\cdot)$ is any real-value function of $\\boldsymbol{z}$ that has finite norm. In the caption of Figure 2, $e\_1$ denotes the first standard basis in $d$-dim vector space, i.e., $e\_1=[1\ 0\ 0\ \cdots]^T$. We will clarify this in the final version.
>
> ---
>
> > *One of the main claims by the authors is that the middle layer width plays a more central role in decreasing the model test error based upon their upper bound result. It is not justified why such behavior cannot be an artifact of their very choice of the model setup i.e. the middle layer itself is trained.*
>
> We agree that our current analysis is for the specific model where middle layer is trained. However, even in this setting, the result that $p\_2$ is more important than $p\_1$ is not trivial/obvious. The reason is that the number of weights in the middle layer (i.e., its width) is determined by the product of $p\_1$ and $p\_2$ (as it is the layer connected the first hidden-layer and the second hidden-layer, see Fig. 1). Therefore, without our analysis, it is not easy to tell whether $p\_1$ or $p\_2$ is more important even if only the middle layer is trained.
>
> It would indeed be interesting and important to explore whether similar implications hold for other model setups. Nonetheless, we believe that our work is still the first to study this implication for a 3-layer model. Thanks for pointing out the connection between our claims and the choice of model. We will add a discussion in the final version of the paper.
>
> ---
>
> **Followed by "Response to Reviewer RBFS (Part 2)".**

---

> ### Author Response · Authors · 2022-08-02
> **Response to Reviewer RBFS (Part 2)**
>
> **Continued from "Response to Reviewer RBFS (Part 1)".**
>
> ---
>
> > *While understood that it was performed for the sake of mathematical simplicity, the authors assume a uniform distribution of the inputs,bottom, and middle layer weights. Can the authors provide a reasoning and qualitative formulation of what might happen for the more standard Gaussian assumption?*
>
> We believe that replacing uniform distribution with standard Gaussian assumption will not fundamentally change the current results. The uniform distribution on a hyper-sphere is very close to the standard Gaussian distribution since both distribution has uniformly distributed direction in the hyperspace. The only difference is that the uniform distribution has an additional normalization process that makes the vector's norm/length constant. When the number of elements of the random vector $a\in \mathbb{R}^{k}$ is large, the scaled norm $\frac{\lVert a \rVert\_2}{\sqrt{k}}$ of a Gaussian distributed vector becomes more concentrated on its mean (by law of large numbers) and thus the difference between a uniform distributed vector and a Gaussian distributed vector should become negligible.
>
> ---
>
> > *While stated in Definition 1 to be the learnable set of 3-layer NTK, it is not totally evident that the stated set encompasses the entire learnable function space of the chosen model.*
>
> The learnable set is derived by the expressiveness of the corresponding NTK with infinite width (see Line 185-202). If a function is learnable, then it at least should be able to be expressed by an infinite width NTK (more neurons usually mean stronger expressiveness). Therefore, we believe those expressible functions are the only reasonable candidates of learnable functions. We then use Theorem 1 to prove that those functions can indeed be learnt. We note that similar definition of learnable sets has been used in [Ju et al '21] for 2-layer NTK without bias. Specifically, [Ju et al '21] shows that, for 2-layer NTK without bias, if a function is at a positive distance away from the learnable set derived from expressiveness, then a lower bound can be provided to show that the test error of the min $\ell\_2$-norm overfitted solution will not go to zero even when there is no noise and when $n,p\to \infty$. Therefore, for our analysis on 3-layer NTK, we believe that the method of defining the learnable set by its expressiveness is also reasonable.
>
> [Ju, P., Lin, X., and Shroff, N. (2021, July). On the generalization power of overfitted two-layer neural tangent kernel models. In International Conference on Machine Learning (pp. 5137-5147). PMLR.]
>
> ---
>
> > *In Theorem 1, the authors claim that the result holds with a high probability? Can they provide a formal estimation of the probability in question?*
>
> Yes. As mentioned in Line 208-209, a more precise version of the upper bound, with a clear statement of the meaning of high probability, can be found in Supplementary Material, Appendix C.
>
> ---
>
> > *Can the authors explain the observation of essentially zero reduction in the test error for $\sigma^2=0.1,0.4$ in Fig 2 (b)?*
>
> We believe such phenomenon is because the noise is relatively large, which leads to an error floor. In other words, the performance gain brought by larger $p\_1$ is dominated by the "noise floor" (and thus is not observable in the figure). Indeed, still in Fig 2(b), by using smaller noise of $\sigma^2=10^{-4},0$ (red curve and green curve), we can see the descent trend stretching to larger values of $p\_1$.

---

### Official Review · Reviewer_sEV3 · 2022-07-15

**Rating:** 5
**Confidence:** 3
**Soundness:** 2 fair
**Presentation:** 2 fair
**Contribution:** 2 fair

**Summary:**

The authors studied generalization error of 3-layer neural networks with two caveats (1) only the middle layer weights are trained (2) only a class of regression functions are considered. An upper bound is provided in Theorem 1, with interesting dependence on the second hidden layer width.

Overall, I had some trouble reading the paper and trying to understand the contributions. I hope the authors can join me during the discussion period and help me clarify some of my questions, after which I would be happy to raise my scores if we reach a clearer understanding.

**Questions:**

I understand this work is highly technical and perhaps not the easiest to present in a straight forward manner. So I hope the authors can help clarify many of my questions, as I cannot make a confidence judgement based on my first read of the paper.

# Major Questions

1. The authors did not precisely define the training dynamics considered, in particular the variable $\overline{\Delta W}$ on line 140 is undefined. Can the authors clarify this part? I believe this is really important.

2. The authors introduced some heavy notation, which is possibly well motivated, but I am having a lot trouble interpreting. Can the authors give a high level intuition and motivation behind the following definitions?

2.(a) For weight matrices and hidden layers, why do the authors avoid using regular matrix multiplication? That why define $h^{RF}_{V,x}[ j ]$ based on $x^\top V[j] $ instead of just using $ ( V x )_+ $? Is there a distinct advantage for treating the elements separately?

2.(b) How should I interpret $h^{Three}_{V, W_0, x}[k]$? This object contains a first layer random feature, but also the activation pattern of the next layer as well?

3. Perhaps because of the previous question, I don't understand why "overfitted gradient descent" converges to the min $\ell_2$-norm solution defined by $\Delta W^\{\ell_2}$. At the same time, can the authors motivate the $\Delta$ notation here? It seems me that this is just the optimal weights? This is also confusing because if we train the second layer weights, wouldn't the activation pattern on the next layer also change as a result, therefore leading to a non-linear model?

4. The authors defined $\mathcal{F}^{\ell_2}_{(3)}$ as the set of learnable functions. Maybe I'm misunderstanding this, can the authors explain if this definition is motivated by heuristics, or if there is a concrete learnability statement somewhere in the paper?

5. I have several questions about the upper bound in Theorem 1.

5.(a) It appears that the product of term D and E gets larger as $n$ increases, which leads to the question: is there a step that is loose, or does $p_1$ need to significantly larger than $n$ in order for the bound to vanish?

5.(b) In Ghorbani et al. (2019), the paper showed that polynomials of higher degrees require a higher sample complexity to learn for random feature models. However, in the current work, the authors suggested that the learnable set of functions contains all polynomials of finite degrees. Can the authors explain which part is the key difference in the settings that led to this discrepancy?

5.(c) The authors drew the conclusion that the second hidden layer is more important from this upper bound. However, do the authors authors have any insights towards whether or not this upper bound is tight in terms of $p_1,p_2$? Otherwise, I'm sure the authors understand, I can always make an upper bound worse to draw the opposite conclusion following this line of logic.

6. The authors showed that the learnable set is strictly larger than $\mathcal{F}^{\ell_2}_{(2)}$, which is for the two layer NTK. Do the authors have any insights towards a negative result, i.e. are there actually functions contained in $\mathcal{F}^{\ell_2}_{(3)}$ such that a three layer network can learn, but a two layer NTK cannot? Because otherwise again, I'm sure the authors understand, I can modify the definitions to draw the opposite conclusion.

# Minor Questions

1. Can the authors motivate the uniform distribution assumptions made on the weights, and explain what roles this plays in the proof? This seems pretty non-standard, and adds towards my concern on how useful the conclusions are given an additional caveat.

2. The authors defined $K^{RF}, K^{Two}, K^{Three}$ based on the RF kernel and NTK kernel, however with extra factors of $d$ in the definitions. Is this important?

3. In equation 7, what is the measure $\mu$?

**Strengths And Weaknesses:**

# Strengths
1. The generalization error bound contains an interesting dependence on layer width, which may be of interest for understanding deeper networks outside of the kernel regime.

# Weaknesses
1. I had some trouble reading the paper and understanding the contributions.
2. The results the authors achieved contain enough caveats that I'm not sure if we can really draw strong conclusions as the authors have claimed.

---

> ### Author Response · Authors · 2022-08-02
> **Response to Reviewer sEV3 (Part 1)**
>
> Thank you. We will further attempt to make the paper more readable and address your questions as below.
>
> ---
>
> > *The authors did not precisely define the training dynamics considered, in particular the variable $\overline{\Delta W}$ on line 140 is undefined. Can the authors clarify this part? I believe this is really important.*
>
> We define the $\overline{\Delta W}$ as the change of middle-layer weights after training, i.e., $\overline{\Delta W}= W\_1-W\_0$. Note that on Line 140 we do not need to specify the training dynamics. The reason is that we later consider an overfitted solution to a linear model, which means that the trained model can be written down directly (see later by Line 149). Please also see our response to the comment about overfitted gradient descent below.
>
> ---
>
> > *For weight matrices and hidden layers, why do the authors avoid using regular matrix multiplication? That why define $h\_{V,x}^{RF}[ j ]$ based on $x^\top V[j]$ instead of just using $( V x )\_+ $? Is there a distinct advantage for treating the elements separately?*
>
> Compared with regular matrix multiplication, we treat the elements separately to clearly show the effect of ReLU for each neuron. For example, the tangent of a RELU neuron $w^T x \cdot \mathbf{1}\_{\\{w^T x \ge 0\\}}$ is $x\cdot  \mathbf{1}\_{\\{w^T x \ge 0\\}}$, which is the form that we are using. Notice that we only treat the elements of $h^{RF}\_{V,x}$ separately once in Eq. (1), while in the rest of the main text of this paper we always treat $h\_{V,x}^{RF}$ as a whole. We thank the reviewer for pointing out the possible confusion and we will add additional explanation for the equivalence to the regular matrix multiplication.
>
> ---
>
> > *How should I interpret $h\_{V,W\_0,x}^{Three}[k]$? This object contains a first layer random feature, but also the activation pattern of the next layer as well?*
>
> Yes, it is of this form because we train the middle-layer weights. The first-layer random-feature output feeds into the second layer. The tangent of the second layer (including the activation) becomes the ultimate feature vector based on the activation pattern of the second hidden-layer. This explains why $h\_{V,W\_0,x}^{Three}[k]$ is formed by both the first-layer random features and the activation pattern of the second hidden-layer. As mentioned in Line 146-147, we provide an illustration of the formation and structure of these vectors in Fig. 4, Appendix A.1 in Supplementary Material.
> Notice that $h\_{V,x}^{RF}[j]$ is a scalar, while  $h\_{V,W\_0,x}^{Three}[k]$ is a vector consisting of $p\_1$ elements (so $h\_{V,W\_0,x}^{Three}$ is vector of size $(p\_1p\_2)\times 1$).
>
> ---
>
> > *I don't understand why "overfitted gradient descent" converges to the min $\ell\_2$-norm solution defined by $\Delta W^{\ell\_2}$. Can the authors motivate the notation $\Delta$ here? It seems me that this is just the optimal weights? This is also confusing because if we train the second layer weights, wouldn't the activation pattern on the next layer also change as a result, therefore leading to a non-linear model?*
>
> Generally speaking, if we do gradient descent training on a linear model from zero initial point until the training error is zero (i.e., overfitting), then the solution will be exactly the min $\ell\_2$-norm solution. (This fact is also mentioned in the literature, e.g., [Satpathi et al 2021].) To see this, consider a simple example that uses gradient descent to minimize the training error $\lVert y-Xa\rVert\_2^2$, where each row of $X$ is one training sample. Because the initial point is zero, after each iteration/update the solution vector $a$ must always be in the row space of $X$ (since the gradient is in the row space of $X$), i.e., $a=X^Tb$ for some vector $b$. Therefore, when overfitting occurs (i.e., training error becomes zero), we have $y=Xa=XX^Tb$, which implies that $b=(X^TX)^{-1}y$ and thus $a=Xb=X^T(XX^T)^{-1}y=X^{\dagger}y$. As we can see, this is exactly the min $\ell\_2$-norm solution.
>
> The notation $\Delta W[k]$ is given in Line 143, which is the top-layer weight $w\_k$ (untrained) multiplied by the change of second-layer weight $\Delta\overline{W}[k]$, $k=1,2,\cdots,p\_2$. The key assumption of NTK model is that the activation pattern does not change much, in which case the model can be approximated by a linear model (as we described in Line 141-142). We apply this assumption to the second hidden-layer. For the top layer, it is a linear sum. Hence, there is no issue of activation patterns.
>
> [Satpathi, Siddhartha, and Rayadurgam Srikant. "The Dynamics of Gradient Descent for Overparametrized Neural Networks." Learning for Dynamics and Control. PMLR, 2021.]
>
> ---
>
> **Followed by "Response to Reviewer sEV3 (Part 2)".**

---

> > ### Comment · Reviewer_sEV3 · 2022-08-08
> > **Response to the Authors**
> >
> > Thank you for the detailed response. I believe the authors have addressed some of my confusions while reading the paper, but I remain concerned on a high level regarding how I can interpret the contributions of this paper.
> >
> > In particular, the authors seem to draw conclusions based on a number of caveats:
> > 1. a somewhat heuristic definition of a learnable function class
> > 2. training only the middle layer
> > 3. interpreting an upper bound
> >
> > As such, I do not feel very convinced that
> > 1. the widths of different layers play significantly different roles
> > 2. adding a third layer in the NTK helps beyond breaking symmetries for odd polynomials
> >
> > While I remain open to raising my score if the authors can convince me that I am misinterpreting the paper and the claims, I do want to say I am skeptical. I believe making modifications to a standard setup and drawing conclusions based on an upper bound discounts the value of all the hard analysis in this work, and I hope the authors can understand my concerns.
> >
> > By all means, I welcome additional comments from the authors, and hopefully can address this concern of mine.

---

> > > ### Author Response · Authors · 2022-08-09
> > > **Response to Reviewer sEV3**
> > >
> > > Thanks for the comments.
> > >
> > > **"Concerns about the high-level contribution"**: The high-level contribution is that this is the first work that studies the generalization performance on the overfitted 3-layer NTK, especially with finite width. By providing an upper bound, we rigorously prove that the generalization error of the overfitted solution will descend to zero as the number of neurons increases, if the ground truth is in the learnable set. From a technical aspect, we analyze the interaction between two hidden-layers by training the middle layer, which is quite different from, and is more complex than the 2-layer NTK where there is only one hidden layer. The implications of our results are also supported by numerical results consistent with the theoretical results.
> > >
> > > For your detailed concerns:
> > >
> > > 1. **"Heuristic definition of learnable function"**: We respectfully disagree. The definition of the learnable function class is not heuristic. First, the learnability has been rigorously proven by our upper bound (i.e., the generalization error for those learnable ground-truth functions approaches zero in the ideal situation). Second, the expression of the learnable set is derived rigorously in Appendix B. In other words, all functions that the overfitted NTK model can express are of this form, as $n, p \to \infty$. Third, as we mentioned in the previous response, a similar definition of the learnable set for 2-layer NTK can be found in [Ju et al '21], which not only proves the learnability for functions in the learnable set, but also proves the non-learnability of functions outside the learnable set. Therefore, such a definition of the learnable set is well supported (and not heuristic).
> > > 2. **"Training only the middle layer"**: We acknowledge that our results are shown only for the current setup, where only the middle layer is trained. Nonetheless, our results still provide valuable insights that have not been reported by other papers (regardless of which layer is trained) before. In particular, our insights on the learnable sets, the impact of bias, and the descent speed are all new. Besides, training only one layer of a neural network is not uncommon when theoretically analyzing neural networks. For example, the well-known random feature (RF) model only trains the top layer weights of a two-layer neural network.
> > > 3. **"Interpreting an upper bound, the widths of different layers play significantly different roles"**: Although our insights are based on the upper bound of the generalization error, they are supported by simulation results consistent with theoretical results (additional simulations can be found in Appendix A). For example, both the simulation results and the upper bound expression suggest different speeds with which the generalization error descends with respect to $p\_1$ and $p\_2$.
> > > 4. **"Will adding a third layer in NTK helps?"**: We agree that our current results on the advantage of 3-layer NTK are mostly related to the odd polynomials and the impact of bias. We are also curious whether there are other advantages of 3-layer NTK that are not reported in our paper. Note that there may indeed be no other advantages. One possible reason could be the NTK regime (near the initial point) itself imposes a strong restriction on the structure of the solution, so that other differences (e.g., the exact form of the NTK of 2-layer or 3-layer) do not matter as much. Therefore, the lack of “other” advantages of 3-layer NTK should not be viewed as a shortcoming of our work. Instead, by conducting the first such study of 3-layer NTK in the literature, our results already reveal non-trivial differences between 3-layer NTK and 2-layer NTK. Further, our work could provide valuable motivations for research beyond the NTK regime, to fully understand the impact of depth on neural-network performance.
> > > 5. **"Making modifications to a standard setup"**: We are not sure what "modifications to a standard setup" are of concern to the reviewer. To derive a theoretical result, some simplifications and assumptions are inevitable, and the ones we used have also been used in other papers in the field of overfitted generalization performance analysis. We have also tried to address the reviewer's concerns on some assumptions (see our previous response). If the reviewer still has questions on any specific assumption, we are also happy to explain more.
> > >
> > > In summary, although our paper does not provide a complete answer, it is a first step in the literature to understand the overfitted generalization performance of 3-layer NTK. As mentioned by the reviewer, there exists room for considering other setup and providing matching lower bounds, which we agree and will be interesting directions for future work. However, even for our current setup, our analysis of 3-layer NTK is already technically involved, and our results already reveal non-trivial insights. We hope this response addresses your concerns on our contribution at a high level.

---

> ### Author Response · Authors · 2022-08-02
> **Response to Reviewer sEV3 (Part 2)**
>
> **Continued from "Response to Reviewer sEV3 (Part 1)"**
>
> ---
>
> > *The authors defined $\mathcal{F}\_{(3)}^{\ell\_2}$ as the set of learnable functions. Maybe I'm misunderstanding this, can the authors explain if this definition is motivated by heuristics, or if there is a concrete learnability statement somewhere in the paper?*
>
> The set of learnable functions is from the expressiveness of the learned model. We explain this in Line 185-202. In short, the learnable set is derived by the expressiveness of the corresponding NTK with infinite width. If a function is learnable, then it at least should be able to be expressed by an infinite width NTK (more neurons usually mean stronger expressiveness). Therefore, we believe those expressible functions are the only reasonable candidates of learnable functions. We then use Theorem 1 to prove that those functions can indeed be learnt (i.e., the generalization error goes to zero under ideal situation). We note that similar definition of learnable sets has been used in [Ju et al '21] for 2-layer NTK without bias. Specifically, [Ju et al '21] shows that, for 2-layer NTK without bias, if a function is at a positive distance away from the learnable set derived from expressiveness, then a lower bound can be provided to show that the test error of the min $\ell\_2$-norm overfitted solution will not go to zero even when there is no noise and when $n,p\to \infty$. Therefore, for our analysis on 3-layer NTK, we believe that the method of defining the learnable set by its expressiveness is also reasonable.
>
> [Ju, P., Lin, X., and Shroff, N. (2021, July). On the generalization power of overfitted two-layer neural tangent kernel models. In International Conference on Machine Learning (pp. 5137-5147). PMLR.]
>
> ---
>
> > *I have several questions about the upper bound in Theorem 1. 5.(a) It appears that the product of term D and E gets larger as $n$ increases, which leads to the question: is there a step that is loose, or does $p\_1$ need to significantly larger than $n$ in order for the bound to vanish?*
>
> Indeed, the product of terms D and E does increases with $n$ in our current bound. This is actually expected, i.e., the model error tends to increase when the number of samples increases. Our experiments also shows consistent behavior, i.e., Fig. 8(a) in Appendix D shows that (in the case of finite $p\_1$ and $p\_2$) the noise effect increases when the number of samples increases. An intuitive explanation is that the overfitted solution needs to completely fit the noise component of all training samples, and thus more data also means more noise to fit (and thus contributes to larger model error). We discuss the relationship between the noise effect and the parameters $p\_1$, $p\_2$ and $n$ (along with additional simulation results) in Appendix D (due to the page limit of main text). On the other hand, our current forms of terms D and E may still have room to be tightened, as we discussed in Appendix D, Line 560-564: "We notice that Term E increases with $n$ at a speed faster than $\sqrt{n}$. However, since it is only an upper bound, the actual noise effect may grow much slower than $\sqrt{n}$. Therefore, precisely estimating the relationship between $n$ and the noise effect of NTK model would be an interesting future research direction."
>
> ---
>
> > *5.(b) In [Ghorbani et al. (2019)], the paper showed that polynomials of higher degrees require a higher sample complexity to learn for random feature models. However, in the current work, the authors suggested that the learnable set of functions contains all polynomials of finite degrees. Can the authors explain which part is the key difference in the settings that led to this discrepancy?*
>
> Our upper bound is still consistent with the intuition that more complex ground-truth function needs more samples. Although both low degree polynomials and high degree polynomials are in the learnable set, the magnitude of the corresponding $g$ will be different, which leads to different generalization performance. Roughly speaking, higher degree polynomials have larger $g$, so we need a larger $n$ to achieve the same value of Term A in our upper bound Eq. (8).
>
> ---
>
> **Followed by "Response to Reviewer sEV3 (Part 3)".**

---

> ### Author Response · Authors · 2022-08-02
> **Response to Reviewer sEV3 (Part 3)**
>
> **Continued from "Response to Reviewer sEV3 (Part 2)"**
>
> ---
>
> > *5.(c) The authors drew the conclusion that the second hidden layer is more important from this upper bound. However, do the authors authors have any insights towards whether or not this upper bound is tight in terms of $p\_1,p\_2$? Otherwise, I'm sure the authors understand, I can always make an upper bound worse to draw the opposite conclusion following this line of logic.*
>
> Thank you for your comment. We agree with this limitation due to the lack of a matching lower bound. While we have not shown tightness of the upper bound, the numerical results that we have provided agree with the intuition of the upper bound. See the paragraph in Line 249-258.
>
> ---
>
> > *The authors showed that the learnable set is strictly larger than $\mathcal{F}\_{(2)}^{\ell\_2}$ which is for the two layer NTK. Do the authors have any insights towards a negative result, i. e. are there actually functions contained in $\mathcal{F}\_{(3)}^{\ell\_2}$ such that a three layer network can learn, but a two layer NTK cannot? Because otherwise again, I'm sure the authors understand, I can modify the definitions to draw the opposite conclusion.*
>
> Yes, there is a distinct difference between $\mathcal{F}\_{(3)}^{\ell\_2}$ and $\mathcal{F}\_{(2)}^{\ell\_2}$. For example, odd-power polynomials except linear functions (e.g., $f(x)=(x^T e\_1)^3$) cannot be learnt by 2-layer NTK without bias (i.e., not belong to $\mathcal{F}\_{(2)}^{\ell\_2}$), but can be learnt by 3-layer NTK without bias (i.e., belong to $\mathcal{F}\_{(2)}^{\ell\_2}$). Detailed comparison can be found in Table 1. On the other hand, with bias, we do not know yet whether there is a non-empty difference in the size of the learnable sets between 2-layer NTK and 3-layer NTK. As stated in Proposition 2, We can only show that the learnable set of 3-layer NTK is at least as large as 2 layer.
>
> ---
>
> > *Can the authors motivate the uniform distribution assumptions made on the weights, and explain what roles this plays in the proof? This seems pretty non-standard, and adds towards my concern on how useful the conclusions are given an additional caveat.*
>
> We use the uniform distribution on a sphere to make the proof simpler. We believe it is very close to the standard Gaussian distribution since both distribution has uniformly distributed direction in the hyperspace. The only difference is that the uniform distribution has an additional normalization process that makes the vector's norm/length constant. When the number of elements of the random vector $a\in \mathbb{R}^{k}$ is large, the scaled norm $\frac{\lVert a\rVert\_2}{\sqrt{k}}$ of a Gaussian distributed vector becomes more concentrated on its mean (by law of large numbers) and thus the difference between a uniform distributed vector and a Gaussian distributed vector should become negligible.
>
> ---
>
> > *The authors defined $K^{RF},K^{Two},K^{Three}$ based on the RF kernel and NTK kernel, however with extra factors of $d$ in the definitions. Is this important?*
>
> The factor $d$ is just a normalization factor (as the norm of input of each sample equals to $\sqrt{d}$ by Assumption 1).
>
> ---
>
> > *In equation 7, what is the measure $\mu$?*
>
> We use $\mu(\cdot)$ to denote the distribution of input $x$. (See Line 154.)

---

### Meta-Review · Area_Chair_uEVQ · 2022-08-27

**Recommendation:** Accept
**Confidence:** Certain

**Metareview:**

This paper studies the generalization error of three-layer relu neural networks, when only the middle layer weights are trained.
The focus is the regression setting, and the goal is to capture the hidden layer interactions. The paper aims to determine how the (hidden) layer interactions influence the double-descent curve. The generalization error bound established depends on the layer width in an interesting manner, which may shed light on understanding deeper networks outside of the kernel regime.

All reviewers rated this work above the bar. As such, I recommend accepting this paper.

There were a few parts that the reviewers found unclear/needs improvement. In particular, some of the clarifications made by the authors in their rebuttal can help make this paper more clear for its future readers.




**Award:**

No

---

### Decision · Program_Chairs · 2022-09-14

Accept